# Clonal dynamics after allogeneic haematopoietic cell transplantation





Michael Spencer Chapman[1,2,3], C. Matthias Wilk[4], Steffen Boettcher[4], Emily Mitchell[1,2,3], Kevin Dawson[1], Nicholas Williams[1], Jan Müller[4], Larisa Kovtonyuk[4], Hyunchul Jung[1], Francisco Caiado[4], Kirsty Roberts[1], Laura O'Neill[1], David G. Kent[2,3,5], Anthony R. Green[2,3], Jyoti Nangalia[1,2,3], Markus G. Manz[4]✉ & Peter J. Campbell[1,2,3]✉

Allogeneic haematopoietic cell transplantation (HCT) replaces the stem cells responsible for blood production with those from a donor[1,2]. Here, to quantify dynamics of long-term stem cell engraftment, we sequenced genomes from 2,824 single-cell-derived haematopoietic colonies of ten donor–recipient pairs taken 9–31 years after HLA-matched sibling HCT[3]. With younger donors (18–47 years at transplant), 5,000–30,000 stem cells had engrafted and were still contributing to haematopoiesis at the time of sampling; estimates were tenfold lower with older donors (50–66 years). Engrafted cells made multilineage contributions to myeloid, B lymphoid and T lymphoid populations, although individual clones often showed biases towards one or other mature cell type. Recipients had lower clonal diversity than matched donors, equivalent to around 10–15 years of additional ageing, arising from up to 25-fold greater expansion of stem cell clones. A transplant-related population bottleneck could not explain these differences; instead, phylogenetic trees evinced two distinct modes of HCT-specific selection. In pruning selection, cell divisions underpinning recipient-enriched clonal expansions had occurred in the donor, preceding transplant—their selective advantage derived from preferential mobilization, collection, survival ex vivo or initial homing. In growth selection, cell divisions underpinning clonal expansion occurred in the recipient's marrow after engraftment, most pronounced in clones with multiple driver mutations. Uprooting stem cells from their native environment and transplanting them to foreign soil exaggerates selective pressures, distorting and accelerating the loss of clonal diversity compared to the unperturbed haematopoiesis of donors.

Performed first in 1956 and routinely since the 1970s, allogeneic HCT is used to replace a defective haematopoietic system[1] or to treat haematological cancers[2]. When treating malignancy, the goal is to fully replace the recipient's haematopoietic system with that of the donor and harness the transplanted immune system to kill malignant cells in the recipient[2]. Fundamental questions about the biology of HCT remain, such as how many transplanted haematopoietic stem/progenitor cells (HSPCs) maintain blood production; how different stem cell clones contribute to the various mature blood cell compartments; why recipients have elevated morbidity and mortality even decades after transplant[4,5]; and why older age of donor or recipient is associated with worse outcomes[6,7].

Much of our understanding of stem cell dynamics in transplantation comes from experiments in model organisms based on clone-tracking methods such as retroviral integration sites[8], lentiviral barcodes[9,10], transposon tagging[11,12], or CRISPR–Cas9 and Cre-Lox-induced editing[13,14]. Direct human studies are limited by a paucity of applicable methodologies. One exception is the tracking of vector integration sites in gene therapy trials, which has provided estimates of engrafting haematopoietic stem (HS) cell numbers in this autologous setting[15,16]. Most clone-tracking approaches barcode cells at a single timepoint, usually at or immediately before transplant. While this facilitates quantification of the output of individual transplanted HS cells, these approaches are blind to any pre-existing clonal structure that could influence engraftment.

Spontaneous somatic mutations can be used as dynamic lineage markers. They are acquired at a constant, clock-like rate throughout life[17–19], meaning that they can be used to infer lineage relationships all the way back to foetal development. This principle has been used to quantify clonal dynamics of the haematopoietic system through the healthy lifespan[18–20] and in disease[21,22]. We used genome-wide somatic mutations to reconstruct the phylogeny of the haematopoietic system within matched HCT recipient and donor pairs, using samples taken a decade or more after the HCT procedure. This enabled us to quantify

[1]Cancer, Ageing and Somatic Mutation Programme, Wellcome Sanger Institute, Hinxton, UK. [2]Wellcome–MRC Cambridge Stem Cell Institute, Jeffrey Cheah Biomedical Centre, Cambridge, UK. [3]Department of Haematology, University of Cambridge, Cambridge, UK. [4]Department of Medical Oncology and Hematology, University of Zurich and University Hospital Zurich, Zurich, Switzerland. [5]York Biomedical Research Institute, University of York, Wentworth Way, York, UK. ✉e-mail: Markus.Manz@usz.ch; pc8@sanger.ac.uk

the long-term impact of HCT on blood production by contrasting clonal dynamics in the recipient with the native, unperturbed haematopoiesis of the donor.

## WGS analysis of HSPC colonies

We obtained samples from ten fully HLA-matched sibling donor and recipient pairs who had been recruited for a previous study[3] (Fig. 1a). In each case, the recipient had undergone HCT many years before sampling (range, 9–31 years) and had complete replacement of their haematopoietic system with that of the donor. The most common indication for HCT was acute myeloid leukaemia; the conditioning regimen was myelo-ablative ($n = 7$) or reduced intensity ($n = 3$); the stem cell source was bone marrow ($n = 5$) or mobilized peripheral blood ($n = 5$); and recipients were of similar age to their sibling donors (age difference: −7 to +11 years; Extended Data Table 1).

Peripheral-blood-derived $CD34^+$ HSPCs were used to seed single-cell-derived colonies in methylcellulose medium. For each of the 20 individuals, whole-genome sequencing (WGS) to a mean depth of 11.5× was performed on 96–230 colonies, a total of 3,399 whole genomes (Supplementary Fig. 1a). We excluded 46 colonies with low coverage, 58 technical duplicates, 10 derived from a different germline (likely contamination) and 468 that were non-clonal, leaving a final dataset of 2,824 genomes (Supplementary Fig. 1b). There was no evidence of residual recipient-derived haematopoiesis, as evidenced by germline polymorphisms in the colonies or embryonic mutations in deep targeted sequencing data (Supplementary Fig. 1c).

As both donor and recipient samples originated from the donor haematopoietic system, somatic mutations could be used to reconstruct phylogenetic trees for donor and recipient separately (Fig. 1b and Extended Data Fig. 1), as well as in a single, combined tree (Fig. 1c and Extended Data Fig. 2).

## Mutation burden and signatures after HCT

Somatic single-nucleotide variants (SNVs) in HSPCs have consistent, endogenous mutational signatures[17,19]. Using de novo signature decomposition, we found that the endogenous, clock-like signatures seen in HSPCs predominated in both donors and recipients (Extended Data Fig. 3a,b). Two HCT recipients who had received platinum-based chemotherapy after HCT carried SBS31, attributed to platinum agents[23], contributing a mean of 184 and 162 mutations per HSPC (Extended Data Fig. 3c). None of the patients received ganciclovir, so we did not observe the mutational signature previously described with this antiviral therapy[24].

Notably, we found a signature of APOBEC genome editing in 63 out of 2,824 HSPCs (2.2%), usually contributing hundreds or even thousands of additional mutations (Extended Data Fig. 3d,e)—almost all (61 out of 63, 97%) of these HSPCs were from recipients, not donors, and were restricted to the peri- or post-transplant periods (Extended Data Fig. 4a). The occurrence of APOBEC mutations was not predicted by presence of driver mutations, the type of conditioning, donor sex, the source of stem cells or whether the cell was in a clonal expansion (Extended Data Fig. 4b). Across the genome, APOBEC mutations were more likely to occur near cruciform inverted repeats, as described previously[25], and in regions with higher Alu repeat density or lower GC content (Extended Data Fig. 4c). APOBECs are an important component of the host defence systems against viruses, and there is evidence that viral infection and interferon signalling can induce APOBEC activity[26]. In the HCT setting, recipients are routinely immunosuppressed and are therefore prone to opportunistic infections, so such a mechanism could explain the recipient bias and post-transplant timing observed here.

In donors, the overall burden of point mutations accumulated linearly with age at a rate of 15.8 per HSPC per year (95% confidence interval (CI) = 12.8–18.7; $P = 6 \times 10^{-6}$; linear mixed-effect regression; Extended Data Fig. 5a), consistent with previous findings[17,19]. To assess whether the HCT procedure itself causes additional mutation, we assessed mutation burdens of donors and their recipients, after removing contributions from the sporadic APOBEC and platinum signatures. We estimated that, on average, recipient HSPCs had around 23 excess mutations (95% CI = 7–37; $P = 0.005$), equivalent to around 1.5 years of normal ageing. However, this difference was not consistent across pairs, with half of pairs having very similar donor and recipient mutation burdens (Extended Data Fig. 5b). The finding that HCT causes, at most, relatively small numbers of additional somatic mutations is consistent with results from cord blood transplants[24].

Across the 10 pairs, we found 71 independent mutations that were probably driver mutations. Consistent with previous studies[27–29], the most frequent genes were *DNMT3A* ($n = 23$) and *TET2* ($n = 9$), with *CHEK2*, *BCOR* and *TP53* each having 4 mutations (Extended Data Fig. 5c). *DNMT3A* mutations were unevenly distributed among donor–recipient pairs, with one pair containing eight independent hits (Extended Data Fig. 5d). On the basis of the estimated age of branches in the phylogenetic tree, three *DNMT3A* mutations were probably acquired in utero, as observed in clonal haematopoiesis[21] and myeloproliferative neoplasms[22] (Extended Data Fig. 5e).

Autosomal copy-number alterations (CNAs) and structural variants (SVs) were rare, affecting 0.5% ($n = 14$) and 0.8% ($n = 23$) of HSPCs, respectively (Supplementary Fig. 2a,b). Autosomal CNAs comprised either copy-neutral loss of heterozygosity ($n = 8$) or duplications ($n = 6$). SVs were predominantly deletions ($n = 7$), inversions ($n = 7$) or reciprocal translocations ($n = 6$). Recipient HSPCs were more likely to have an SV or autosomal CNA (Pearson's $\chi^2$ test, $P = 0.03$; Supplementary Fig. 2c). Pair 9 had highly prevalent loss of the Y chromosome, present in 200 out of 430 of HSPCs (46.5%) and representing at least 9 independent events enriched in expanded clones (Supplementary Fig. 2d). This suggests that Y chromosome loss confers a positive selective advantage, possibly through the Y-linked tumour suppressor gene *KDM6C*[30].

## Numbers of long-term engrafting HSPCs

Population bottlenecks, which might be caused by collecting and transplanting a small proportion of a donor's HSPCs, leave characteristic patterns in a phylogeny. As seen from simulation, tighter population bottlenecks increase the number of coalescences (Supplementary Fig. 3). In our setting, a tighter bottleneck implies fewer long-term engrafting stem cells, confirming that we should be able to estimate the numbers of engrafting cells from the distribution of coalescences in the recipient compared to the donor.

Using the estimated mutation rate in HSPCs during adult life, we scaled the phylogenetic trees from molecular time to chronological time (Fig. 1c and Extended Data Fig. 2). Reassuringly, superimposing the time of HCT onto these phylogenies showed that, first, all mutations shared by both donor and recipient colonies were assigned to branches preceding the estimated molecular time of HCT and, second, there were increased coalescences at or after the time of HCT in many recipient phylogenies compared with the donor (Fig. 1b and Extended Data Fig. 1). This latter observation suggests that transplantation does cause a measurable population bottleneck.

To estimate the numbers of long-term engrafting HSPCs ($n_{HSC}$) in each recipient, we used approximate Bayesian computation (ABC), based on simulations of ageing haematopoiesis with regular driver acquisition[19] that included a superimposed population bottleneck of varying sizes. We then compared the simulated phylogenetic structure with that of our observed data (Extended Data Fig. 6). Median posterior estimates for $n_{HSC}$ ranged from 700 to 25,000 across donor–recipient pairs (Fig. 2a). The higher estimates are similar to those from the autologous gene therapy setting[31], despite HCT engraftment facing additional challenges of allo-immunity and a host bone marrow

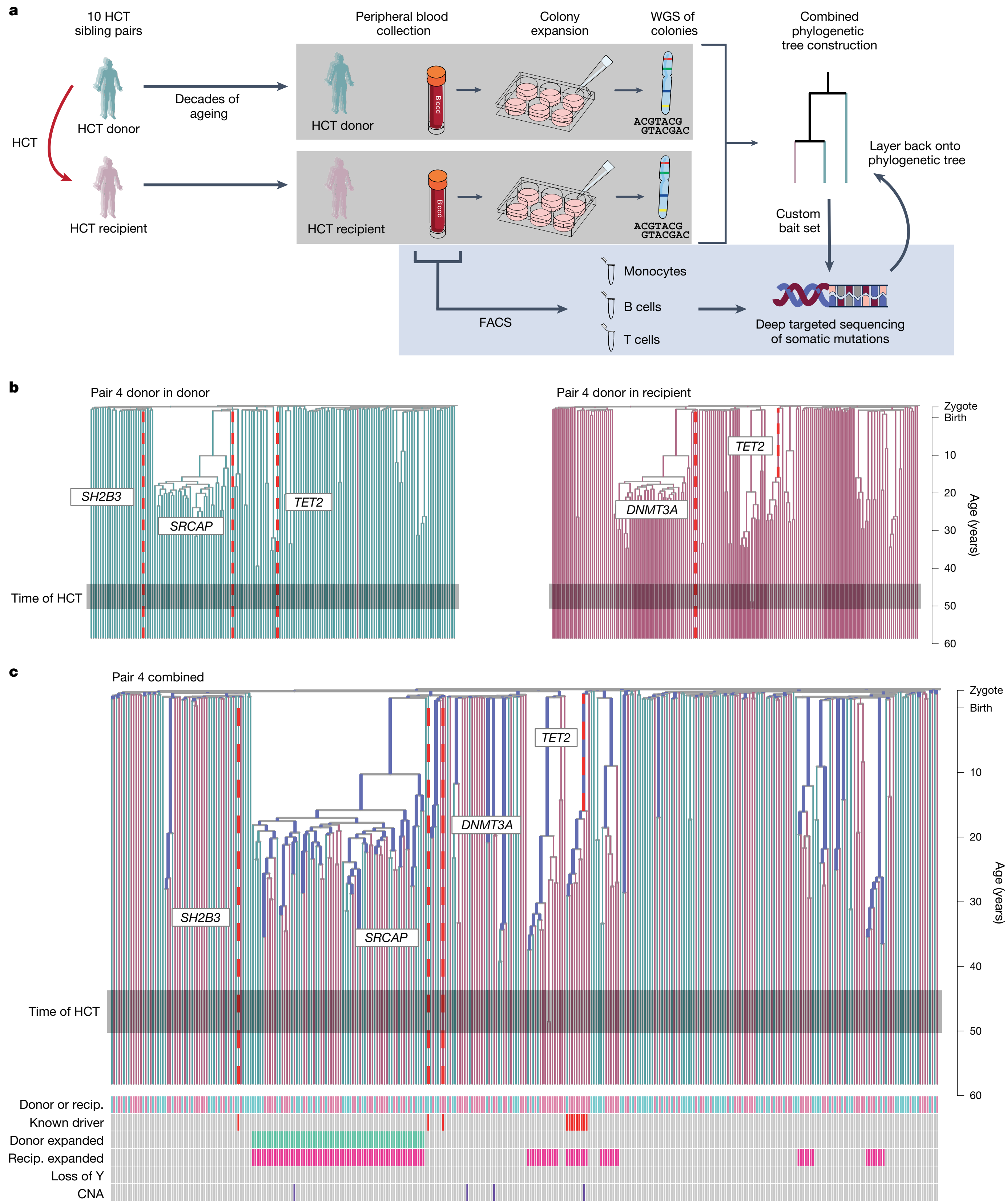


**Fig. 1 | Experimental design and phylogeny building. a**, The study outline. Blood was sampled from ten sibling pairs who had been the donor and recipient of HCT years previously (range, 9–31 years). CD34$^+$ cells were used to seed colonies in culture medium. Single colonies were analysed using WGS (298–430 per pair), and somatic mutations were used to reconstruct phylogenies. Mature cell subsets were sorted using fluorescence-activated cell sorting (FACS) and underwent custom targeted sequencing for the mutations found in WGS. **b**, Illustrative separate donor-in-donor/donor-in-recipient phylogenies built from samples from each individual from a pair. Branches with putative driver mutations (red dashed lines) are labelled with the variant. Time of HCT is indicated by the grey box. Branch lengths are scaled to chronological time. **c**, As in **b**, but combined into a single phylogeny. Branches were found in donor colonies only (cyan), recipient colonies only (pink) or both (black). The heat map shows additional colony-level information. Recip., recipient.

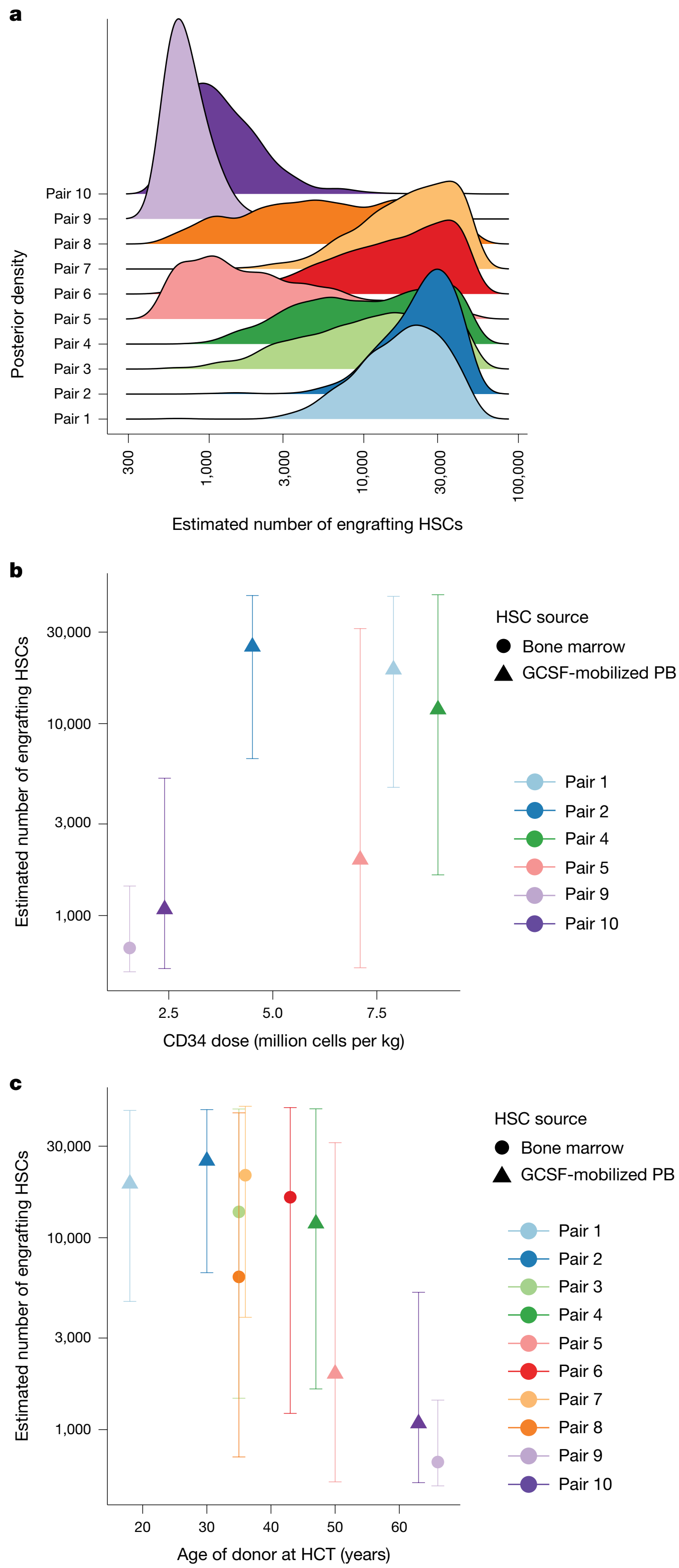


**Fig. 2 | The numbers of long-term engrafting haematopoietic stem cells.** **a**, The posterior distributions for the number of long-term engrafting HS cells for each HCT, as estimated by approximate Bayesian computation (Methods). **b**, The relationship between the number of engrafting HS cells and the infused $CD34^+$ cell dose per kg of recipient body weight. The points show the median posterior value, and error bars show the 95% posterior intervals, calculated from the 1% of $n$ = 100,000 simulations of which the summary statistics best matched the observed data. $CD34^+$ cell dose was not available for pairs 3, 6, 7 or 8. **c**, As in **b**, but illustrating the relationship between the numbers of engrafting HS cells and the age of the donor at the time of HCT. GCSF, granulocyte colony-stimulating factor; PB, peripheral blood.

damaged by chemotherapy and leukaemia. While the small number of donor–recipient pairs precludes definitive statements, some possible associations did emerge. Estimates of the numbers of engrafting stem cells were higher with donors who had younger ages at transplant ($P$ = 0.001; mixed-effects models), but minimal correlation with counts of infused $CD34^+$ cells ($P$ = 0.14; Fig. 2b,c). Within this small cohort, there was no evidence of correlation with stem cell source or conditioning type ($P$ = 0.99 and $P$ = 0.23, respectively).

## Loss of clonal diversity after HCT

With age, normal haematopoiesis loses clonal diversity such that, after the age of 70 years, as few as 10–20 dominant clones account for 30–60% of all blood production[19,21]. The mutations driving these clones are typically acquired in childhood to early adulthood, triggering decades of slow but exponential expansion. Consistent with these findings, phylogenies from older donor–recipient pairs demonstrated higher proportions of haematopoiesis derived from expanded clones (those contributing ≥2% of haematopoiesis). Across the pairs, there were 75 independent expansions, with only 20% containing known driver mutations (Fig. 3a).

Compared with their matched donors, HCT recipients showed an accelerated progression towards this aged, oligoclonal haematopoiesis, with a mean 23 percentage point increase (range, −5 to +46 percentage points) in the proportion of haematopoiesis derived from expanded clones (Fig. 3a). Accordingly, global measures of clonal diversity were lower in recipients than their matched donors (Fig. 3b). To quantify this, we inferred a phylogenetic age by comparing each individual's tree to simulations of normal ageing haematopoiesis. This suggested that the accelerated loss of clonal diversity seen in HCT recipients would be equivalent, on average, to an additional 12.0 years (95% CI = 11.7–12.2 years) of ageing compared with donor siblings (Fig. 3c).

## Lineage biases of engrafted stem cells

HSPCs are pluripotent stem cells, responsible for long-term production of multiple differentiated cell types. To assess whether different HSPC clones showed biases in these lineage outputs, we performed deep targeted sequencing on purified populations of granulocytes, monocytes and B and T lymphocytes (mean target coverage, 1,720×; range, 751–3,485×; Supplementary Fig. 4). We developed a phylogeny-informed Bayesian model to infer posterior distributions for the clonal fraction of each somatic mutation in each cell type (Supplementary Fig. 5). We found that, in general, clonal fractions inferred directly from the colonies used for the phylogeny correlated well with total peripheral blood myeloid cells, suggesting that the colonies sampled accurately from the whole-body pool of active HSPCs (Extended Data Fig. 7a). Overall, 114 clones had expanded to >1% clonal fraction in at least one cell type, accounting for 0–87% of each mature compartment (Fig. 3d). Only 17% had known drivers.

We estimated the overall clonal diversity in each mature cell type using the Shannon diversity index (SDI). Clonal diversity decreased with age (−0.09 SDI per year; 95% CI = −0.12 to −0.06; $P$ = 0.0003), and was significantly lower in recipients than donors (−0.53 SDI; 95% CI = −0.81 to −0.25; $P$ = 0.0006; mixed-effects model; Fig. 3e,f). While the overall clonal diversity decreased in recipients, the trajectory of individual clones was more variable, with some clones expanding more in donors than recipients (Extended Data Fig. 7b,c).

There was marked clone-to-clone variability in contributions to different mature cell types, with 93% (106 out of 114) clones showing significant bias toward one or the other lineage (Extended Data Fig. 8a,b).

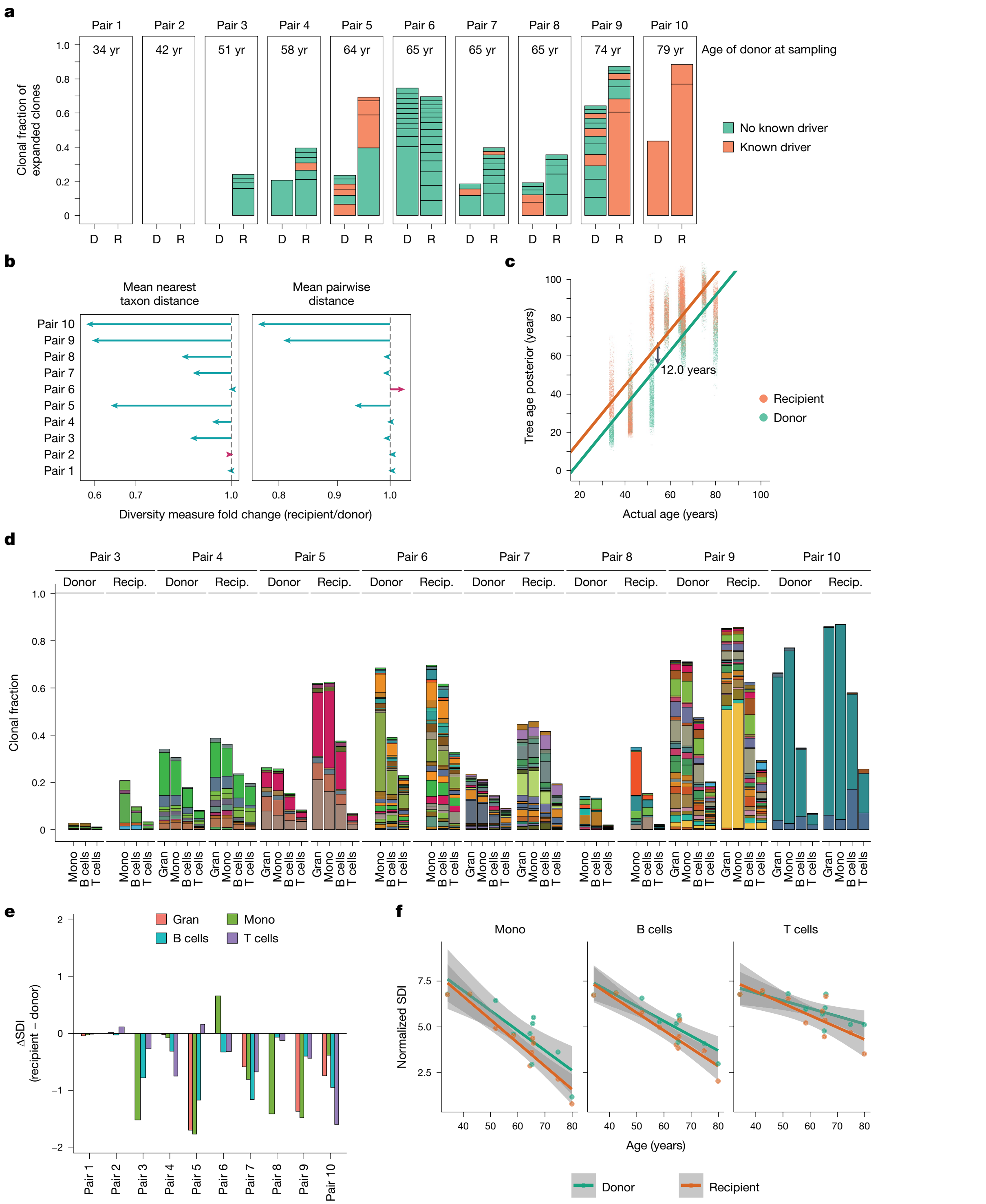


**Fig. 3 | Loss of clonal diversity in recipients of HCT. a**, The number and size of clades with a clonal fraction ≥2% of haematopoiesis. Plots are divided by pair, and by recipient (R) or donor (D). Clones are defined as a lineage originating from 100 mutations of molecular time (corresponding to the first few years after birth). Bars are coloured by whether the clone has a known driver (orange) or not (green). **b**, The fold change in phylogenetic diversity in recipients compared with their donors as measured by the mean pairwise distance (left) or mean nearest taxon distance (right). The line colour illustrates whether the recipient diversity is decreased (cyan) or increased (red). **c**, The posterior estimates of the phylogenetic age of recipients (orange) and donors (green) as estimated by ABC, compared with the true donor age at the time of sampling. The solid lines indicate the relationship between phylogenetic age and true age, split by donors (green) and recipients (orange), as estimated by linear mixed-effects regression. **d**, The size of clonal expansions in each mature cell compartment, as found in targeted sequencing. Includes clones contributing ≥1% clonal fraction in at least one compartment. Bar colours are arbitrary, but are consistent within pairs to enable comparisons between cell types and donors/recipients. Pairs 1 and 2 have no clones of ≥1% and are therefore not shown. **e**, The change in Shannon diversity index (SDI) (recipient SDI minus donor SDI) in each mature cell compartment. **f**, The relationship between SDI and donor age, divided by donors (green) and recipients (orange), and split by mature cell type. The solid lines indicate the line derived from the maximum likelihood of the linear relationship split by donor (green) and recipient (orange). The grey shaded areas show the 95% confidence interval of this relationship. The points show the individual estimates of SDI for the $n = 10$ donors and $n = 10$ recipients. Gran, granulocytes; mono, monocytes.

However, on average, compared with myeloid cells, expanded clones contributed less to B cells and much less to T cells, meaning that the aggregate cellular fraction attributable to observed clones was always lower in lymphoid than in myeloid populations (Fig. 3d).

Thus, T lymphocytes in recipients were donor derived, but drawn from a pool of stem cells that was broader than or only partially overlapped with the pool that generated mature myeloid cells. Individual memory T cells turn over infrequently and have long half-lives[32,33], with T cell memory lineages persisting for 8–15 years[34,35] or longer[36]—this means that the composition of T lymphocytes at the time of sampling would represent cells that differentiated from the HSC compartment as it existed a decade earlier in life. By contrast, the shorter half-life of myeloid progenitors means that differentiated myeloid cells would reflect a more contemporaneous HSC compartment. Simulations using lifespans of 8–15 years for T cell clones resulted in the overall T cell fraction attributable to expanded clones reaching around 60–80% of the levels observed in the myeloid fraction (Extended Data Fig. 8c). This explanation could therefore account for much, but not all, of the observed differences in cellular fraction between myeloid and T lymphoid cells.

We also assessed the clonal fraction of early embryonic mutations to infer the existence of large clonal expansions in lymphocytes not detected in the myeloid colonies. For younger pairs (pairs 1–5), embryonic mutations had similar clonal fractions across mature cell types between donor and recipient (Extended Data Fig. 8d). For one of the older pairs, pair 10, one clonal expansion barely contributed to donor T cells but comprised around 20% of T cells in the recipient. However, for the other older pairs (pairs 5–9), the clone fractions comparing donor and recipient for each cell type were generally more closely aligned with one another than with the other cell types. Thus, the clonal composition of lymphocytes in the recipient closely resembled that seen in the donor, without detectable transplant-related skewing towards a few dominant clones.

## Pruning versus growth selection

We tested whether a simple population bottleneck at the time of transplant coupled with the expected age-related selection for HSPCs[19] was sufficient to explain the reduced clonal diversity in the recipients compared with donors. With formal model testing, the closest-matching simulations from these models could not accurately recapitulate the distributions of branch points across observed phylogenetic trees, especially for the older donor–recipient pairs (three donors with posterior $P < 0.05$; a further three donors with $P = 0.05$–$0.06$; Extended Data Fig. 9). In particular, whereas simulated bottlenecks led to coalescences randomly distributed among clones (Supplementary Fig. 3), trees from the older donors often exhibited pronounced asymmetry, with branching enriched in one or a few clones (Extended Data Fig. 1). Thus, while a model of age-related selection plus transplant bottleneck was sufficient for younger donors, it could not adequately explain trees from the older donor–recipient pairs.

Two distinct patterns of branching were evident within individual clones in the recipient trees, indicative of two alternative modes of selection. In the first, exemplified by pair 3 (Fig. 4a), the coalescences underpinning clonal expansions occurred long before the time of HCT—consistent with this, these clones were detectable in the donor in the deep targeted sequencing data, albeit at much lower frequency than in the recipient, evidencing that their expansions did indeed begin before transplant (Extended Data Fig. 10). In fact, this pattern was evident across most of the older recipients but was not seen in population bottleneck simulations (Supplementary Fig. 3), where coalescences cluster at the time of the bottleneck. In the second pattern of branching, coalescences occurred in the recipient tree at the estimated time of transplant, when the bottleneck would have been most pronounced, as exemplified by pair 9 and pair 7 (Fig. 4b,c).

These two patterns imply two separate modes of selection, which we term pruning selection and growth selection, respectively. In the first, the cell divisions underpinning clonal expansion occurred in the donor, even though the expansion was considerably more evident in the recipient—this is analogous to a tree undergoing rigorous pruning everywhere except for one or two selected branches. Pruning selection would result from any cell-intrinsic factor that increased the likelihood of a clone successfully engrafting, such as more abundant mobilization from the donor into peripheral blood; increased collection from the bone marrow; better survival ex vivo; or more successful homing to the bone marrow in the recipient. In the second pattern, clustering of coalescences in selected clones occurred in the recipient at the time of transplant—this is analogous to a specific bough or two of a tree growing and branching more extensively than others. This growth selection would result from any cell-intrinsic factor that promotes the preferential proliferation of HSPCs from a given clone after engraftment in the recipient's bone marrow.

Models that included either pruning selection or growth selection generally improved fit to the observed data, although three donor–recipient pairs remained poorly explained (Extended Data Fig. 9d). It is possible that, for these donors, both forms of selection were operative in different clones (Fig. 4c,d) or that other factors contributed, although these factors would have had to exert clone-specific effects to generate the asymmetry we observed.

## Dynamics of driver mutations through HCT

Several studies have assessed the impact of pre-existing driver mutations in the donor on HCT outcomes[37–39]. While most driver clones engraft, their dynamics are unpredictable, with more than 50% expanding through HCT, but others remaining stable or decreasing in size. These studies have been unable to assess the long-term effect of HCT on clones, as they have only had donor samples from pre-HCT, and therefore no control comparison for the clone trajectory in the absence of HCT. With the benefit of matched donor and recipient samples, unbiased sampling across the entire haematopoietic system and deep targeted sequencing in multiple mature cell types, we assessed the dynamics of clones with known driver mutations.

There were 52 known driver mutations, defined as hotspot mutations in oncogenes and truncating or hotspot missense mutations in tumour suppressor genes (Supplementary Tables 1 and 2). Comparing recipient and donor monocytes, 14 clones expanded to higher fractions in the recipient and 5 to lower factions, but most (33 clones) had no significant difference (Fig. 5a). Only *CHEK2* mutations had a consistent effect, with all four mutations being at higher fractions in recipients than donors. Clonal trajectories in B and T cells were generally similar to monocytes (Extended Data Fig. 11). Most clones with two or more known drivers were significantly larger in recipients, while clones with a single driver were just as likely to be smaller as larger (Fig. 5b).

To investigate the potential roles of either negative selection or positive selection for unidentified drivers, we measured the ratio of non-synonymous to synonymous mutations (dN/dS ratio)[40]—here a value of around 1 implies balanced positive and negative selection (or neutrality); >1 implies predominance of positive selection; and <1 implies more negative than positive selection. The combined dN/dS ratio across all coding genes was 1.09 (95% CI = 1.06–1.13), similar to values observed in healthy individuals[19] (Extended Data Fig. 10d). This implies a net positive selection, with about 1 in 11 non-synonymous mutations being drivers[40]; thus, across a total of 7,809 such variants in our recipient colonies, at least 775 (95% CI = 454–1,081) are under positive selection. Only 70 non-synonymous mutations occurred in known blood cancer and clonal haematopoiesis genes, suggesting that there are many driver genes remaining to be discovered that could confer a transplant-specific selective advantage.

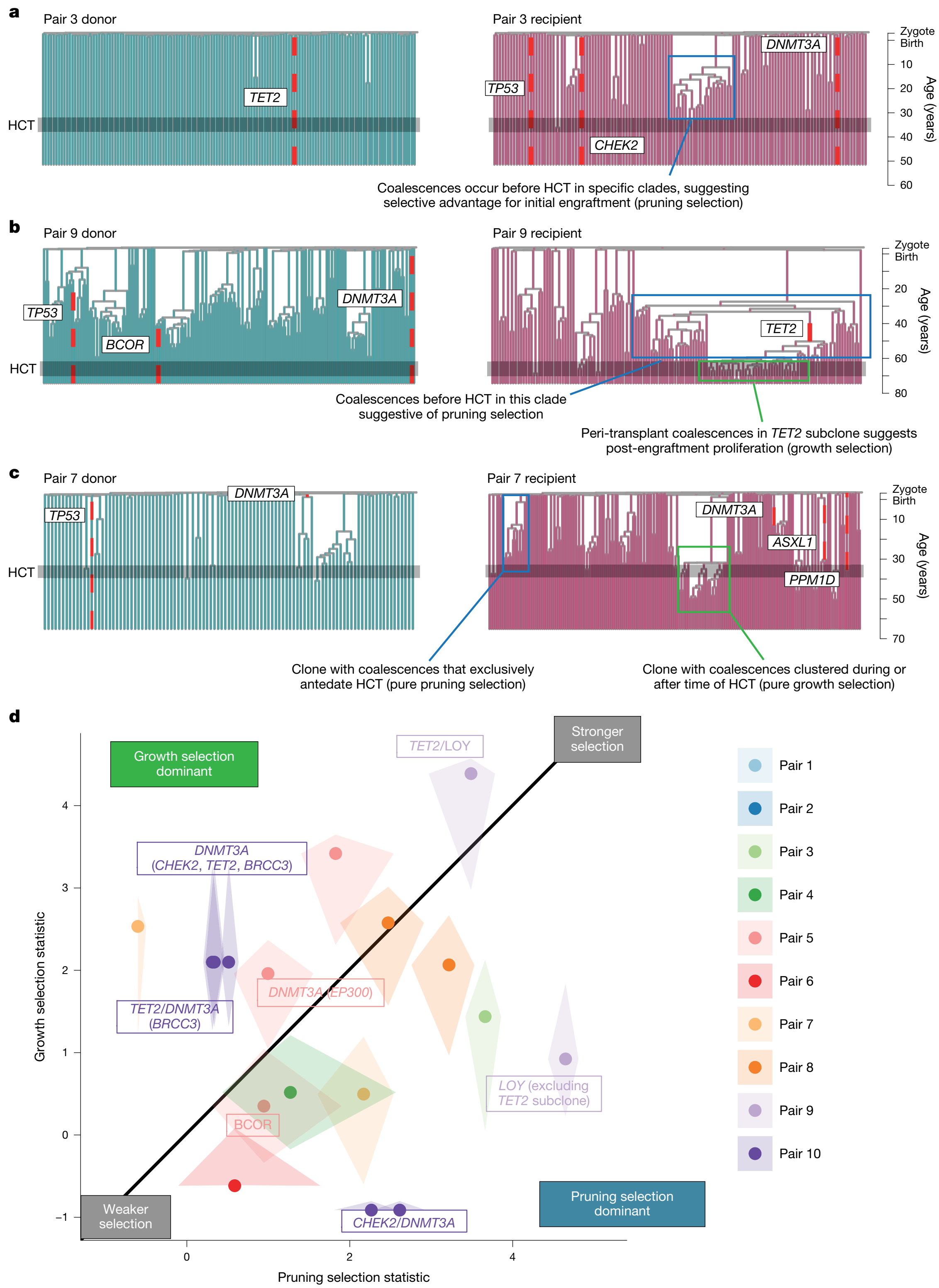


**Fig. 4 | HCT-specific selection contributes to decreased recipient diversity.** **a**, Recipient and donor phylogenies for pair 3, for which there is a large clonal expansion within the recipient that is not evident in the donor. The increased coalescences are from before the time of HCT, consistent with an initial engraftment advantage for this clone (pruning selection). **b**, Recipient and donor phylogenies for pair 9, for which there is a large clonal expansion within the recipient that is not expanded in the donor. The clone component with loss of Y, but no mutation in *TET2*, has increased coalescences from before the time of HCT, consistent with an engraftment advantage. The component with both loss of Y and a *TET2* mutation has both increased coalescences before and at the time of HCT, consistent with an engraftment and post-engraftment proliferation advantage (pruning and growth selection). **c**, Recipient and donor phylogenies for pair 7, for which there are large clonal expansions within the recipient that are not evident in the donor that show either pure pruning selection or pure growth selection. **d**, The pruning and growth selection statistics for each clone that has preferentially expanded in the recipient, illustrating the differences between clones. Shaded areas are estimates of the 95% confidence intervals of these values estimated by node bootstrapping. Clones with driver mutations are labelled with the mutated gene.

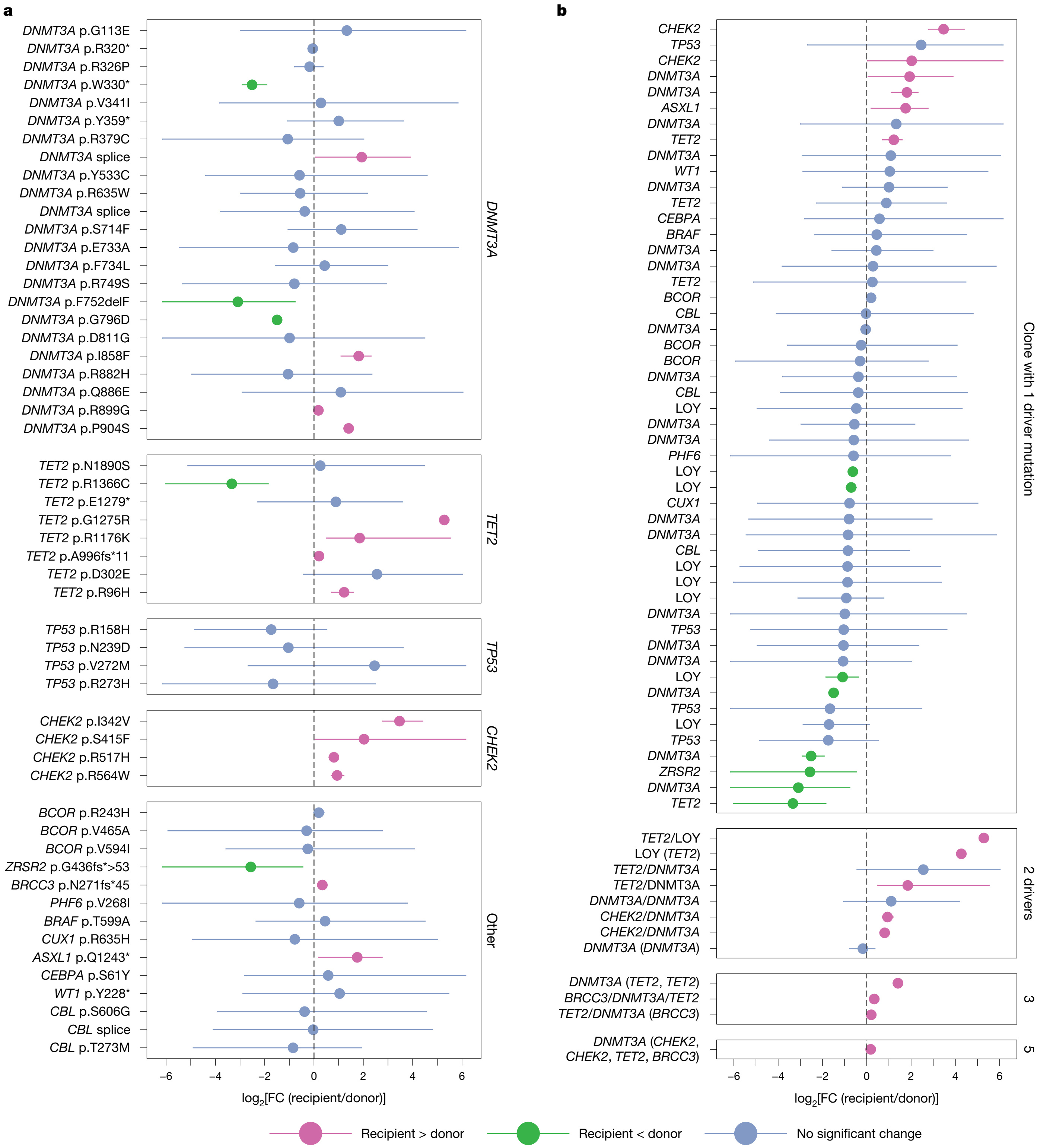


**Fig. 5 | Clones with driver mutations can have differing dynamics in donors and recipients. a**, The $log_2$-transformed fold change in putative driver variant allele fractions (VAFs) in recipients compared with donors. The circles denote point estimates, and are coloured by whether fractions are lower (orange) or higher (green) in recipients, or show no significant difference (blue). The error bars show the 95% confidence interval of the $log_2$-transformed fold change. Variants are grouped by the affected gene. **b**, The fold change in putative driver variant VAFs as in **a**, but now including loss of Y events, and showing the clonal hierarchy. Clones are grouped by the total number of driver variants within the clone. Where mutations occur on the background of another driver mutation, genes are shown in the format GENE1/GENE2, indicating that all cells in the clone have driver mutations in both genes. Where the mutant clone contains a major subclone with an additional driver mutation, these are shown in the format GENE1 (GENE2), indicating that some, but not all cells in the clone have driver mutations in both genes. LOY, loss of the Y chromosome.

## Discussion

Immune reconstitution after allogeneic stem cell transplantation determines many clinical outcomes—it restores adaptive immunity to pathogens; it can cause graft-versus-host disease; and it often mediates a graft-versus-leukaemia reaction, the original immunotherapy. While myeloid recovery after HCT usually occurs within 2–3 months, lymphoid recovery takes years, delayed by immunosuppressive agents and the slower trajectory to full diversification of the adaptive immune repertoire. We find that transplanted stem cells typically show long-term multipotency, with expanded clones contributing not only to myeloid populations but also B and T lymphocyte production after engraftment. We note that, by studying long-term survivors, we have biased against transplants with poor outcomes; it would be fascinating to study whether clonal dynamics of myeloid and immune reconstitution are different in patients with poor graft function or graft-versus-host disease.

Of the hundreds of millions of $CD34^+$ cells infused into the recipient, only a few thousand to tens of thousands will still be contributing to haematopoiesis a decade or more later. Like the hero of a picaresque novel, a transplanted stem cell must navigate serial perils in this quest—it must mobilize from its native niche in the donor marrow, withstand direct bone marrow collection or peripheral blood apheresis, survive hours to days ex vivo, home to a new niche extensively reconditioned with chemotherapy and then proliferate to enable multilineage blood production. The observation of two distinct modes of selection in the phylogenetic trees argues that a stem cell's fitness is not a constant, all-encompassing property. Rather, its advantage may only manifest at specific points along this journey. Notably, the lower clonal diversity of haematopoiesis that we observed in older donors arises from the preferential growth of stem cells that are especially well-adapted to survive these particular perils, better adapted than wild-type stem cells—that is, the deterioration with age does not operate through a general decline in fitness of transplanted HSPCs but, rather, the acquisition of increased fitness in a small subset of HSPCs. The drawback is that these clones may then have properties that disadvantage the recipient in the long term, such as lineage bias, poor responsiveness to haematopoietic stress or insufficient immune diversification, thereby explaining the poorer outcomes of HCT from older individuals[6,7]. By identifying which genes carry somatic mutations or epigenetic changes enriched at different stages of the transplant procedure, it may be feasible to identify pathways that promote successful engraftment of clonally diverse transplanted stem cells.

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

# Methods

## Study population

Ten donor–recipient pairs were selected from the original 45 HCT recipients and HLA-matched sibling donors who were enrolled in the original study[3]. All donors and recipients gave written informed consent, and the original study and subsequent amendment were approved by the local ethics committee (KEK-ZH, 2015-0053 and 2019-02290; Kantonale Ethikkommission-Zurich) in accordance with the declaration of Helsinki. The priority for selecting patients for our analysis was the potential availability of follow-up material, as some patients had gone on to develop haematological malignancies, been found to have lost donor chimerism or been lost to follow-up since the initial study. Beyond that, we aimed to include a variety of sibling pair ages, stem cell source and conditioning type. We also had awareness of some CHIP clones detected in recipients and/or donors in the original study, and hence also wanted a mixture of scenarios.

## Cell isolation

Granulocytes were isolated from 10 ml of EDTA anticoagulated peripheral blood using the EasySep Direct Neutrophil Isolation Kit (Stem Cell Technologies) according to the manufacturer's instructions. $CD34^+$ HSPCs were isolated from 20 ml of EDTA anticoagulated peripheral blood using the human CD34 MicroBead Kit (Miltenyi Biotec) according to the manufacturer's recommendations. B cells, T cells and monocytes were flow-sorted from $CD34^-$ cell fractions using the FACSAria III flow cytometer (BD Biosciences). Antibodies used were PE/Cyanine7 anti-human CD14 (BioLegend, 301814), APC anti-human CD3 (BioLegend, 317318) and FITC anti-human CD19 (BioLegend, 363008), diluted and used according to manufacturer instructions, with the validation of antibodies performed by the manufacturer.

## Clonal expansion

$CD34^+$ HSPCs were plated in 9 ml cytokine-supplemented methylcellulose medium (Stem Cell Technologies) as described previously[41]. After 14 days of culture at 37 °C and 5% $CO_2$ single colony-forming units were picked and were each resuspended and processed in 20 µl QuickExtract DNA Extraction Solution (Lucigen).

## DNA extraction

DNA from granulocytes, monocytes, B cells and T cells was isolated using the QIAamp DNA Mini Kit according to the manufacturer's recommendations.

## Library preparation and WGS

A target of 1–6 ng of DNA from each colony underwent low-input library preparation as previously described using 12 cycles of PCR amplification[42]. Paired-end sequencing reads (150 bp) were generated using the Illumina NovaSeq 6000 platform resulting in around 8–15× coverage per colony (Supplementary Fig. 1a). BWA-MEM was used to align sequences to the human reference genome (NCBI build37).

## Mutation calling in clonal WGS data

SNVs and indels were initially called against a synthetic unmatched reference genome using the in-house pipelines CaVEMan (cgpCaVEMan) and Pindel (cgpPindel)[43,44]. For all mutations passing quality filters in at least one sample, in-house software (cgpVAF; https://github.com/cancerit/vafCorrect) was used to produce matrices of variant and normal reads at each site for all HSPC colonies from that donor–recipient pair.

Multiple post hoc filtering steps were then applied to remove germline mutations, recurrent library preparation and sequencing artefacts, and probable in vitro mutations, as detailed below:

(1) A custom filter to remove artefacts specifically associated with the low-input library prep process (https://github.com/MathijsSanders/SangerLCMFiltering). This is predominantly targeted at artefacts introduced by cruciform DNA structures.
(2) A binomial filter was applied to aggregated counts of normal and variant reads across all samples. Sites with aggregated count distributions consistent with germline single-nucleotide polymorphisms were filtered.
(3) A beta-binomial filter was applied to retain only mutations of which the count distributions across samples came from an overdispersed beta-binomial distribution consistent with an acquired somatic mutation.
(4) Mutations called at sites with abnormally high or low mean coverage were considered unreliable/possible mapping artefacts and were filtered.
(5) For each mutation call, normal and variant reads were aggregated from positive samples (≥2 variant reads). Sites with counts inconsistent with a true somatic mutation were filtered.
(6) The remaining mutations were retained only if there was at least one sample that met all minimum thresholds for variant read count (≥3 for autosomes, ≥2 for XY chromosomes), total depth (≥6 for autosomes, ≥4 for XY chromosomes) and a VAF > 0.2 for autosomal mutations or >0.4 for XY mutations in males.

Copy-number changes were called using ASCAT-NGS (ascatNgs)[45] and SVs with GRIDSS[46]. Protein-coding consequences were annotated using VAGrENT[47] and these were used for inferring the presence of positive selection using dNdScv[40].

## Genotyping each sample for somatic mutations

Each sample was genotyped for each somatic mutation in the filtered mutation set. For each mutation, samples with a VAF > 0.15 and at least 2 variant reads were considered positive; samples with no variant reads and a depth of at least 6 were considered negative; samples not meeting either of these criteria were considered uninformative.

## Phylogeny inference

Phylogenies were inferred using the maximum parsimony algorithm MPBoot[48]. This efficient algorithm has been shown to be effective for the robust genotypes built with WGS of clonal samples, as is performed here, and is comparable to other maximum-likelihood-based algorithms[18,19]. To test this, we performed phylogeny inference for all trees with the maximum-likelihood algorithm IQtree (http://www.iqtree.org/) and compared the resulting phylogenies to those from MPBoot. These showed extremely similar structures in all cases as shown by high Robinson–Foulds (range, 0.955–0.933) and Quartet similarity scores (range, 0.903–1.000). In almost all cases, the differences were in the orientation of early developmental splits that would have no bearing on the downstream analysis (Supplementary Fig. 6).

Many different algorithms have been developed to reconstruct phylogenetic trees based on DNA sequences. These character-based algorithms rely on different approaches: maximum parsimony, maximum likelihood or Bayesian inference[49]. Maximum parsimony-based algorithms seek to produce a phylogeny that requires the fewest discrete changes on the tree. As the number of nucleotide changes is minimized, this approach implicitly assumes that mutations are likely to occur only once. Thus, maximum parsimony may produce erroneous phylogenies when there is a high likelihood of recurrent or reversal mutations, such as with long divergence times or high mutation rates, neither of which generally apply to mutations in normal somatic cells. Phylogenetic tree algorithms relying on maximum likelihood or Bayesian inference are model-based, in that they require a specific notion of the parameters governing genetic sequence evolution to calculate either distances or likelihoods. Often, this involves a general time-reversible model of sequence evolution[50]. All these approaches have been widely applied to the reconstruction of phylogenetic trees between species or individuals[49]. However, the task of constructing a phylogeny of somatic

cells derived from a single individual is fundamentally different from reconstructing species trees in three ways:

(1) Precise knowledge of the ancestral state: in contrast to the unknown ancestral genetic state in alignments of sequences from multiple species, the ancestral DNA sequence at the root of the tree (namely the zygote) can readily be inferred from the data. As all cells in the body are derived from the fertilized egg, any post-zygotic mutation will be present in only a subset of the leaves of the tree. Thus, the genetic sequence at the root of the tree is defined by the absence of all of these mutations. This simple observation effectively roots the phylogeny.

(2) Unequal rates of somatic mutation versus reversion: to accommodate the uncertainty in the ancestral state and the direction of nucleotide substitutions, model-based phylogeny reconstruction has relied on a time-reversible model of nucleotide changes[50]. In principle, this states that the probability of a certain substitution (for example, C>T) is equal to its inverse (T>C). In somatic mutagenesis, as the direction of change is known, assuming general reversibility of mutational probabilities fails to acknowledge the genuine discrepancies in the likelihood of certain (trinucleotide) substitutions. For example, a C>T mutation in a CpG context is much more probable than a T>C at TpG due to the specific mutational processes acting on the genome—in this case, spontaneous deamination of methylated cytosine (commonly referred to as SBS1).

(3) Low somatic mutation rates in a human lifespan—when accounting for the size of the human genome, the number of mutations that are informative for purposes of phylogeny reconstruction, namely SNVs shared between two or more samples, is generally low compared to the settings of phylogenies of species or organisms. This means that the probabilities of independent, recurrent mutations at the same site or reversals of those nucleotide changes (back mutations) are small and have negligible effects on the accuracy of phylogenetic reconstruction. Thus, a mutation shared between multiple samples can generally be assumed to represent a single event in an ancestral cell that has been retained in all its progeny—the underlying principle of maximum parsimony.

Thus, on both empirical metrics and theoretical grounds, maximum parsimony methods perform as accurately as model-based methods for reconstructing phylogenies of somatic cells, and require fewer additional assumptions.

### Exclusion of non-clonal samples

Haematopoietic colonies embedded within methylcellulose may grow into one another, or derive from more than one founder cell, resulting in colonies that are not single-cell derived. Such samples may interfere with phylogeny building and have lower numbers of called mutations, and were therefore excluded. Detection was done in two steps. The first was based on the principle that somatic mutations from clonal samples should have a peak VAF density of around 0.5. Thus, after exclusion of germline mutations and recurrent artefacts using the exact binomial and beta-binomial filtering steps, the VAF distributions of positive mutations in a sample were assessed. Samples with a maximum VAF distribution density of <0.4 (corresponding to a sample purity of <80%) were excluded. The second step was performed following a first iteration of phylogeny building using samples passing the first step. Each sample was tested against the phylogeny to see if the mutation VAFs across the tree were as expected for a clonal sample. A clonal sample should have either branches that are positive (mutation VAFs, ~0.5) or negative (mutation VAFs, ~0). Thus, for each branch in each sample, the variant and total read counts were combined across all branch mutations. These counts were then tested for how likely they were to come from either (1) at least that expected for a heterozygous somatic mutation distribution, with some contamination allowed (one-sided exact binomial test, alternative hypothesis = less than probability, probability = 0.425) or (2) no more than that expected for absent mutations, with some false positives allowed (one-sided exact binomial test, alternative hypothesis = greater than probability, probability = 0.05). If the samples had any branches with read counts that were highly inconsistent with both tests (maximum $q$-value < 0.05, Bonferroni correction) or had three or more branches that were minorly inconsistent with both tests (maximum $P$ value of 0.05, no multiple-hypothesis testing correction) the sample was considered to be non-clonal and excluded. A second iteration of phylogeny inference was then performed without the non-clonal samples. These steps have a degree of tolerance of minimally contaminated samples, and samples with >80–85% purity will generally be retained. However, even this lower level of contamination will have an impact on the sensitivity of mutation calling and sample purity was therefore taken into account for mutation burden correction.

### Recognition of different germline background for samples

Initial phylogeny building was done using all samples with a maximum VAF distribution density of >0.4. In three cases (pairs 3, 4 and 9), this initial phylogeny revealed an outlier clade with an apparent extremely high mutation burden of >30,000. The outlier clades contained only colonies grown from recipient samples, which raised the possibility that these may represent recipient haematopoiesis. For pair 3, the samples within the outlier clade were in fact identified as deriving from pair 10, and therefore represented interindividual contamination. This was clear, as 80% of the mutations in this clade were germline mutations from pair 10, and it also included the *DNMT3A* p.R899G and *TET2* p.A996fs*11 mutations. For pairs 4 and 9, this was not the case. There were no known pathogenic variants in the outlier clade. Feasibly, the samples may derive from residual recipient-derived haematopoiesis, or from contamination from another individual not in the study. As the donors are siblings, recipients will share around half the same germline variants of the donor. Accordingly, if the outlier clade were from residual recipient chimerism, the branch length of the outlier clade should be half the number of the ~30,000 germline mutations identified in the donors, that is, 15,000 mutations. However, in all cases, the outlier clade contained around 30,000 mutations, consistent with contamination from an unrelated individual rather than residual recipient haematopoiesis. In the two individuals where there was >1 sample within the outlier clade, these were from adjacent wells of the 96-well plate into which colonies were picked, making it likely that in fact the separate samples derived from the same original founder cell, that presumably grew into a large branching colony structure that was picked multiple times. Mutation filtering and phylogeny building was rerun excluding contaminating samples.

### Removal of sample duplicates

Some haematopoietic colonies grown in methylcellulose have an irregular branching appearance and are misinterpreted as multiple separate colonies, resulting in several samples being inadvertently picked from the same colony. Such samples appear highly related on the phylogenetic tree, with only a few private mutations, representing predominantly in vitro acquired mutations. Recognition of these duplicates is aided by the fact that (1) in many cases, duplicates are picked into adjacent/nearby wells, as colony picking is performed systematically around the well; and (2) in most biological scenarios, such highly related sample pairs are extremely rare due to the larger short-term HSC/HSPC pool[19]. Thus, pairs of samples with fewer than 30 private mutations, and close positions on the 96-well plate were assumed to be duplicates of the same colony, and one sample was removed.

### CNAs

CNAs were called from WGS data using ASCAT[45,46]. A good-quality matched sample from the same pair was used as a ‘normal reference’

after manual inspection of raw copy-number plots to exclude abnormalities. Copy-number profiles were manually reviewed, and alterations that were clearly distinguishable from background noise were tabulated.

### SV calling

SVs were called with GRIDSS[46] (v.2.9.4) with the default settings. SVs larger than 1 kb in size with QUAL ≥ 250 were included. For SVs smaller than 30 kb, only SVs with QUAL ≥ 300 were included. Furthermore, SVs that had assemblies from both sides of the breakpoint were considered only if they were supported by at least four discordant and two split reads. SVs with imprecise breakends (that is, the distance between the start and end positions > 10 bp) were filtered out. We further filtered out SVs for which the s.d. of the alignment positions at either ends of the discordant read pairs was smaller than five. Filtered SVs were rescued if the same SVs passing the criteria were found in the other samples. To remove potential germline SVs and artefacts, we generated the panel of normal by adding in-house normal samples ($n = 350$) to the GRIDSS panel of normal. SVs found in at least three different samples in the panel of normal were removed. Variants were confirmed by visual inspection and by checking whether they fit the distribution expected based on the SNV-derived phylogenetic tree.

### Mutational signature extraction

Mutational signatures were extracted de novo using a hierarchical Dirichlet process[51] as implemented in R package HDP (https://github.com/nicolaroberts/hdp). These reflect the signatures of underlying mutational processes that have been active in the HSPC colonies. Each branch on the phylogeny was treated as an independent sample, and counts of mutations at each trinucleotide context were calculated. Branches with <50 mutations were excluded as, below this threshold, random sampling noise in the mutation proportions becomes problematic.

Plots of signature contributions in each sample in Extended Data Fig. 3 represent the means of signature contributions of individual branches included within the sample (weighted by the branch length), with final values then scaled by the sample total mutation burden to reflect absolute signature contributions. Note that branches with <50 mutations—primarily early embryonic branches—are not included in this estimate as they are excluded from the signature extraction step. This means that processes primarily operative in embryogenesis are under-represented in these estimates.

### Correction of mutation burden

The number of somatic mutations called in any given sample depends not only on the number of mutations present, but also on the sequencing coverage and on the colony purity. For each individual, reference sets of germline polymorphisms (separate sets for SNVs and indels) were defined ($n > 30{,}000$ SNVs in all cases). These were mutations that had been called in many samples (as mutation calling was performed against an unmatched synthetic normal), and for which aggregated variant/reference mutation counts across samples from an individual were consistent with being present in the germline. For each sample, the proportion of germline SNVs called by CaVEMan and passing the low-input filter was considered the 'germline SNV sensitivity', and the proportion of germline indels called by Pindel was the 'germline indel sensitivity'. For pure clonal samples, the sensitivity for germline variants should be the same as for somatic variants. Therefore, for samples with a peak VAF > 0.48 (corresponding to a purity of >96%), this germline sensitivity was also considered the 'somatic variant sensitivity' and was used to correct the number of somatic variants. However, for less pure samples (purity, 80–96%), the sensitivity for somatic variants will be lower than for germline variants as the former will not be present in all cells of the sample. Thus, an additional 'clonality correction' step was applied. The expected number of variant reads sequenced for a heterozygous somatic mutation in a non-clonal sample will be $n_v \sim \text{binomial}(N,p)$ where $N$ is the sequencing coverage at the mutation position, and $p$ is the sample peak VAF (rather than $p = 0.5$ as is the case for a pure clonal sample). The likelihood of the mutation being called given $n_v$ variant reads and $N$ total reads was taken from a reference sensitivity matrix. This matrix was defined from the germline polymorphism sensitivity data across 20 samples, where for all combinations of $n_v$ and $N$, the proportion of mutations called in each sample's final mutation set was assessed. The sequencing coverage distribution across putative somatic mutations was considered the same as that across the germline polymorphism set. Thus, for each value of $N$ (the depths across all germline polymorphisms in that sample), a simulated number of variant reads $n_v$ was taken as a random binomial draw as described above, and whether this resulted in a successful mutation call taken as a random draw based on the probability defined in the sensitivity matrix. The total proportion of simulated somatic mutations successfully called was defined as the 'somatic variant sensitivity' for that sample.

The somatic variant sensitivities were then used to correct branch lengths of the phylogeny in the following manager. For private branches, the SNV component of branch lengths was scaled according to

$$n_{\mathrm{cSNV}} = \frac{n_{\mathrm{SNV}}}{p_i}.$$

Where $n_{\mathrm{cSNV}}$ is the corrected number of SNVs in sample i, $n_{\mathrm{SNV}}$ is the uncorrected number of SNVs called in sample i and $p_i$ is the somatic variant sensitivity in sample $i$.

For shared branches, it was assumed (1) that the regions of low sensitivity were independent between samples, (2) if a somatic mutation was called in at least one sample within the clade, it would be 'rescued' for other samples in the clade and correctly placed. Shared branches were therefore scaled according to

$$n_{\mathrm{cSNV}} = \frac{n_{\mathrm{SNV}}}{1 - \prod_i (1 - p_i)}.$$

Where the product is taken for $1 - p_i$ for each sample $i$ within the clade. Neither assumption is entirely true. First, areas of low coverage are non-random and some genomic regions are likely to have below average coverage in multiple samples. Second, while many mutations will indeed be rescued in subsequent samples once they have been called in a first sample—because the treemut algorithm for mutation assignment goes back to the original read counts and therefore even a single variant read in a subsequent sample is likely to lead to the mutation being assigned correctly to a shared branch—this will not always be the case. Sometimes samples with a very low depth at a given site will have 0 variant reads by chance. In such cases, a mutation may be incorrectly placed. Both factors may result in under-correction of shared branches, but it is a reasonable approximation. SNV burdens corrected by this approach were then taken as the sum of corrected ancestral branch lengths for each sample, going back to the root.

### Custom DNA capture panel design and targeted sequencing

Three separate custom panels were designed according to the manufacturer's instructions (SureSelect$^{XT}$ Custom DNA Target Enrichment Probes, Agilent) for (1) pairs 6, 7, 9 and 10, (2) pairs 2, 3 and 8 and (3) pairs 1, 4 and 5. Custom panels were designed for groups of pairs such that sequencing error rates could be estimated from individuals without the mutation, although the specific grouping was for logistic reasons. Panel design proceeded similarly for each panel. All SNVs on shared branches of the phylogeny were covered if they met the moderate stringency repeat masking applied within the SureDesign platform (around 60% of loci). For short shared branches with no covered mutation loci after moderate stringency repeat masking, loci included after low stringency repeat-masking were accepted. A total of 10,000 SNVs per transplant pair from across private branches was selected based on more stringent

criteria to maximize capture efficiency. They were considered only if (1) they met more stringent mutation filtering thresholds than those used for mutation calling (VAF > 0.35 for autosomal mutations, or VAF > 0.8 for XY mutations in males; beta-binomial rho value > 0.3); (2) mutation loci were included after the most stringent repeat masking; and (3) minimal capture bait boosting was required to compensate for high DNA GC content. After this, mutations were ranked according to sequencing error rates, and those with lowest error rates selected first. Error rates were taken from the site-specific error rate information used for the Shearwater mutation-calling algorithm[52]. Typically, 5–10% of private SNVs were covered. Indels were included only if within driver-gene-coding sequences. Moreover, ten putative driver genes from a WGS study of clonal haematopoiesis[53] were covered in their entirety (*DNMT3A*, *TET2*, *ASXL1*, *PPM1D*, *ATM*, *MTA2*, *ZNF318*, *PRKCG*, *SRSF2* and *KPNA7*).

Four separate aliquots of 50 ng of DNA from each bulk sorted cell type (granulocytes, monocytes, B cells and T cells) from each individual underwent low-input library preparation using nine cycles of PCR amplification. Paired-end sequencing reads (100 bp) were generated, hybridized to the appropriate custom bait capture panel, multiplexed on flow cells and then sequenced using the NovaSeq 6000 platform. In several cases, there was insufficient DNA to permit four aliquots of 50 ng. In such cases, decreased input DNA down to 25 ng and/or fewer aliquots were used. If <20 ng total DNA was available, aliquots of 5 ng were used with 12 cycles of PCR amplification during library preparation.

### Driver mutation annotation

A broad 122-gene list of driver genes associated with haematological malignancy and/or clonal haematopoiesis was compiled from the union of (1) a 54-gene Illumina myeloid panel (TruSight myeloid sequencing panel; https://www.illumina.com/products/by-type/clinical-research-products/trusight-myeloid.html); (2) the 92-gene list used in a study of chemotherapy-associated clonal haematopoiesis[29,54]; (3) a 32-gene list of genes identified recently as under positive selection within the UK Biobank whole-exome blood sequencing data (Supplementary Table 1). We then looked for missense, truncating or splice variants in these genes, yielding 174 such variants (Supplementary Table 2). These were then manually curated down to 70 variants considered to be potentially pathogenic, with the remainder classified as variants of unknown significance. This was done using the COSMIC database of somatic mutations (https://cancer.sanger.ac.uk/cosmic), the broader literature and, in some cases, variant effect prediction tools such as SIFT and PolyPhen.

### Gibbs sampler for inferring true VAF of mutations from deep sequencing data

The data comprise deep targeted sequencing of known somatic mutations from a given sample. Control samples (typically from another patient, where the mutations are absent) are also sequenced to enable estimation of sequencing error rates at each mutation position. Clonal relationships among the somatic mutations arise from a phylogenetic tree—it is assumed that this phylogenetic tree is known (and therefore considered fixed in the algorithm that follows).

We want to estimate a posterior distribution for the true VAF of every mutation in the bait set. The structure of the phylogenetic tree provides considerable constraint on the solution space of VAFs for clonally related mutations—for example, a mutation on a descendant branch cannot have a higher VAF than a mutation on a direct ancestral branch. Moreover, for a given node on the tree, comprising an ancestral branch and two or more descendant branches, the sum of the maximum VAFs for mutations on the descendant branches must be less than the minimum VAF of mutations on the ancestral branch.

The blocked Gibbs sampler infers the posterior VAFs of each mutation subject to the constraints imposed by the phylogenetic tree. Essentially, we use data augmentation to assign a maximum and minimum VAF for each branch in the tree ($\lambda_j$ and $\kappa_j$ in the notation below)—the VAFs for each mutation on that branch must fall within that range.

Let $\rho_i \equiv$ VAF of mutation $i$ in the sample, the variable of interest; $\varepsilon_i \equiv$ error rate of mutation $i$ in the control samples; $\pi_i \equiv$ error rate of mutation $i$ in the control samples; $Y_i \equiv$ number of variant-specific reads reporting mutation $i$ in the sample; $N_i \equiv$ total coverage of mutation $i$ in the sample (read depth); $B_j \equiv$ branch $j$ from the phylogenetic tree, $T$, comprising a set of mutations assigned to it; $\lambda_j \equiv$ maximum allowable VAF in the sample for mutations on $B_j$; $\kappa_j \equiv$ minimum allowable VAF in the sample for mutations on $B_j$.

**Block 1: updating $\rho_i$ for all mutations.** Proceeding branch by branch, the VAF of each mutation on a given branch, $B_j$, must fall within the range $[\kappa_j, \lambda_j]$. We assume an uninformative prior—that is, $\rho_i \sim U(\kappa_j, \lambda_j)$.

Reads reporting the variant allele can arise either from a read that correctly reports a mutant DNA molecule or a sequencing error on a read from a wild-type DNA molecule. This means that the expected proportion of reads reporting the variant allele is calculated as

$$\pi_i = \rho_i + \varepsilon_i - 2\rho_i\varepsilon_i$$

We assume a binomial distribution of the variant read counts given the VAF—that is, $Y_i \sim \mathrm{Bin}(\pi_i, N_i)$.

We use a Metropolis-Hastings approach to update the estimates for $\rho_i$. A new, proposed VAF for iteration $k$ is drawn from a truncated Beta distribution

$$\dot{\rho}_i^{(k)} \sim \text{Beta_truncated}\left(\frac{\rho_i^{(k-1)}\sigma}{\left(1-\rho_i^{(k-1)}\right)}, \sigma; \kappa_j^{(k-1)}, \lambda_j^{(k-1)}\right)$$

where $\sigma$ is a user-defined scale factor to be chosen to optimize the acceptance rate of the Metropolis–Hastings update. The acceptance ratio is then calculated from the distribution functions of the binomial under the current and proposed values for the VAF in the usual way, and the new value is either accepted or rejected.

**Block 2: updating $\lambda_j$ and $\kappa_j$ for all branches.** To update the maximum and minimum VAFs for each branch, we proceed node by node across the tree (where a node represents coalescences in the tree, comprising one inbound, ancestral branch and two or more outbound, descendant branches). As above, the sum of the maximum VAFs for mutations on the outbound branches must be less than the minimum VAF of mutations on the inbound branch. This means that there is an amount of 'unallocated VAF' that represents the difference between these values:

$$\mathrm{VAF}_{\mathrm{Unallocated}} = \min\left\{\rho_i^{(k)}\, on\, B_{\mathrm{Inbound}}\right\} - \sum_{x \epsilon B_{\mathrm{Outbound}}} \max\left\{\rho_i^{(k)}\, on\, x\right\}$$

We partition this unallocated VAF among the inbound and outbound branches using draws from a uniform distribution. Essentially, if there are $n$ branches coming in or leaving the current node, we draw $n$ values from $U(0, \mathrm{VAF}_{\mathrm{Unallocated}})$, sort them and take adjacent differences: $u_{(1)} - 0$, $u_{(2)} - u_{(1)}$, $\cdots$, $\mathrm{VAF}_{\mathrm{Unallocated}} - u_{(n)}$. These are then allocated to the branches:

$$\kappa_{\mathrm{Inbound}}^{(k)} = \min\left\{\rho_i^{(k)}\, on\, B_{\mathrm{Inbound}}\right\} - (u_{(1)} - 0)$$

$$\lambda_{\mathrm{Outbound}}^{(k)} = \max\left\{\rho_i^{(k)}\, on\, B_{\mathrm{Outbound,c}}\right\} + (u_{(c)} - u_{(c-1)})$$

**Implementation.** We doubled the total read depth, $N_i$, for mutations on the sex chromosome in males. We used a scale parameter of $\sigma = 50$. The root node was assigned a fixed VAF of 0.5 and terminal

nodes a fixed VAF of $10^{-10}$. The Gibbs sampler was run for 20,000 iterations, with 10,000 discarded as burn-in and thinning to every 100 iterations.

### Defining posterior distribution of post-developmental clone fractions

The output of the Gibbs sampler is the posterior distribution of the VAF of each mutation covered by the custom hybridization panel. This was converted into clonal fractions of post-development clones. First, mutation VAFs were multiplied by two to give clonal fractions (assuming heterozygosity). The tree was then cut at a height of 100 mutations of molecular time to define when clones were considered to originate. While this is somewhat empirical, any molecular timepoint soon after development (which ends ~50–60 mutations) would yield similar results. For each branch traversing the defined clone cut-off point, the position of the cut-off along the branch was calculated, for example, if the branch goes from a height of 50 mutations to 150 mutations, a molecular time of 100 mutations would be halfway along the branch. Depending on the number of mutations covered from that branch, the position along that branch best reflecting the molecular time cut-off was calculated, for example, in the above example, if 60 out of the 100 mutations on the branch were included in the custom panel, the posterior clonal fraction of the 30th ranked mutation (ordered by decreasing median posterior clonal fraction) best approximates that of a clone originating at 100 mutations of molecular time. Where point estimates are displayed, the median posterior value is used.

### Measures of clonal diversity

Clonal diversity was assessed (1) from the individual phylogenetic tree structures and (2) from the clonal fractions in the targeted sequencing results of mature cell types.

**Phylogenetic diversity.** We first calculated the mean pairwise distance[55] by taking the mean of the distance matrix obtained using the cophenetic.phylo function from the R package ape. This is the mean phylogenetic distance (that is, the sum of branch lengths of the shortest path between samples) across all sample pairs in the phylogeny. We next calculated the mean nearest taxon distance[55], again starting with the distance matrix from the cophenetic.phylo function, but this time taking the minimum non-zero value from each row, and calculating the mean of these values. This represents the mean of the phylogenetic distance to the nearest sample, across all samples. For both measures, the ultrametric version of the phylogenies was used.

**SDI analysis.** The SDI ($H$) is defined as:

$$H = -\sum_{i=1}^{k} p_i \log(p_i)$$

where $k$ is the total number of groups within a population, and $p_i$ is the size of group $i$ as a proportion of the total population. For our purposes, $k$ is the total number of post-developmental clones determined from the phylogeny (again defining a clone as originating at 100 mutations of molecular time) and $p_i$ is a clone's fraction determined from the targeted sequencing results (as described above), normalized to the total captured clonal fraction in that individual/cell type. For example, if clone $i$ has a clonal fraction of 0.1 and the sum of fractions across all clones is 0.5, $p_i = 0.2$.

### Estimating the relative size of driver mutations in donors and recipients

For each mutation of interest, the 100 posterior value estimates of the true mutation VAF in recipients were divided by the 100 estimates of the VAF in donors, giving a posterior distribution for the ratio. The median and 95% posterior intervals of this distribution were calculated.

### Simulation frameworks

Inference of engrafting cell number and demonstration of transplant-specific selection was performed using an ABC methodology, described in the next section. In ABC, a large number of simulated datasets, generated under the proposed model, takes the place of computation of the likelihood function for the model. Such simulations will never perfectly emulate the real-life scenario, but they can be useful to get a sense of biological parameters, within the constraints of the model used. To this end, we implemented several simulation models of allogeneic transplantation within the in-house developed R package 'rsimpop' v.2.2.4 (www.github.com/nickwilliamssanger/Rsimpop). This package allows simultaneous simulation of multiple-cell compartments, each with their own target population sizes, while recording the population phylogeny. It also allows subcompartments with differential fitness, mirroring the consequences of driver mutations. Population growth occurs through a birth–death process. Population growth occurs without cell death until a population reaches the target size, at which point the population is maintained with balanced cell birth/death.

The starting point of our simulations was the posterior distribution for the parameters of a model of normal ageing developed previously[19,20]. In our study of normal ageing[19], the ABC method was first applied to a neutral model of haematopoietic stem cell dynamics, which is applicable to younger individuals. Using this approach, it was possible to generate a large sample ($N$ = 2,000) of parameter values from the joint posterior distribution of the parameter $Nt$ (where $N$ is the HSC population size, and $t$ is the time between symmetric HSC cell divisions). In our study of ageing haematopoiesis, we further found that the changes in haematopoietic phylogeny structure seen with increasing age could be explained by constant acquisition of driver mutations with varying selection coefficients introduced into the HSC population through life[19].

The ABC method was used to generate a large sample ($N$ = 2,000) from the joint posterior distribution for the parameters of this model (specifying the rate of introduction of driver mutations into the population, and the distribution of selection coefficients of those mutations). We used this posterior distribution (as represented by the samples of parameter values) as the prior distribution for these same parameters in the ABC analysis of the transplant phylogenies reported here (Extended Data Fig. 9). We also returned to the neutral model, and applied it to the phylogeny data from the two youngest donors (aged 29 and 38) in that study, to generate a large sample from the posterior distribution of the parameter $Nt$. This posterior distribution was used as the prior distribution for the parameter $Nt$ in the ABC analysis of the transplant phylogenies.

**Simulation model 1: no transplant-specific selection.** Simulation begins with a single cell—the zygote of the HCT donor. Population growth occurs through a birth process until a target population size—the size of the HSC pool—is reached. As the previous estimates were for the value of $N_{HSC} \times t$, we keep a fixed value of $t$ for all simulations (the time between HSC symmetric divisions = 1 year) and choose $N$ as a random draw from the posterior estimates from a previous study[19]. Once reached, the target population size $N_{HSC}$ is maintained by matching cell division rates with cell death/differentiation. Driver mutations are added into single cells in the population at a fixed rate through time (random draw of posteriors from ref. 19), by assigning cells a selection coefficient $S_{homeostatis}$ (a random draw from a gamma distribution with shape and rate parameters themselves taken as a random draw from the posteriors from ref. 19), which is then passed on to all future cell progeny. This $S_{homeostatis}$ results in cells from driver clones being more likely to undergo symmetric cell division than others.

Simulation of donor haematopoietic ageing continues accordingly until the age of the donor at HCT, $Donor_age_{HCT}$. At this point, a number of HSCs ($N_{trans}$) are selected at random from the donor population of

HSCs to be transplanted into the recipient. This number was picked from a prior distribution:

$$\log_{10}(N_{trans}) \sim \text{Uniform}(\text{min} = 2.7, \text{max} = 4.7)$$

This results in absolute values of $N_{trans}$ ranging between 500 and 50,000. Within rsimpop, these engrafting HSCs are assigned to a new recipient compartment. Selection coefficients of transplanted clones harbouring driver mutations are maintained, but not altered, during HCT. Regrowth of the HSC population from $N_{trans}$ to the target $N_{HSC}$ population size and subsequent homeostatic haematopoietic ageing then proceeds independently within the donor and recipient until the time of the blood draw, donor_age$_{BD}$. At this point, the simulation is stopped and HSCs are picked at random from the donor and recipient compartments, corresponding experimentally to the cells grown into colonies that underwent WGS.

**Simulation model 2: incorporation of engraftment-specific selection.** Simulations initially proceed as in model 1. However, at the point of selecting the $N_{trans}$ HSCs to be transplanted, clones harbouring driver mutations were given an additional 'engraftment fitness' coefficient $S_{engraftment}$, independent of the usual steady-state selection coefficient $S_{homeostasis}$, which then was used as a weighting for the probability of their selection for transplant within the base R function sample. Engraftment fitness coefficients for each driver clone were chosen as a random draw from a truncated gamma distribution:

$$S_{engraftment} \sim \text{Gamma}(\text{shape} = 0.5, rate = 0.5)$$

These gamma distribution parameters were chosen empirically. The engraftment fitness of non-driver-containing cells was then set as the 30th centile value of all values of $S_{engraftment}$, such that some clones with driver mutations, conferring a selective advantage during homeostasis, may in fact have reduced fitness at engraftment.

**Simulation model 3: incorporation of post-engraftment selection.** Simulations proceed as in model 1. However, after transplantation, 10–30% of driver-containing clones within the recipient may have an exaggeration of their selection coefficient $S_{homeostasis}$ by 50–600%. This exaggeration of their selective advantage in the post-engraftment period is time-limited, continuing for 5 years, before reverting to the previous value. The motivation for the time-limited selective advantage is that the immediate post-transplant environment is unusual for several reasons: there is profound pancytopenia and the recipient bone marrow is hypoplastic after conditioning chemotherapy; the marrow microenvironment has recently been affected by leukaemia and intensive chemotherapy that may alter the selective landscape; there are frequently multiple infective or inflammatory episodes during the first few years after transplant as the innate and adaptive immune systems reconstitute; there is often residual host immunity that wanes over time. All of these factors are most pronounced in the early post-transplant period and are likely to resolve, at least partially, with time.

### ABC of engrafting cell number

Simulations were run for each pair ($n$ = 100,000) and key features of the separate donor–recipient phylogenies summarized by 13 statistics (illustrated examples of summary statistics from the recipient phylogenies shown in Extended Data Fig. 6): (1–3) the sizes of the largest 3 clades within the donor phylogeny; (4–6) as 1–3, but for the recipient phylogeny; (7) the number of singleton samples within the donor phylogeny (singleton is defined as a sample with no related samples from after the time of development); (8) as 7, but for the recipient phylogeny; (9) the number of coalescences within the donor phylogeny from around the estimated time of HCT, where this peri-HCT window is defined as coalescences occurring at an estimated age of between 5 years before, and 5 years after HCT; (10) as 9, but for the separate recipient phylogeny; (11) the number of coalescences in the donor phylogeny from an estimated timepoint after development, but before HCT, where this pre-HCT window is defined as coalescences occurring at an estimated age of between 5 years old, and 5 years before HCT; (12) as 11, but for the separate recipient phylogeny; (13), the maximum number of coalescences in the peri-HCT window (as defined in 9) within a single clade of the recipient phylogeny. This statistic was designed the capture the features of growth selection seen in the data.

Each vector of summary statistics computed from a simulated dataset was then compared to the vector of summary statistics computed from the experimentally generated data by calculating a Euclidean distance between these vectors. For this purpose, empirically modified versions of the experimentally generated phylogenies were used to provide best estimates of time trees, that is, those for which the height of a branch point represents the actual age at which that cell division occurred. For this, branch lengths were first corrected for sensitivity and sample clonality. The branch lengths were then shortened based on the estimated contribution of platinum and APOBEC mutational signatures—the sporadic signatures that are unlinked to time. Finally, terminal branches were shortened by 60 mutations, an estimate for the combined number of in vitro- and differentiation-associated mutations. This number was approximated based on (1) the excess of the $y$ intercept of the linear regression of SNV burden against age ($y$ intercept = 137; Extended Data Fig. 5a) over the known mutation burden at birth from other studies (SNV burden of ~60 in cord blood[19]). Moreover, the sum of estimates of the number of differentiation associated mutations (~30 mutations[19]) and typical numbers of in vitro acquired mutations during clonal expansion on methylcellulose (10–20 mutations, unpublished data) are of a similar order. After these branch-length corrections, the tree was made ultrametric using the previously described iteratively reweighted means algorithm, which assumes greater confidence for branch lengths where branches are shared by multiple samples[19].

Inevitably, the definitions of transplant epoch used in the summary statistics could have a key role in informing the parameter estimates. It is also the case that the timing of the coalescences is subject to some random variation in that mutations are acquired at a fairly constant rate, but the absolute numbers acquired in a given time period are subject to at least Poisson variation. To assess the robustness of the ABC analysis, we assessed whether this variation leads to significant uncertainty in the numbers of coalescences in each epoch. First, we used a bootstrapping approach whereby all branch lengths were redrawn from a negative binomial distribution with $\mu$ equal to the original number of mutations, and the $\Theta$ overdispersion parameter estimated from the distribution of HSPC mutation burdens in that pair (100 bootstraps performed for each pair). We then repeated the steps of making the tree ultrametric and scaling to time, and calculated the number of coalescences falling in each epoch used in the ABC. This demonstrated that the numbers are robust, with only subtle variation in some values where coalescences fall close to the borders between epochs (Supplementary Fig. 7).

Second, we assessed whether varying the specific definitions of the epochs used for summary statistics meaningfully altered the posterior distributions of the ABC. Specifically, we assessed four alternative sets of epochs: (1) dividing the pre-transplant interval into more epochs; (2) dividing the peri-transplant interval into more epochs; (3) using a narrower range of molecular time for the peri-transplant interval; and (4) using a wider range of molecular time for the peri-transplant interval. Reassuringly, across the different ABC models and parameters, the different donor–recipient pairs and the different methods for estimating the posterior, we found that the four alternative definitions of HCT epochs had minimal effect on the inferred posterior distributions (Supplementary Fig. 8).

In more detail, in the original set of summary statistics, the peri-transplant interval was an interval of duration 10 years, centred

on the time of transplant; and the pre-transplant interval began at age 5 years and ended at the timepoint where the peri-transplant interval begins (5 years before the time of transplant). In the pre_interval_divided set of summary statistics, the pre-transplant interval was replaced by two pre-transplant intervals, the first beginning at age 5 years and ending at the mid-point between 5 years of age and 5 years before the time of transplant. In the peri_interval_divided set of summary statistics, the peri-transplant interval was replaced by two peri-transplant intervals, each of duration 5 years. In the peri_interval_narrower set of summary statistics, the peri-transplant interval was an interval of duration 5 years, centred on the time of transplant. In the peri_interval_wider set of summary statistics, the peri-transplant interval was an interval of duration 15 years, centred on the time of transplant. At the same time as we compared the posterior densities generated using each of the five alternative sets of summary statistics, we also extended this comparison across four alternative ABC methods. These are the ABC rejection method and three ABC regression methods (ridge regression, local linear regression and a neural network method).

Comparisons were performed using the abc function of R package abc. Within this function, each summary statistic is standardized using an estimate of the standard deviation (the median absolute deviation). The Euclidean distance of each set of summary statistics from the data is then calculated. The closest 1% of simulations are accepted. The parameters from the accepted simulations represent a sample from the (approximate) posterior distribution. In the rejection sampling method, no regression step is performed. Where a regression model is used, this is applied as implemented within the abc function. However, for the primary results present in Fig. 2, the rejection sampling method was used as this was most robust to alternate summary statistics.

### ABC for estimates of phylogenetic age

Phylogenetic structure has been shown to become increasingly oligoclonal with age. In a previous study, the phylogenetic trees of 8 adults of varying ages were used to inform posterior estimates of fundamental parameters governing these features[19]. We ran an identical simulation framework—incorporating introduction of driver mutations into the HSC population at a constant rate—using the posterior parameter estimates from ref. 19 as starting parameter values. We ran 25,000 simulations, varying the age of the final trees from 20 to 100 years, and varying the size of the simulated phylogenetic trees to match that of the different individuals.

We used the abc function from the R package abc to infer posterior estimates of the age of each individual, looking at recipient and donor phylogenies separately. In contrast to the other ABC, phylogenies were assessed per individual (not HCT pair) and therefore a smaller set of seven summary statistics was used to compare with the data: (1–3) the size of the largest 3 clades; (4) the number of singleton samples; (5–6) the number of coalescences in the 20–40th and 40–60th centile bins of the tree; and (7) the proportion of samples lying within expanded clades, defined here clades containing at least 3 of the sequenced samples. A clade here is defined as a set of samples with a common ancestor after 50 mutations of molecular time (corresponding approximately to post-embryonic development).

The age of the top 5% of simulations were chosen for initial estimates of phylogenetic age. As before, a neural network regression was then performed to refine these estimates.

Using the lme function from the R package lme4, we performed a linear mixed-effects regression to estimate the impact of donor/recipient status on phylogenetic age. All individual posterior estimates of phylogenetic age were used in the regression. Fixed effects in the model were donor age (continuous predictor variable), and donor/recipient status (categorical predictor variable). No interaction term was used. HCT pair ID was considered a random effect to account for the non-independence of the sets of posterior estimates.

### Statistics for pruning and growth selection

We wanted to design statistics to capture and quantify the features of pruning and growth selection described in the 'Causes of reduced clonal diversity' section and shown in Fig. 4a–c, in a clone-specific manner, to reflect that different clones may experience an advantage at different points.

For each expanded clade, we wanted to quantify the increase of coalescences either (1) before the time of HCT, or (2) around the time of HCT, in the recipient compared to the donor. However, the growth selection statistic may be increased by neutral mechanisms in the context of a population bottleneck, and therefore is only strong evidence of selection where the total number of peri-transplant coalescences from across the tree are biased to that specific clade.

**Pruning selection statistic.** We first calculate 1 + the number of recipient coalescences in an expanded clade that time to the pre-HCT time window as a proportion of 1 + the total number of coalescences in that clade.

$$\text{Pruning selection statistic} = \left(\frac{1+\ n_{\text{pre},R}}{1+N_R}\right)\Bigg/\left(\frac{1+\ n_{\text{pre},D}}{1+N_D}\right)$$

Where $n_{\text{pre},R}$ is the number of recipient coalescences in the specific expanded clade that time to the pre-HCT time window, $N_R$ is the total number of recipient coalescences in that expanded clade, and $n_{\text{pre},D}$ and $N_D$ are the equivalent numbers for the same expanded clade in the donor phylogeny. All values have one added to avoid dividing by zero.

**Growth selection statistic.** This is similar to the pruning selection statistic, but is focused instead on coalescences in the peri-HCT time window (those that time from five years before until five years after HCT).

$$\text{Growth selection statistic} = \left(\frac{1+\ n_{\text{peri},R}}{1+N_R}\right)\Bigg/\left(\frac{1+\ n_{\text{peri},D}}{1+N_D}\right)$$

Where $n_{\text{peri},R}$ is the number of recipient coalescences in the specific expanded clade that time to the peri-HCT time window, $N_R$ is the total number of recipient coalescences in that expanded clade, and $n_{\text{peri},D}$ and $N_D$ are the equivalent numbers for the same expanded clade in the donor phylogeny. All values have one added to avoid dividing by zero.

### Estimating the anticipated impact of long-lived T cell clones on T cell clonality

Our targeted sequencing results show that substantially lower proportions of T cells compared to myeloid cells derive from expanded clones at a given timepoint (when considering clones known from the phylogeny). We put forward several potential contributors to this difference in the 'Clonal output within lymphoid compartments' section, one of which is that, at any given time, the clonal make-up of T cells reflects HSC output from up to 8–15 years earlier. Given that oligoclonality of the HSC compartment increases with age, the decreased clonality of T cells may simply reflect the more polyclonal output of these younger HSCs.

To assess how much of the observed difference in expanded clone proportions may result from this, we performed simulations (according to simulation framework 1) in which we compared the oligoclonality of the HSC population from 4–8 years before the time of blood sampling (reflecting the average age of T cells that have a lifespan of 8–15 years, that is, the average age is approximately half the lifespan at steady state), compared to that at the time of blood sampling (reflecting the age of the short-lived myeloid cells). We performed 3,000 simulations per individual, varying the lifespan between 8–15 years, and comparing the total proportion of T cells from expanded clones to myeloid cells from expanded clones as a function of the life span of T cell clones.

## Reporting summary

Further information on research design is available in the Nature Portfolio Reporting Summary linked to this article.

## Data availability

Whole genomes and targeted sequencing data have been deposited in the European Genome–Phenome Archive under accession numbers EGAD00001010872 (WGS data) and EGAD00001010874 (targeted sequencing data).

## Code availability

Code underpinning the analyses reported here is available on GitHub (https://github.com/mspencerchapman/Clonal_dynamics_of_HSCT). Larger files of data necessary to reproduce some of the analysis in the GitHub repository are available on Mendeley Data (https://data.mendeley.com/datasets/m7nz2jk8wb/1).

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

**Acknowledgements** This work was supported by the WBH Foundation. Investigators at the Sanger Institute are supported by a core grant from the Wellcome Trust. Work in the D.G.K. laboratory is supported by a Bloodwise Bennett Fellowship (15008), a Cancer Research UK Programme Foundation Award (DCRPGF\100008) and a European Research Council Starting Grant (ERC-2016-STG–715371). Work in the A.R.G. laboratory is supported by the Wellcome Trust, Bloodwise, Cancer Research UK, the Kay Kendall Leukaemia Fund, the Leukemia and Lymphoma Society of America and a core support grant from the Wellcome Trust and Medical Research Council to the Cambridge Stem Cell Institute.

**Author contributions** P.J.C., M.S.C. and M.G.M. designed the experiments. M.S.C. performed the data analysis and designed the simulation frameworks. M.S.C. and P.J.C. wrote the manuscript. C.M.W., S.B., J.M., L.K. and F.C. recruited the patients, collected clinical data, processed the cells, grew the single-cell-derived colonies and performed the cell-sorting experiments. N.W. designed and optimized tools to analyse phylogenies, and developed software to simulate somatic population phylogenies. H.J. performed the SV analysis. P.J.C., M.G.M., J.N., A.R.G. and D.G.K. supervised the project. K.R. and L.O. processed the samples for sequencing. K.D. developed a pipeline for running the simulation models and developed and generated the posterior predictive checks. All of the authors reviewed and edited the manuscript.

**Competing interests** P.J.C. is a co-founder, stock holder and consultant for Quotient Therapeutics.

**Additional information**


**Correspondence and requests for materials** should be addressed to Markus G. Manz or Peter J. Campbell.


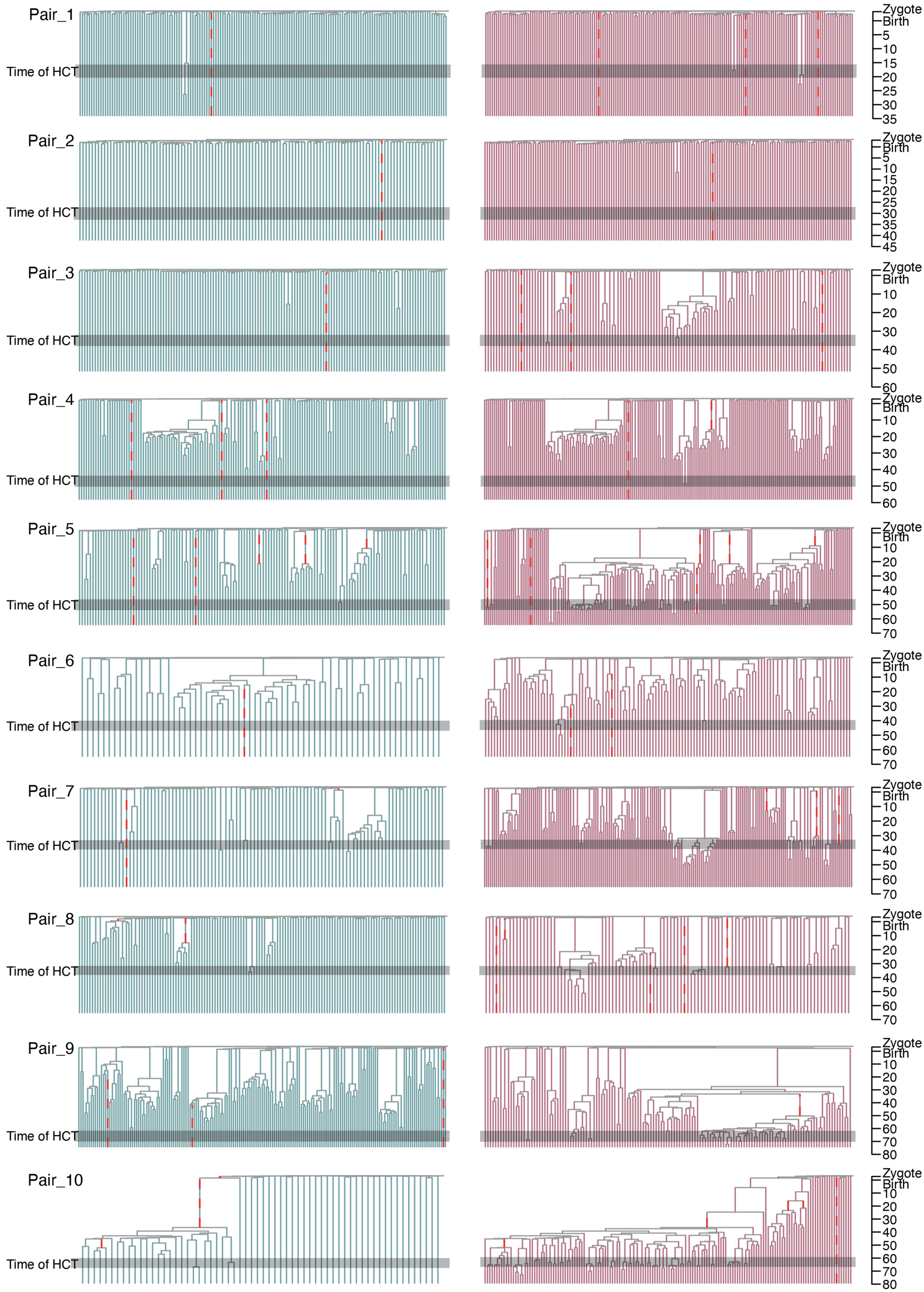


**Extended Data Fig. 1 | Separate donor-in-donor and donor-in-recipient phylogenies.** Phylogenies of HSPCs collected from donors (cyan, left side) and recipient (pink, right side). Phylogenies have been made ultrametric and scaled to real time (Methods). Grey shaded boxes indicate the approximate time of HCT. Branches with driver mutations are highlighted (red dashed lines).

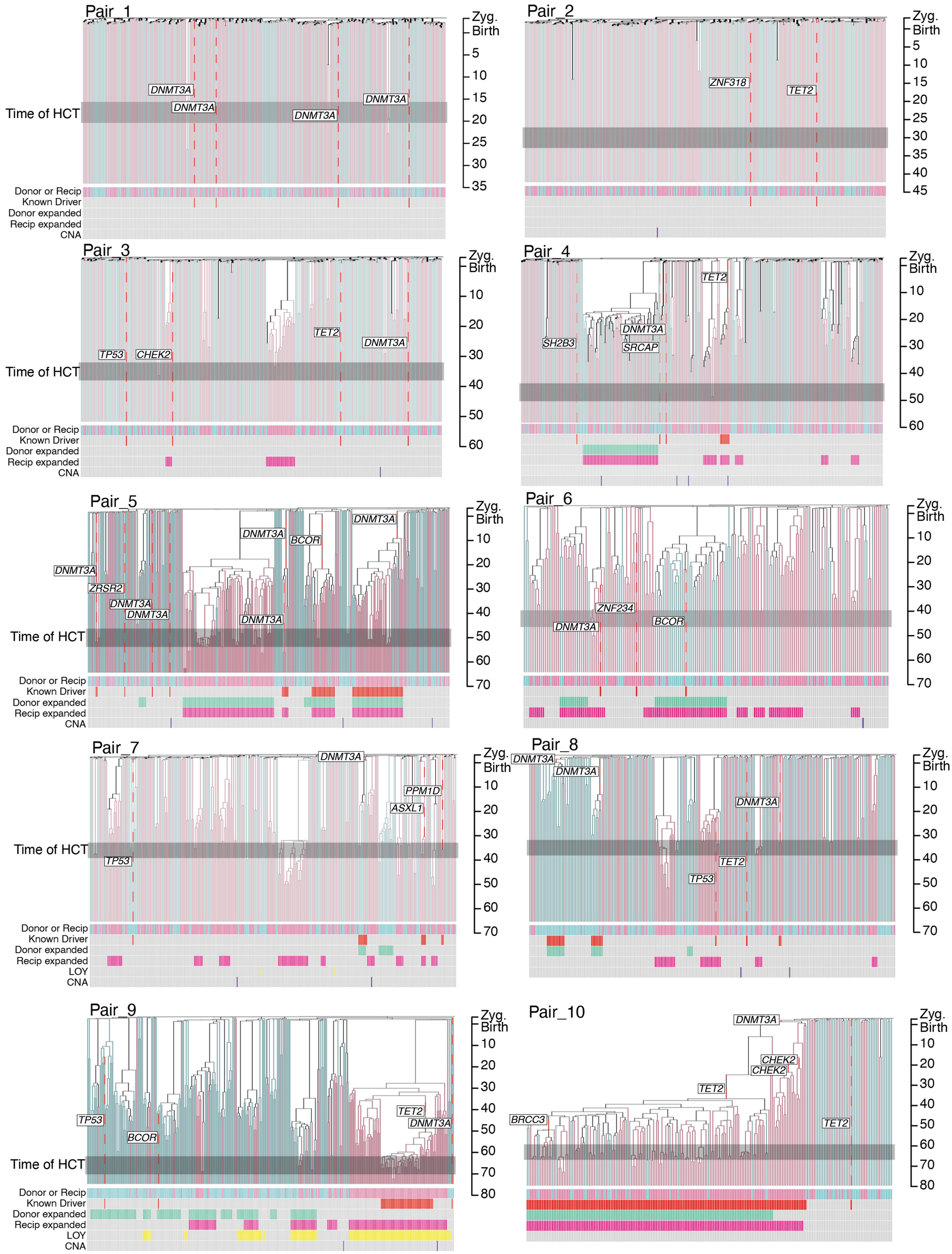


**Extended Data Fig. 2 |** See next page for caption.

**Extended Data Fig. 2 | Combined donor-in-donor and donor-in-recipient phylogenies.** Phylogenies have been made ultrametric and scaled to real time (Methods). Grey shaded boxes indicate the approximate time of HCT. Branches with driver mutations are shown (red dashed lines) labelled with the mutant gene. Branches were found in donor HSPCs only (cyan), recipient HSPCs only (pink) or both (black). Heatmaps show additional colony-level information. HCT, haematopoietic cell transplantation; LOY, loss of Y chromosome; CNA, Copy number alteration; Recip, Recipient, Zyg., Zygote.

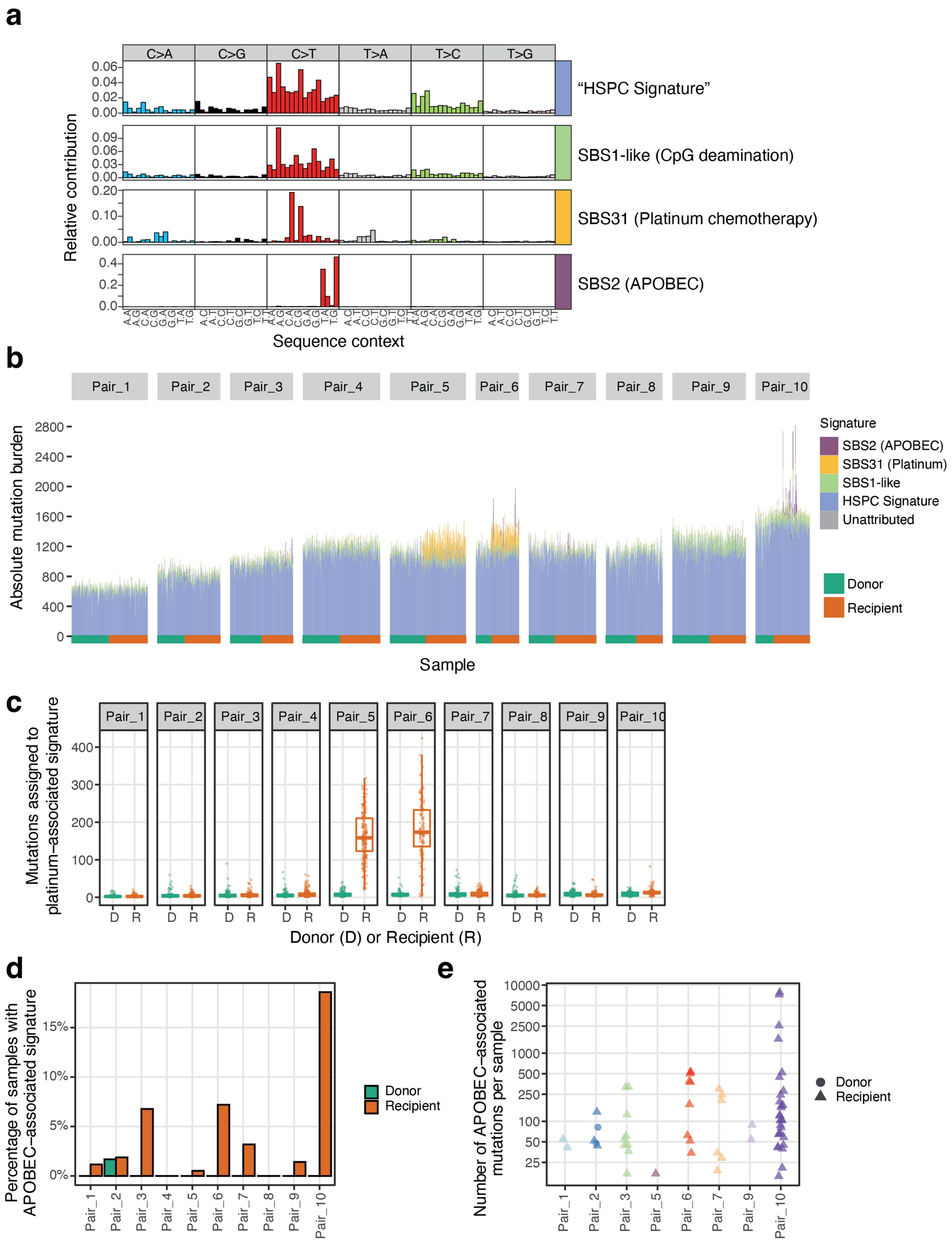


**Extended Data Fig. 3** | See next page for caption.

**Extended Data Fig. 3 | Mutational signatures. a**, 96-profile mutational signatures of mutational processes active in HSCs/HSPCs, as extracted using a hierarchical dirichlet process (Methods). Interpretation of each signature, by comparison with COSMIC signatures, is shown to the right of each profile. **b**, Stacked barplot showing the absolute contribution of each signature to each sample. Each column is a single sample, with samples grouped by pair. Tiles below the columns indicate whether the sample is from the donor (green) or recipient (orange). Three outlier samples in the Pair 10 recipient had extremely high burdens and these have been attenuated to aid visualization. **c**, Box-and-whisker plot showing the per sample burden of N3 mutations (platinum-associated signature), divided by pair and donor/recipient origin. Lines show the median values, boxes show the interquartile range, and whiskers the range for the n = 10 independent donor-recipient pairs. **d**, Bar plot showing the percentage of HSPCs from each donor (green) and recipient (orange) that are "positive" for the SBS2 signature (APOBEC-associated, ≥10 mutations). **e**, Jittered dot plot showing the absolute burden of SBS2 mutations in positive samples. Points are coloured by pair, and are either triangles (recipient origin) or circles (donor origin). HSPC, haematopoietic stem or progenitor cell; SBS, single base substitution.

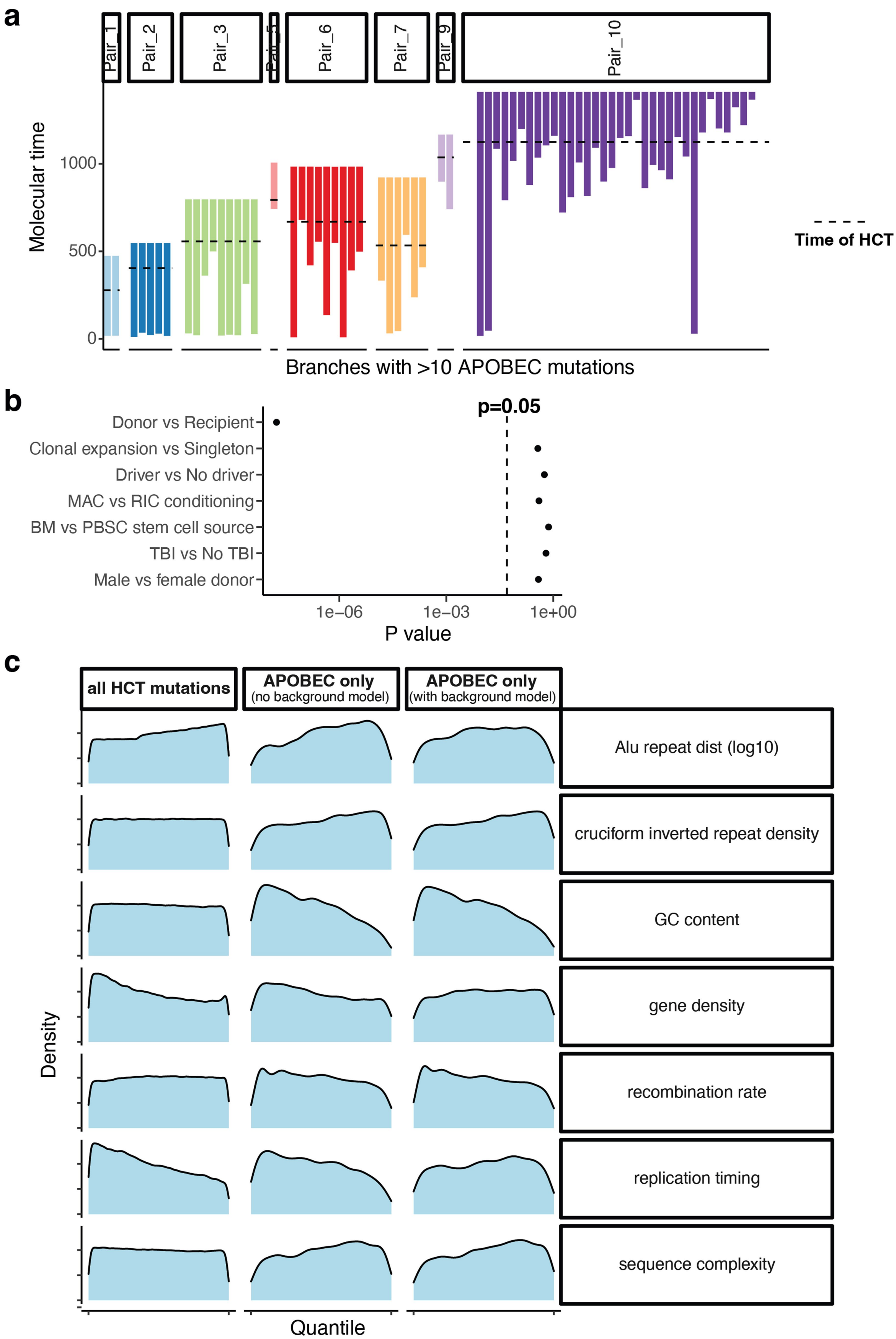


**Extended Data Fig. 4** | 

**Extended Data Fig. 4 | Properties of APOBEC mutations. a**, Estimated timings of branches containing >10 APOBEC mutations. Using the estimated clock-like mutation rate for HSPCs, branch start- and end-points in molecular time were converted to estimated chronological age and plotted as a vertical bar. Estimated timing of transplant for each recipient is plotted as a horizontal dashed line. All bars end above this age-of-transplant line, and many begin above it, suggesting that APOBEC mutations occur at the time of or subsequent to the transplant. **b**, P values of a generalized linear mixed effects model to identify factors predicting presence of APOBEC mutations in a given colony, showing that the only significant variable was the enrichment of the signature in recipient versus donor colonies. **c**, Genomic features significantly associated with distribution of APOBEC mutations in recipient colonies. Associations between different genomic properties (rows) and all mutations (left column), APOBEC mutations (middle column) and APOBEC mutations normalized by the density of non-APOBEC mutations (right column). Each density curve represents the quantile distribution of the genomic property values at observed positions of mutations compared to random genome positions. Shown are the genomic properties that are statistically significant using generalized additive models after multiple hypothesis test correction ($q < 0.1$).

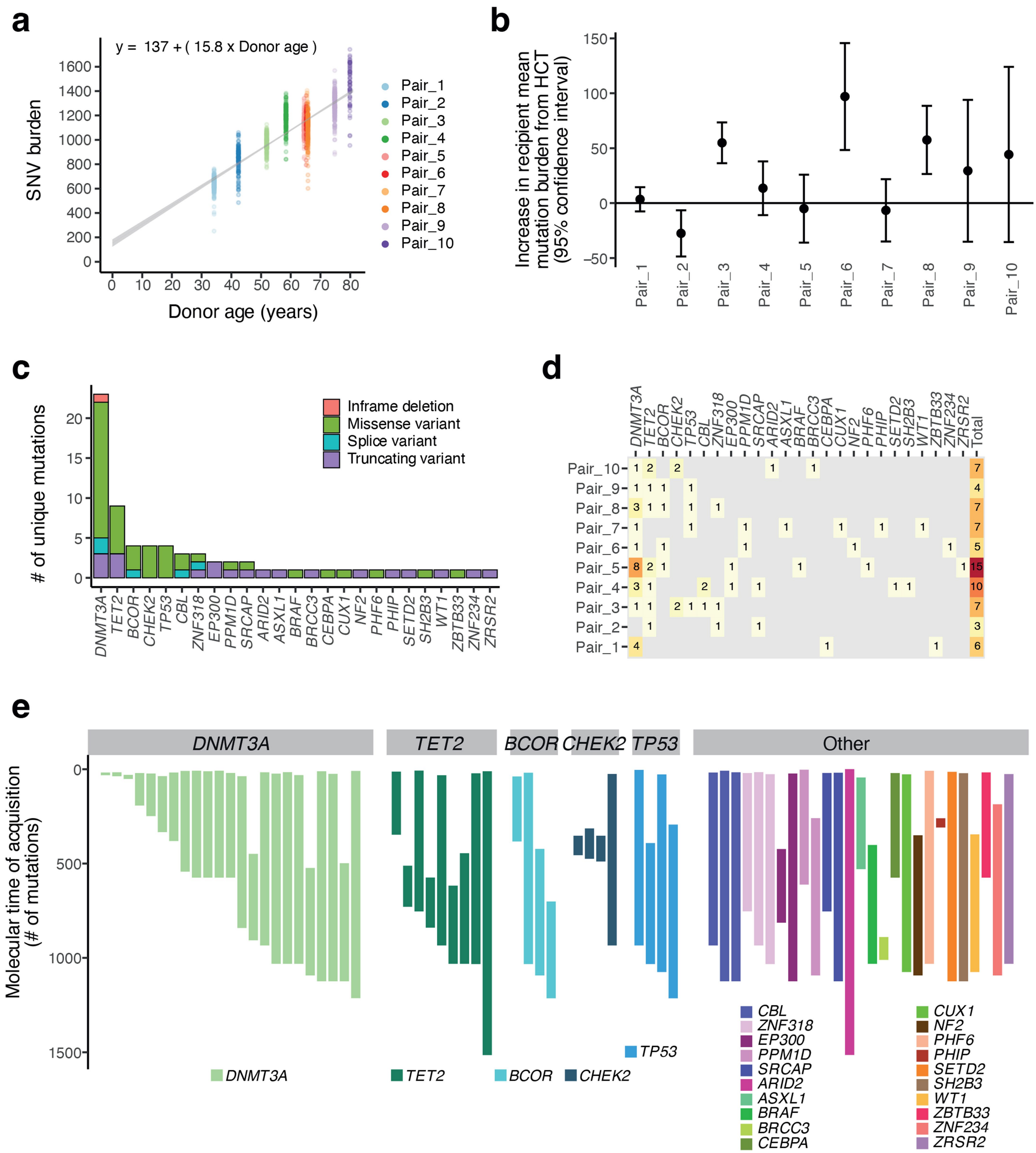


**Extended Data Fig. 5 | Mutation burdens and driver mutations. a**, Dot plot showing the corrected single nucleotide variant mutation burden of HSPCs from donors against donor age. Solid black line shows the results of a linear regression of this relationship, with the grey shaded area the 95% confidence interval. **b**, Dot plot showing the number of additional mutations in recipient colonies, after bioinformatically removing burdens associated with the APOBEC (N4) and platinum chemotherapy (N3) signatures that are sporadic. Where there are multiple HSPCs from a single expansion, only one colony per individual is used for this inference, as the burdens are not independent. Circles denote the point estimate and error bars indicate the 95% confidence intervals calculated from n = 2,824 independent colonies. **c**, Stacked bar plot showing the total numbers of independent driver mutations detected per gene, coloured by mutation consequence. **d**, Heatmap showing the number of independent driver mutations per gene in each pair. The far right column shows the total number of drivers in each individual across genes. **e**, Bar plot showing the possible molecular times of acquisition of each driver mutation. Bars are grouped and coloured by gene. SNV, single nucleotide variant; HCT, haematopoietic cell transplantation.

## Approximate Bayesian Computation (ABC) using rejection sampling method

### Model: Normal haematopoietic ageing (Mitchell et al 2022) + recipient population bottleneck at HCT

#### Priors for parameters with uncertain values

1. $N_{TRANS}$ (Number of engrafting HSCs): Uniform prior for $\log_{10}(N_{TRANS})$ on interval [2.7,4.7] (equivalent to $500 < N_{TRANS} < 50{,}000$)
2. HSC population size: posterior estimates from Mitchell et al.
3. Number of drivers per year: posterior estimates from Mitchell et al.
4. Driver mutation selection coefficient distribution - rate parameter: posterior estimates from Mitchell et al.
5. Driver mutation selection coefficient distribution - shape parameter: posterior estimates from Mitchell et al.

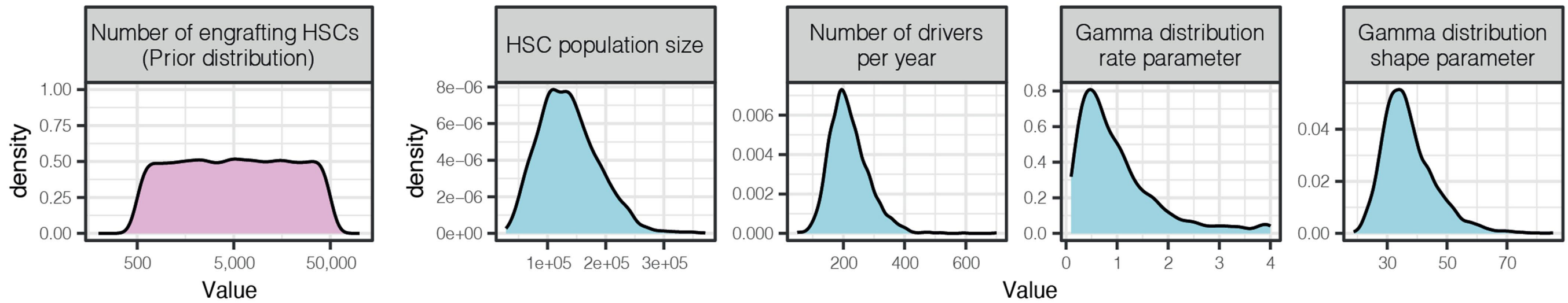


#### Summary statistics for ABC

1. Number of coalescences within (1) pre-transplant and (2) peri-transplant time windows. Pre-transplant window defined as 5 years of age until 5 years prior to HCT. Peri-transplant window defined as 5 years before until 5 years after HCT. Calculated independently for donor and recipient phylogenies (shown below for Pair_5 recipient).
2. Largest three clades. Calculated independently for separate donor and recipient phylogenies (shown below for Pair_5 recipient).
3. Number of singletons. Calculated independently for separate donor and recipient phylogenies (shown below for Pair_5 recipient).
4. Maximum number of peri-transplant coalescences in single clade. Calculated for recipient phylogeny only (shown below for Pair_5 recipient).

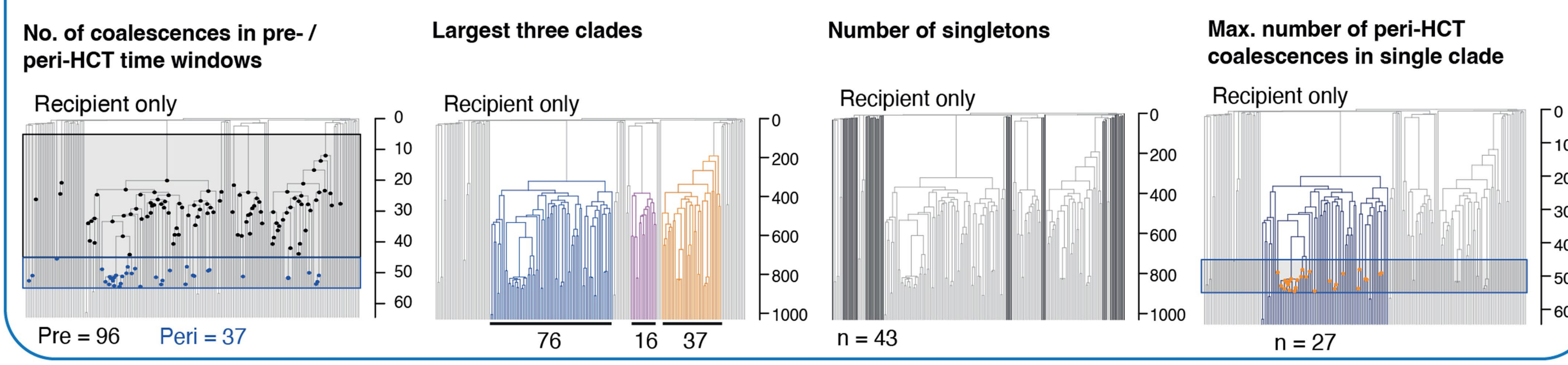


### Approximate Bayesian Computation (ABC) with rejection sampling

Pair-specific ABC performed separately for each donor/ recipient transplant pair. For each pair:

- 100,000 parameter vectors sampled from priors/ input parameter distributions
- Data are simulated from each parameter vector using *rsimpop* — primary model is that without transplant-specific selection (see Methods)
- For each simulated data set (combined/ donor/ recipient phylogenies), summary statistics are calculated
- Each summary statistic is standardised by a robust estimate of the standard deviation
- Simulations are ranked by Euclidean distance between simulated and observed summary statistics
- The closest 1000 simulations are accepted (proportion = 1000/100,000 = 1%)
- The parameters from the accepted simulations represent a sample from the (approximate) posterior distribution.
- In the rejection sampling method, no regression step is performed.

### Posterior predictive checks (PPC)

Pair-specific (posterior predictive) p-value calculated. For each pair:

- For each parameter vector, 1000 new simulated data sets are generated
- From each simulated data set, a vector of summary statistics are calculated
- From each parameter vector and simulated vector of summary statistics, a chi-squared discrepancy is computed
- The proportion of simulations where the simulated chi-squared discrepancy exceeds the observed discrepancy is the estimated p-value

**Summary statistics used for PPC**
For the PPC, the full set of 13 summary statistics was used to generate the posterior and perform the posterior precictive check

**Interpretation of the posterior predictive p-values**
If the p-value is close to zero, this is evidence against the proposed model as an explanation for the features of the data captured by the SS
If the p-value is close to zero, then the observed data is an outlier compared to the data predicted under the proposed model.

**Extended Data Fig. 6 | Bayesian inference for the number of engrafting haematopoietic stem cells during HCT.** Overview of the modelling and inference approaches used to estimate the numbers of long-term engrafting HSCs for each transplant pair. The modelling approach is described in detail in the Methods section.

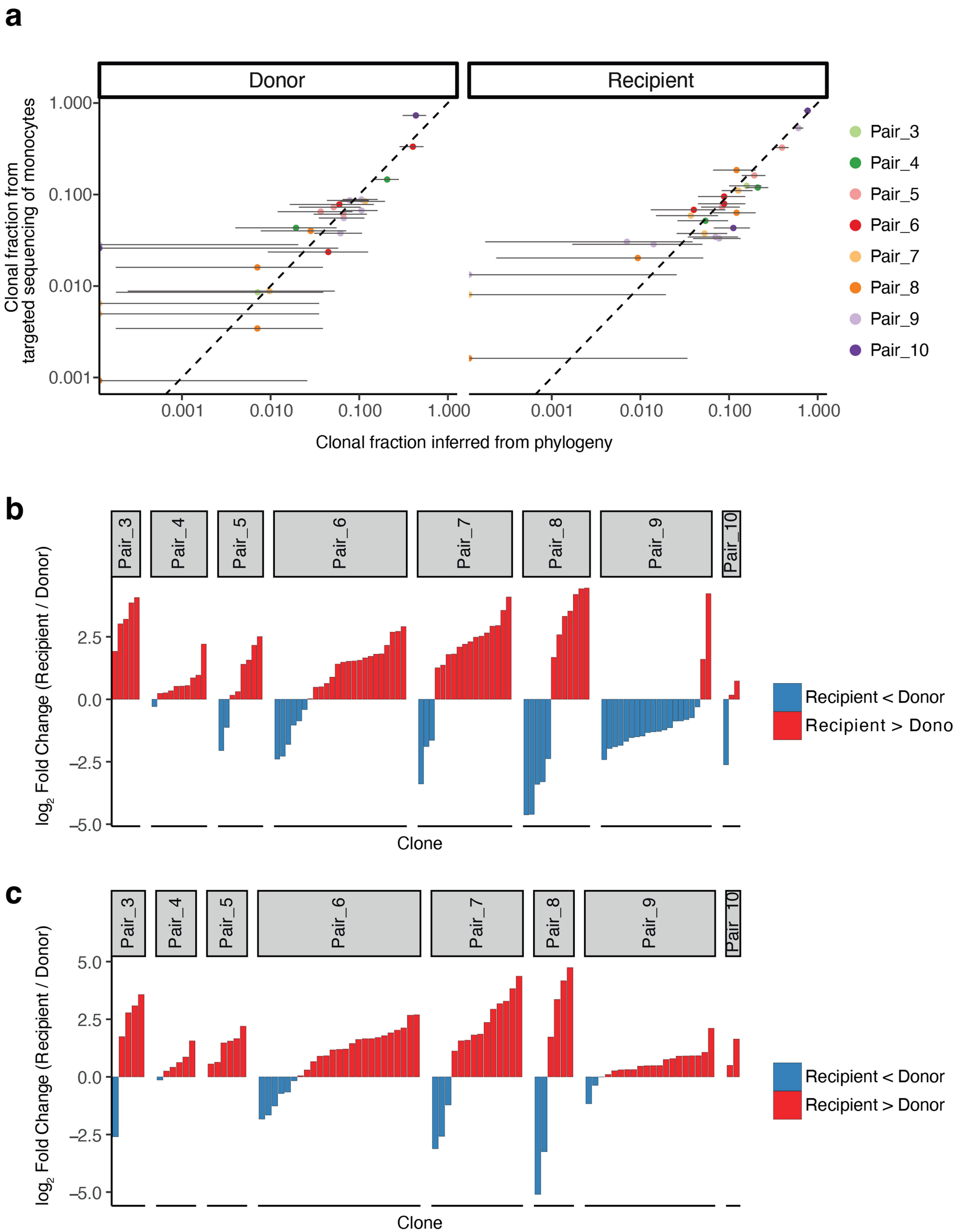


**Extended Data Fig. 7** | 

**Extended Data Fig. 7 | Comparisons of expansion clonal fractions across cell types and individuals. a**, Clonal fractions inferred from the phylogeny compared to targeted sequencing of monocytes. Plot shows only clones that are at least 5% clonal fraction in either donor or recipient. The x-axis shows clonal fractions inferred from the proportion of colonies from that individual coming from that clone, with circles denoting the point estimate and error bars giving the 95% confidence interval (exact binomial test). The y-axis shows clonal fractions inferred from the deep targeted sequencing of monocyte fractions. Confidence intervals for the targeted sequencing data are generally narrow and therefore not shown. **b**,**c**, Bar plots showing the $log_2$ fold change of expanded clone sizes within monocytes (**b**) or B-cells (**c**) comparing recipients to their donors. Bars are coloured red if the clone is larger in the recipient, or blue if larger in the donor.

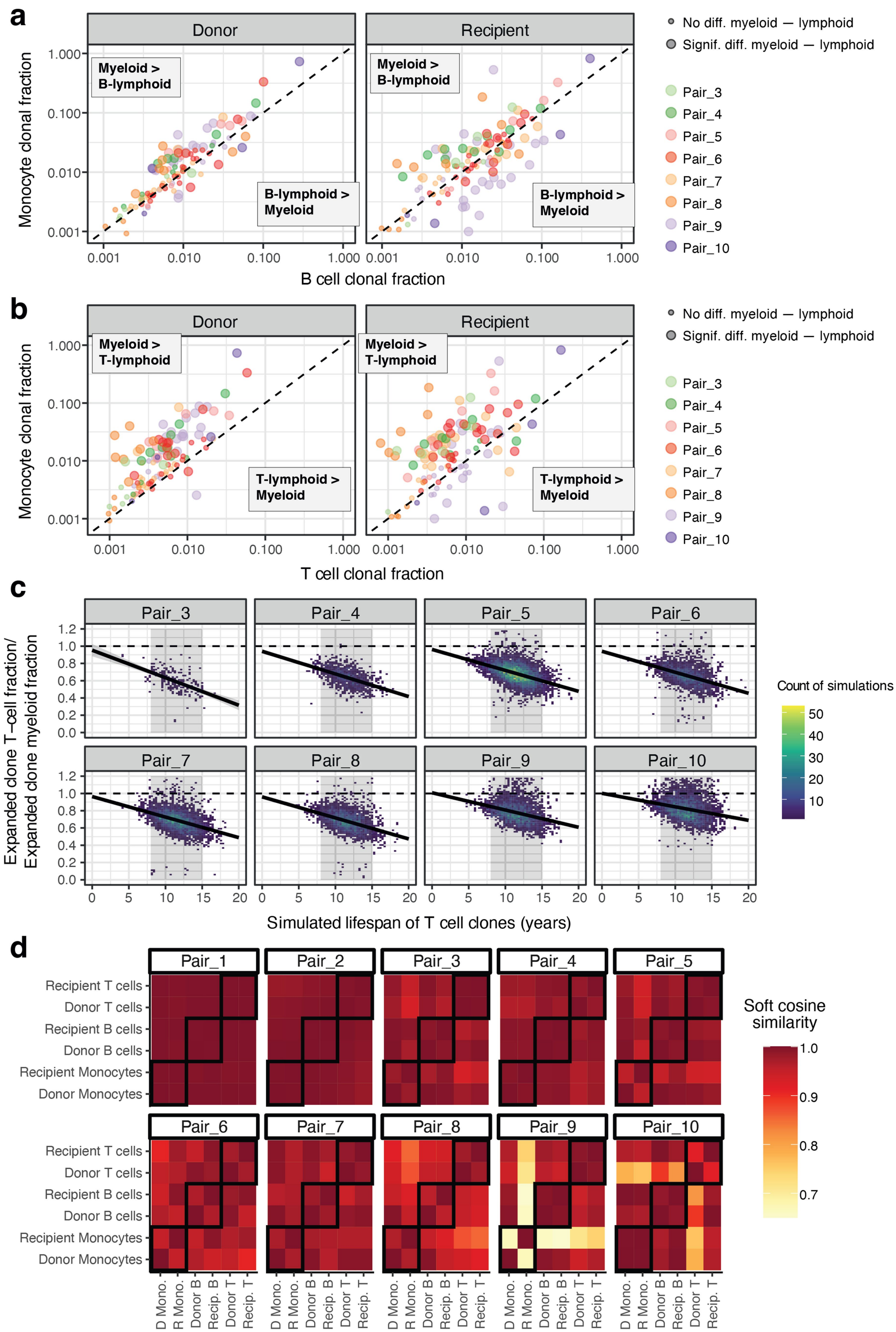


**Extended Data Fig. 8** | See next page for caption.

**Extended Data Fig. 8 | Lower output of detected clones in the lymphoid compartments. a**,**b**, Dot plot showing the clonal contribution of different clones to the myeloid compartment as compared to the B-lymphoid compartment (**a**) or T-lymphoid compartment (**b**) at the time of sampling, split by donor and recipient. **c**, Line plot showing the sum of T-cell clonal fractions across the branches of the phylogenetic tree at different points in molecular time. The earliest time point shows the sum of clonal contributions of the first two blastomeres of the embryo. Solid line shows the median posterior values, shaded areas show the 95% posterior intervals. **d**, Heatmap showing the soft cosine similarities of early embryonic mutations across mature cell types in the 10 donor-recipient pairs.

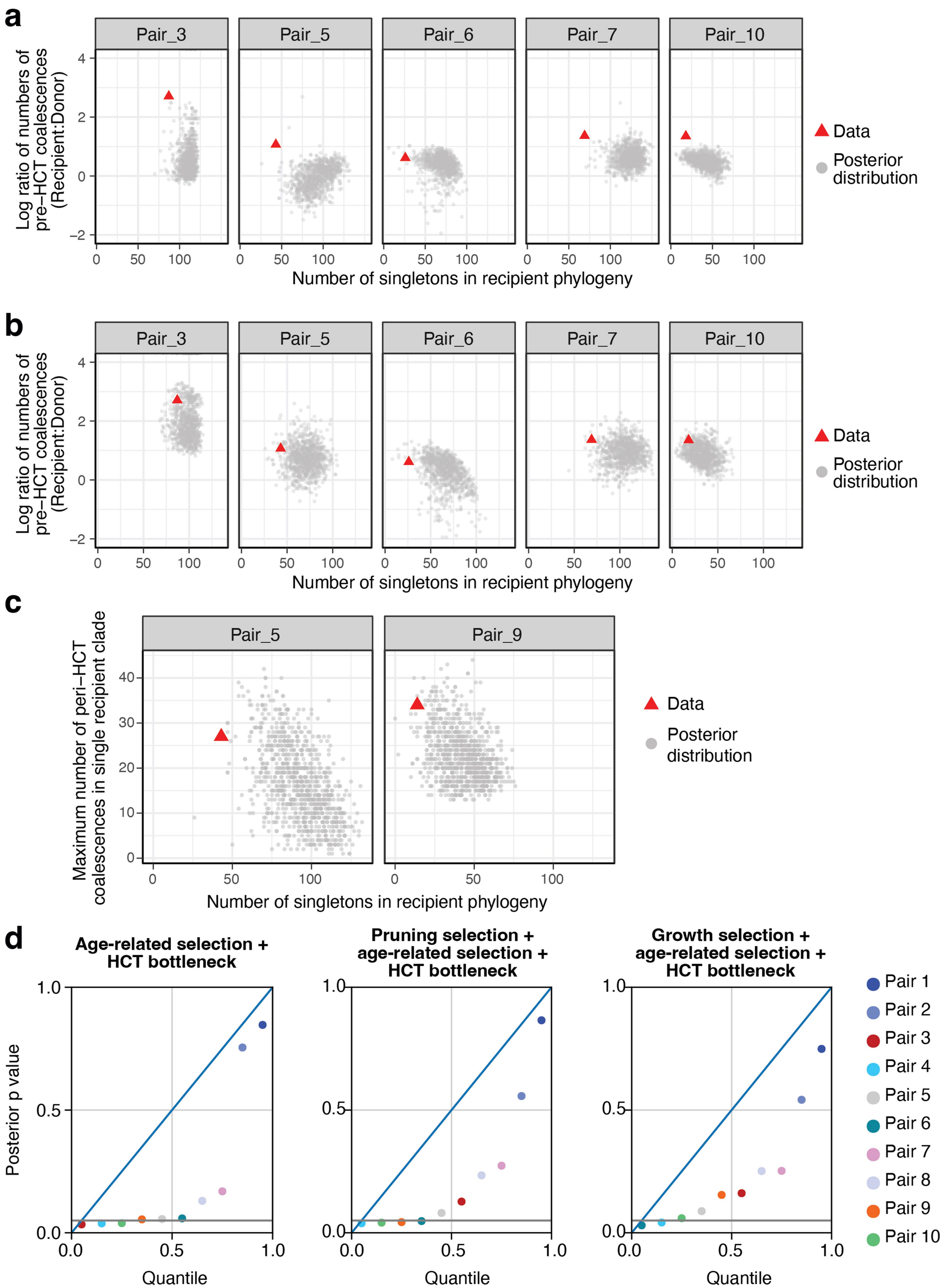


**Extended Data Fig. 9 |** See next page for caption.

**Extended Data Fig. 9 | Simulations incorporating HCT-specific selection are necessary to recapitulate real HCT phylogenies. a**, Dot plot showing selected summary statistics for the samples from the posterior distribution (grey) compared to the data (red), when using a simulation framework that does not incorporate engraftment-specific selection. These summary statistics reflect the degree to which recipient phylogenies have increased pre-HCT coalescences (recipient:donor ratio of pre- HCT time point coalescences, y-axis), while maintaining overall diversity (number of singletons, x-axis). **b**, As in **a**, but now using a simulation framework that allows for engraftment-specific selection (Pruning selection). **c**, As in **a**, but with different summary statistics, now reflecting the degree to which peri-HCT coalescences are concentrated in a single clade (maximum number of peri-HCT coalescences in single clade, y-axis), while maintaining overall diversity (number of singletons, x-axis). **d**, Quantile-quantile (QQ) plots showing distributions of posterior p values for the three ABC models, calculated using Bayesian posterior p value checks with the rejection sampling method. In each panel, the posterior p values are ranked (x axis; quantile) and the posterior p value is shown (y axis), coloured by donor-recipient pair. The blue lines represent x = y and the grey lines represent y = 0.05. Left panel, model of age-related selection combined with a bottleneck for transplant into recipient; middle panel, model of age-related selection, bottleneck plus pruning selection; right panel, model of age-related selection, bottleneck plus growth selection.

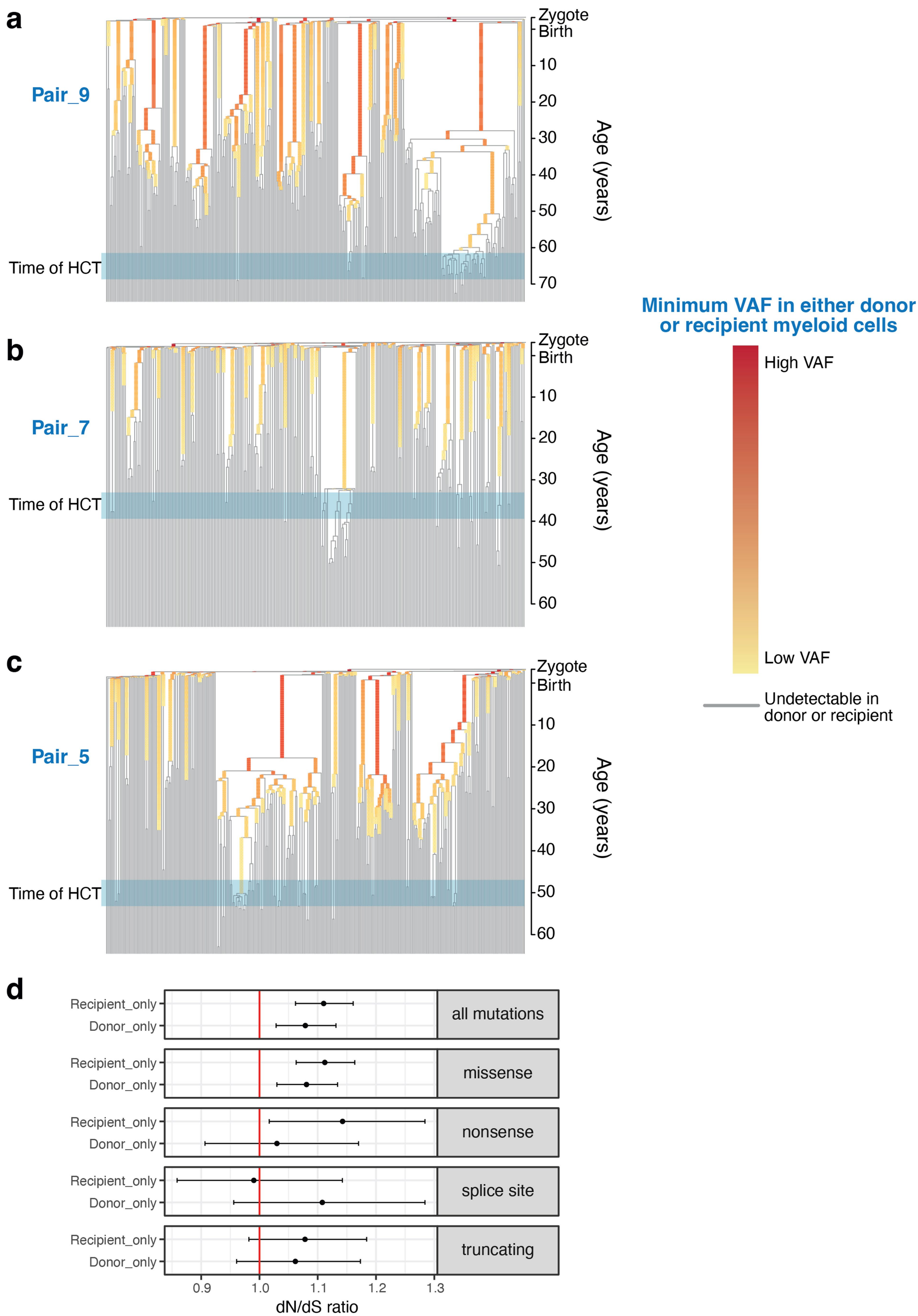


**Extended Data Fig. 10** | See next page for caption.

**Extended Data Fig. 10 | Minimum VAF in either donor or recipient myeloid cells.** **a**, Combined phylogeny for Pair_9, with mutations coloured by the lower value of the estimated posterior median VAF in granulocytes from either the donor or the recipient. This means any coloured branch is detectable in both donor and recipient, including several of the clonal expansions. Trees have been made ultrametric and scaled to chronological age. Shaded grey box indicates approximate time of HCT. **b**, As in **a**, but for Pair_7. **c**, As in **a**, but for Pair_5. **d**, Estimated dN/dS ratios combined across pairs, split by mutations found in donors or recipients and mutation type, corrected for expected freuqncy based on background mutation rates, sequence context and chromatin state. A dN/dS ratio of >1.0 implies more positive than negative selection; a ratio <1.0 implies more negative than positive selection. Circles denote point estimates and bars denote 95% confidence intervals estimated across mutations called from n = 10 donor-recipient pairs.

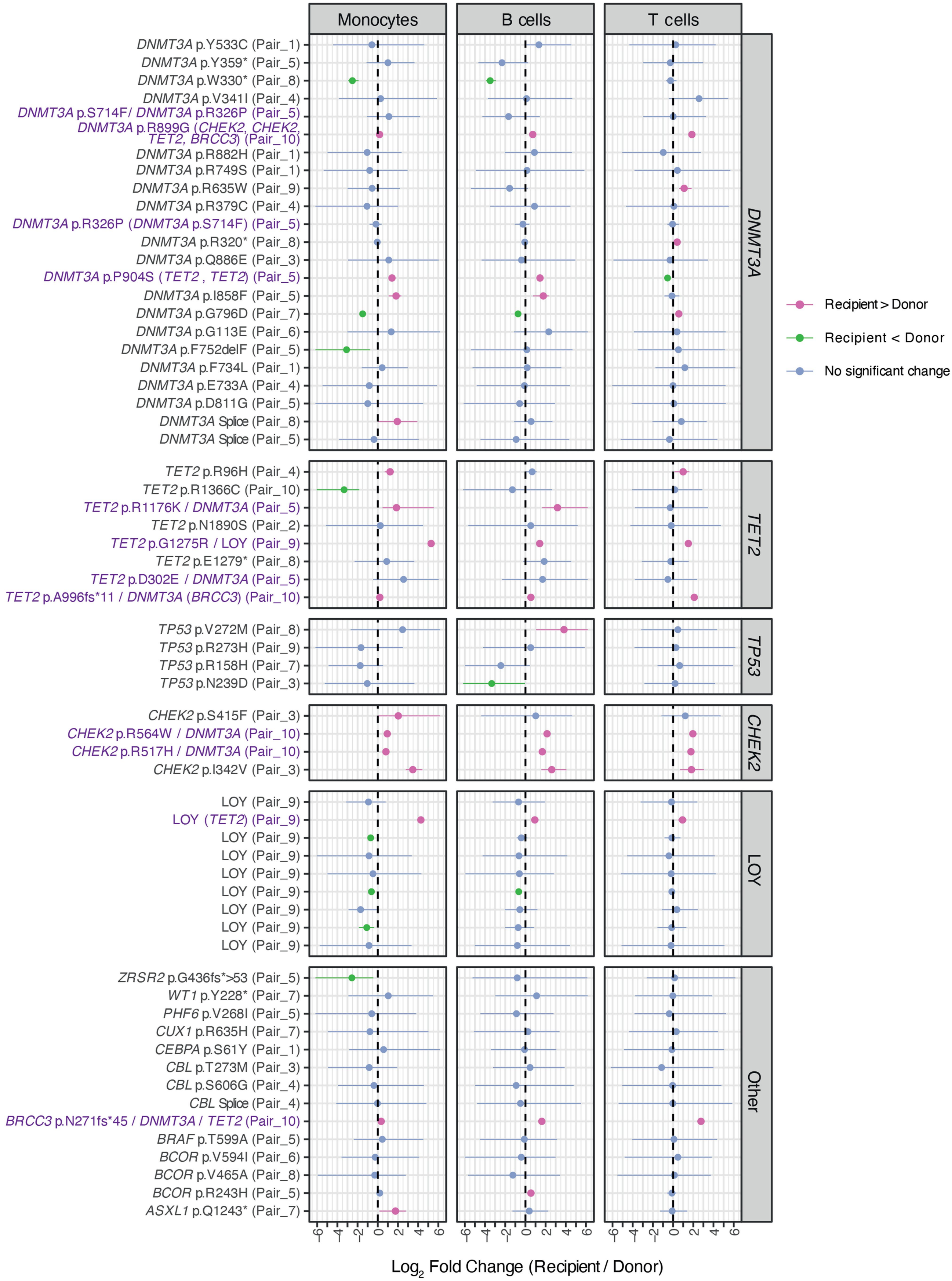


**Extended Data Fig. 11** | See next page for caption.

**Extended Data Fig. 11 | Relative clonal fraction of driver events in different mature cell compartments. a**, Dot plot showing the $\log_2$ fold change of putative driver event VAFs in recipients compared to donors in monocytes (left panel), B-cells (middle panel) and T-cells (right panel). Points are coloured by whether fractions are lower (orange) or higher (green) in recipients, or show no significant difference (blue). Circles show the point estimate and error bars show the 95% confidence interval of the $\log_2$ fold change. Variants are grouped by the affected gene or chromosome. Where mutations occur on the background of another driver mutation, genes are shown in the format GENE1/GENE2, indicating that all cells in the clone have driver mutations in both genes. Where the mutant clone contains a major subclone with an additional driver mutation, these are shown in the format GENE1 (GENE2), indicating that some, but not all cells in the clone have driver mutations in both genes. LOY, loss of the Y chromosome.

## Extended Data Table 1 | Demographic data for research participants

| Pair ID | Donor or Recipient | Individual ID | Diagnosis | Sex | Age at HSCT (years) | Age difference (Recipient - Donor) | Date of original blood sampling | Age at study visit (years) | Time since HSCT (years) | Conditioning regimen | Conditioning type | Stem cell source | MNC x 10^8/kg | Notes |
|---|---|---|---|---|---|---|---|---|---|---|---|---|---|---|
| Pair_1 | Recipient | PD45801b | sAML | Female | 21 | 2.18 | 16/01/2017 | 36.3 | 15 | BuCy | MAC | PBSC | Unnknown | |
| Pair_1 | Donor | PD45800b | | Female | 18 | | 16/01/2017 | 34.2 | 15 | | | | | |
| Pair_2 | Recipient | PD45813b | AML | Male | 24 | -6.56 | 13/03/2017 | 35.8 | 12 | Cy-TBI | MAC | PBSC | 15.01 | |
| Pair_2 | Donor | PD45812b | | Female | 30 | | 13/03/2017 | 42.4 | 12 | | | | | |
| Pair_3 | Recipient | PD45811b | AML | Female | 30 | -4.62 | 13/03/2017 | 47.3 | 16 | Cy-TBI | MAC | BM | 2.48 | |
| Pair_3 | Donor | PD45810b | | Female | 35 | | 13/03/2017 | 51.9 | 16 | | | | | |
| Pair_4 | Recipient | PD45803b | AML | Male | 58 | 11.27 | 16/01/2017 | 69.7 | 11 | Flu-Bu-ATG | RIC | PBSC | 10.94 | |
| Pair_4 | Donor | PD45802b | | Male | 47 | | 16/01/2017 | 58.4 | 11 | | | | | |
| Pair_5 | Recipient | PD45799b | AML | Female | 54 | 4.49 | 08/12/2016 | 69 | 14 | Flu-Bu-ATG | RIC | PBSC | 16.28 | Cisplatin received for lung carcinoma 3 years after HSCT |
| Pair_5 | Donor | PD45798b | | Female | 50 | | 08/12/2016 | 64.5 | 14 | | | | | |
| Pair_6 | Recipient | PD45807b | AML | Female | 46 | 2.32 | 26/01/2017 | 67.5 | 21 | Unknown | MAC | BM | Unknown | Cisplatin received for oesophageal cancer after HSCT |
| Pair_6 | Donor | PD45806b | | Female | 43 | | 26/01/2017 | 65.2 | 21 | | | | | |
| Pair_7 | Recipient | PD45795b | NHL | Male | 30 | -5.86 | 08/08/2016 | 59.6 | 29 | Cy-TBI | MAC | BM | 4.17 | |
| Pair_7 | Donor | PD45794b | | Male | 36 | | 08/08/2016 | 65.5 | 29 | | | | | |
| Pair_8 | Recipient | PD45809b | AML | Female | 37 | 1.93 | 08/03/2017 | 67.7 | 31 | Cy-TBI | MAC | BM | 4.05 | |
| Pair_8 | Donor | PD45808b | | Female | 35 | | 08/03/2017 | 65.8 | 31 | | | | | |
| Pair_9 | Recipient | PD45793b | AML | Male | 61 | -4.93 | 27/06/2016 | 69.9 | 9 | BuCy | MAC | BM | 2.66 | |
| Pair_9 | Donor | PD45792b | | Male | 66 | | 27/06/2016 | 74.8 | 9 | | | | | |
| Pair_10 | Recipient | PD45805b | CML | Female | 56 | -6.96 | 23/01/2017 | 73 | 16 | Flu-Bu-ATG | RIC | PBSC | 13.9 | |
| Pair_10 | Donor | PD45804b | | Female | 63 | | 23/01/2017 | 79.9 | 16 | | | | | |

AML, Acute myeloid leukemia; sAML, secondary Acute myeloid leukemia; CML, Chronic myeloid leukemia; HSCT, Allogeneic Haematopoietic stem cell transplantation; MNC, Mononuclear cell; MAC, Myelo-ablative conditioning; RIC, Reduced intensity conditioning; PBSC, Peripheral blood stem cell; BM, Bone marrow; BuCy, Busulfan and Cyclophosphamide; Cy-TBI, Cyclophosphamide and Total body irradition; Flu-Bu-ATG, Flucytosine, Busulfan and Anti-thymocyte globulin.

nature portfolio

Markus Manz


# Reporting Summary


## Statistics

For all statistical analyses, confirm that the following items are present in the figure legend, table legend, main text, or Methods section.

| n/a | Confirmed | |
|---|---|---|
| ☐ | ☒ | The exact sample size (*n*) for each experimental group/condition, given as a discrete number and unit of measurement |
| ☐ | ☒ | A statement on whether measurements were taken from distinct samples or whether the same sample was measured repeatedly |
| ☐ | ☒ | The statistical test(s) used AND whether they are one- or two-sided<br>*Only common tests should be described solely by name; describe more complex techniques in the Methods section.* |
| ☐ | ☒ | A description of all covariates tested |
| ☐ | ☒ | A description of any assumptions or corrections, such as tests of normality and adjustment for multiple comparisons |
| ☐ | ☒ | A full description of the statistical parameters including central tendency (e.g. means) or other basic estimates (e.g. regression coefficient) AND variation (e.g. standard deviation) or associated estimates of uncertainty (e.g. confidence intervals) |
| ☐ | ☒ | For null hypothesis testing, the test statistic (e.g. *F*, *t*, *r*) with confidence intervals, effect sizes, degrees of freedom and *P* value noted<br>*Give P values as exact values whenever suitable.* |
| ☐ | ☒ | For Bayesian analysis, information on the choice of priors and Markov chain Monte Carlo settings |
| ☐ | ☒ | For hierarchical and complex designs, identification of the appropriate level for tests and full reporting of outcomes |
| ☒ | ☐ | Estimates of effect sizes (e.g. Cohen's *d*, Pearson's *r*), indicating how they were calculated |

*Our web collection on statistics for biologists contains articles on many of the points above.*

## Software and code

Policy information about availability of computer code

**Data collection**: None

**Data analysis**:
List of programs and softwares:
• R: version 4.1.1
• BWA-MEM: version 0.7.17 (https://sourceforge.net/projects/bio-bwa/)
• cgpCaVEMan: version 1.11.2/1.13.14/1.14.1 (https://github.com/cancerit/CaVEMan)
• cgpPindel: version 2.2.5/3.2.0/3.3.0 (https://github.com/cancerit/cgpPindel)
• ASCAT NGS: version 4.2.1/4.3.3 (https://github.com/cancerit/ascatNgs)
• VAGrENT: version 3.5.2/3.6.0/3.6.1 (https://github.com/cancerit/VAGrENT)
• GRIDSS: version 2.9.4 (https://github.com/PapenfussLab/gridss)
• MPBoot: version 1.1.0 (https://github.com/diepthihoang/mpboot)
• cgpVAF: version 2.4.0 (https://github.com/cancerit/vafCorrect)
• dNdScv: version 0.0.1.0 (https://github.com/im3sanger/dndscv)
• Rsimpop: version 2.2.6 (https://github.com/NickWilliamsSanger/rsimpop)
Custom code made available (also stated in manuscript): https://github.com/mspencerchapman/Clonal_dynamics_of_HSCT
No commercial software used.

For manuscripts utilizing custom algorithms or software that are central to the research but not yet described in published literature, software must be made available to editors and reviewers. We strongly encourage code deposition in a community repository (e.g. GitHub). See the Nature Portfolio guidelines for submitting code & software for further information.

## Data

Policy information about availability of data

All manuscripts must include a data availability statement. This statement should provide the following information, where applicable:

- Accession codes, unique identifiers, or web links for publicly available datasets
- A description of any restrictions on data availability
- For clinical datasets or third party data, please ensure that the statement adheres to our policy

Whole genomes and targeted sequencing data have been deposited in the European Genome–phenome Archive (EGA) (https://ega-archive.org/). WGS data have been deposited with EGA accession number EGAD00001010872 and targeted sequencing data have been deposited with accession number EGAD00001010874. Larger files of data necessary to reproduce some of the analysis in the github repository are available on Mendeley Data (https://data.mendeley.com/datasets/m7nz2jk8wb/1).

# Field-specific reporting

Please select the one below that is the best fit for your research. If you are not sure, read the appropriate sections before making your selection.

☒ Life sciences ☐ Behavioural & social sciences ☐ Ecological, evolutionary & environmental sciences

For a reference copy of the document with all sections, see nature.com/documents/nr-reporting-summary-flat.pdf

# Life sciences study design

All studies must disclose on these points even when the disclosure is negative.

| | |
|---|---|
| Sample size | We optimised the number of transplant pairs (10 pairs, 20 individuals) and number of haematopoietic stem cells sequenced per individual (average of 170 cells per individual) to describe the transplanted cell numbers, mutation burden, and clonal structure, across a range of transplant variables. No power calculation was performed, and there was no target effect size. This was an observational study. |
| Data exclusions | Genomes with a sequencing depth of less than 4x (46 samples), a VAF distribution showing evidence of non-clonality or contamination (peak VAF < 40%) (468 samples), or with evidence that they were from a different germline (10 samples) were excluded from the analysis. These data exclusions were made to maintain quality of mutation calls and phylogenetic inference. |
| Replication | While the specific donor samples used have been exhausted, the results from this study should be generally reproducible in separate transplant pairs with similar characteristics, using the protocols and code included in this manuscript. |
| Randomization | This is not relevant to our study because it is an observational, descriptive study. Transplant pairs were selected. |
| Blinding | Blinding was not relevant to our study because outcome variables were computationally determined. There was no test performed that required blinding. |

# Reporting for specific materials, systems and methods

We require information from authors about some types of materials, experimental systems and methods used in many studies. Here, indicate whether each material, system or method listed is relevant to your study. If you are not sure if a list item applies to your research, read the appropriate section before selecting a response.

### Materials & experimental systems

| n/a | Involved in the study | |
|---|---|---|
| ☐ | ☒ | Antibodies |
| ☒ | ☐ | Eukaryotic cell lines |
| ☒ | ☐ | Palaeontology and archaeology |
| ☒ | ☐ | Animals and other organisms |
| ☐ | ☒ | Human research participants |
| ☒ | ☐ | Clinical data |
| ☒ | ☐ | Dual use research of concern |

### Methods

| n/a | Involved in the study | |
|---|---|---|
| ☒ | ☐ | ChIP-seq |
| ☐ | ☒ | Flow cytometry |
| ☒ | ☐ | MRI-based neuroimaging |

## Antibodies

| | |
|---|---|
| Antibodies used | PE/Cyanine7 anti-human CD14 BioLegend #301814<br>APC anti-human CD3 BioLegend #317318<br>FITC anti-human CD19 BioLegend #363008 |

| | |
|---|---|
| Validation | These were all previously validated commercially available antibodies.<br>CD3 FITC: Validated by supplier with the following notes - species reactivity: human; application - flow cytometry<br>CD19 A700: Validated by the supplier with the following notes - species reactivity: human, chimpanzee, rhesus; application: flow cytometry |

## Human research participants

Policy information about studies involving human research participants

| | |
|---|---|
| Population characteristics | The dataset comprised 3,399 whole genomes from the blood of 10 fully HLA-matched sibling donor and recipient pairs (20 individuals) who had been recruited for a previous study. In each case, the recipient had undergone HCT many years prior to sampling (range: 9-31 years) and had complete or almost complete replacement of their haematopoietic system with that of the donor. The most common indication for HCT was acute myeloid leukaemia; the conditioning regimen was myelo-ablative (n=7) or reduced intensity (n=3); the stem cell source was bone marrow (n=5) or mobilised peripheral blood (n=5); and recipients were of similar age to their sibling donors (age difference: -7 to +11 years). The youngest individual was 34 years old at the time of sampling; the oldest was 79 years old. There were 13 females and 7 males in the study. |
| Recruitment | Individuals were recruited in Zurich, Switzerland, and ethical approval was by the local ethics board (Kantonale Ethikkommission - Zurich).<br>Donor and recipient pairs were recruited if both were alive at least 10 years after transplant. This biases for transplant procedures that have resulted in long-term disease remission for the recipient. This may bias against recipients with disease at high risk of relapse, or high transplant-related mortality. |
| Ethics oversight | Kantonale Ethikkommission - Zurich (KEK-ZH No. 2015-0053 & 2019-02290) |

Note that full information on the approval of the study protocol must also be provided in the manuscript.

## Flow Cytometry

### Plots

Confirm that:

☒ The axis labels state the marker and fluorochrome used (e.g. CD4-FITC).

☒ The axis scales are clearly visible. Include numbers along axes only for bottom left plot of group (a 'group' is an analysis of identical markers).

☒ All plots are contour plots with outliers or pseudocolor plots.

☒ A numerical value for number of cells or percentage (with statistics) is provided.

### Methodology

| | |
|---|---|
| Sample preparation | Granulocytes were isolated from 10ml of EDTA anticoagulated peripheral blood using EasySep Direct Neutrophil Isolation Kit (StemCellTechnologies, Vancouver, Canada) according to the manufacturer's instructions. CD34+ HSPCs were isolated from 20ml of EDTA anticoagulated peripheral blood using human CD34 MicroBead Kit (Miltenyi Biotec, Bergisch Gladbach, Germany) following the manufacturer's recommendations. B cells, T cells and monocytes were flow-sorted from CD34- cell fractions using a FACSAria III flow cytometer (BD Biosciences, San Jose, USA). |
| Instrument | FACSAria II flow cytometer (BD Biosciences, San Jose, USA). |
| Software | No analysis of flow cytometry data is presented in this manuscript. FlowJo v10 (BD Biosciences, San Jose, USA) was used to generate the gating strategy image. |
| Cell population abundance | Sorting of CD3+ T cells, CD19+ B cells and CD14+ monocytes was performed on a FACSAria III flow cytometer (BD Biosciences, San Jose, USA) with resulting post-sort cell populations of >90% purity as determined by flow cytometry on the same instrument. |
| Gating strategy | After gating for cellular events in the FSC-A and SSC-A and doublet exclusion in the FSC-A and FSC-H plots; CD3, CD19 and CD14 were used to gate for T cells, B cells and monocytes, respectively. |

☒ Tick this box to confirm that a figure exemplifying the gating strategy is provided in the Supplementary Information.