## [Peer Review File · Nature]

Manuscript Title: Clonal dynamics after allogeneic haematopoietic cell transplantation

Reviewer Comments & Author Rebuttals

Reviewer Reports on the Initial Version:

Referees' comments:

Referee #1 (Remarks to the Author):

This is a paper bringing tools for analysis of clonal phylogenies from somatic mutations to the problem of understanding dynamics of allogeneic haematopoietic cell transplantation (HCT) and explaining some findings of clinical importance, such as loss of diversity in recipients relative to donors. It is a great question and the paper makes a valuable contribution in bringing these kinds of analysis methods to HCT. The use of sequencing here appears nicely done and there is reasonable handling of anticipated technical artifacts. The sample sizes in terms of numbers of cells are very good. While the number of research subjects may be too low to be powered for some purposes, it is understandable given the technical challenges. The work makes effective use of sophisticated computational tools from phylogenetic inference and related ideas such as mutation signature analysis. There are a number of intriguing conclusions, most notably in teasing apart the interplay of “pruning selection” relating to clonal expansions at the time of transplant versus “growth selection” occurring post-transplant and quantifying a concept of “phylogenetic” age with clinical implications. While the paper focuses on the basic research findings, there would seem to be at least translational potential for using these findings to better serve HCT patients. Overall, I think it is a sound and scientifically innovative study that is likely to be of broad interest. There are, however, a few points where I felt I could use more clarity and perhaps statistical rigor, mostly on establishing that the interpretation of the data in terms of clonal selection is compelling and that one can rule out alternative hypotheses:

1. I have a few questions on the analysis of selection for expanding clones. The figures tell an interesting story but I am still unsure what can be asserted with statistical confidence. It is clear that some lineages expand and others presumably are lost, but I was not fully convinced on the assertion that this is explained by selection and cannot be explained by drift. One would expect to see lineages expand and contract even from a uniform branching process and one might expect to see this drift more dramatically post-transplant just because of the relative population bottlenecks in recipient versus donor. The authors commendably consider these two alternative hypotheses and assert that one needs selection to account for the data, and the figures do seem to point that way (e.g., Ext. Fig. 12), but I was unclear what the actual statistical evidence for this was to assert the conclusion.
2. As a more minor technical follow-up to the previous question, what was the motivation in Simulation model 3 in having time-limited selection following engraftment? Is there a specific proposed mechanism behind this that the authors had in mind here to explore?

3. Assuming we accept that this the effects here show selection, I also was not convinced by the argument that it is positive versus negative selection. The paper here seems to go on the assumption that the system works similarly to what we expect in clonal lineages cancers: that the story is ultimately explained by the interplay of neutral evolution with some clones acquiring driver mutations that lead to positive selection. This is not a cancer system, though, and the “normal” evolutionary rule --- that most mutations are deleterious --- would seem to apply. One might then hypothesize that selection would mostly be driven by deleterious mutations one normally would not see in cancers, rather than the driver genes on which the paper focuses. If the results cannot be explained by neutral drift, is it clear that they are explained by acquisition of positive selection in some clones versus acquisition of negative selection by some clones? Is the analysis here ruling out one or the other hypothesis? Or does it not matter to the story what the answer is?

4. Related to the above points, I would like to see a more explicit statement of exactly how the authors are defining drivers or likely drivers. Are the genes identified as likely drivers those known to be active or to drive positive selection in cancers? In leukemias specifically? I ask because the system again is not quite the same as clonal expansion in cancers and I would think that depending on the driver mechanism, we might expect some cancer drivers to be drivers here and others not to be. I think the paper could benefit from a bit more discussion and analysis to explain why we would expect the specific genes flagged as drivers here to be drivers in HCT and if they are behaving as we would expect.

5. I also wanted to request a bit more clarity on the timing of coalescence with respect to HCT. The paper emphasizes the distinction between pruning selection (preference at the time of transplant) and growth selection (selection following transplant). I am not fully convinced that the approximate Bayesian computation (ABC) procedure could not be inadvertently introducing any biases in timings of coalescences that could skew the answer to this question. The major point of ABC is to get an efficiency boost in sampling by collapsing states that are distinct but have the same summary statistics. This is fine if those summary statistics more or less distinguish all of the trajectories that are meaningfully different by the properties we care about. If those collapsed subsets of states are skewed with respect to features missing from the summary statistics, though, that could introduce a skew in the statistics of the final sample. I am not asserting there is definitely a problem, but I would appreciate a bit more convincing that the summary statistics are sufficient to separate trajectories with different coalescence timings in ways that might affect the final statistical analysis of those timings in the paper. For example, might the use of summary statistics grouping trajectories into epochs around HCT still hide a skew within these epochs that could bias the final statistics?

6. The use of a maximum parsimony algorithm is unusual for this domain, although the authors do give citations to justify it. Maximum likelihood is generally viewed as more robust to different assumptions about the tree. And there are now many specialized phylogeny methods specifically for cell lineage analysis and similar sorts of sequence data that the authors might have used and that at first glance would appear more appropriate than a generic phylogeny model like maximum parsimony. Was the choice just driven by a need for efficiency?

7. This is a small and optional thing, but I thought the APOBEC observation was very interesting and was curious if anything more could be said about it. Is there any insight to offer into evolutionary

dynamics here that might at least give suggestions about mechanism? For example, is there a loss of selection against clones with mutant APOBEC signatures post-transplant, that might suggest that there is some mechanism of clearing these clones that is active in donor but missing from recipients? Alternatively, is there a rise in appearance of APOBEC mutants post-transplant that might suggest a distinct source of mutability in recipients not active in donors? I realize, though, this is a tangent and perhaps a story to explore more fully in another paper.

8. One minor error is that for Fig 4a, the figure identifies it as about Pair 3 but the caption says Pair 4.

Referee #2 (Remarks to the Author):

The manuscript by Chapman et al. describes a long-term follow-up cohort of 10 donor and recipient pairs after allogeneic hematopoietic stem cell transplantation in which the authors have used whole genome sequencing of individually grown CD34+ clones to assess interindividual and interclonal differences in mutational load in the hematopoietic system of donor and recipient. The authors use complex statistical modeling to infer longitudinal dynamics of hematopoietic clones from this data, including selection processes preceding and following the transplantation procedure.

The methodology and research questions of this study are similar to other recently published works by the group and others (Mitchell et al., Nature 2022; Fabre et al., Nature 2022 and Williams et al., Nature 2022) which have all demonstrated the life-long selection of hematopoietic clones that can lead to premalignant and malignant transformation.

The strength of this work is that by comparing the same hematopoietic system in its native (donor) and its transplanted (recipient) form, the authors are able to demonstrate additional selective pressures induced by the transplantation process, which would have been predicted in this form by most researchers in the field.

On the other hand, in its present form, the manuscript represents an extension in the transplant context of their previous studies. I think that at this point they need to complement their well-oiled computational pipelines to infer clonal relationships with some functional data/mechanistic insides. For example, they observed an APOBEC signature only in recipients, which would be interesting to dive in, or, alternatively, another interesting avenue could be to understand why CHEK2 mutations seem to favor engraftment and/or long term survival. As it is now, this paper is a well executed exercise in style which does not provide many potentially actionable new insights.

- as a side criticism, they mention in the introduction genotoxicity from antiviral drugs as a factor potentially influencing HSC dynamics, but do not test it in their cohort, which would be highly relevant for transplant recipients

The study is otherwise well-executed and the manuscript is written in a concise and well-structured manner. Other minor comments / queries:

- The recurrent loss of chromosome Y in pair 9 is suggestive of a specific selective pressure. Can the authors comment on what they believe the mechanisms behind this observation may be?
- As B cells and T cells have additional clonal markers in the form of BCR/TCR, did the authors attempt to correlate clonal diversity measured by somatic mutations and BCR/TCR in B/T cells?
- Were there any cases with evidence of residual recipient-derived hematopoiesis?

Referee #3 (Remarks to the Author):

The authors characterize the historical dynamics of the hematopoietic system within ten allogeneic HSCT recipient-donor sibling pairs. They employ two central laboratory-based methods: whole-genome sequencing of ~100-200 distinct peripheral-blood-derived colony-forming units per individual, and targeted bulk sequencing of lineage-sorted cells for mutations identified in the WGS assay. Using somatic variant calls (SNVs and SVs) from standard workflows on the WGS data, the authors create HSPC phylogenies based on adjusted mutational burden that can be aligned to the chronological age of the individuals, with care taken to remove non-clonal samples, duplicated colonies, and contaminants prior to analysis. The WGS data was further used to estimate clonal diversity and contribution of common trinucleotide mutational patterns to the observed mutation landscape. The data from the targeted sequencing was used to directly estimate the clonal contribution to each of the studied lineages, and was also overlaid back onto the WGS-derived phylogenies.

Based on these analyses, the authors make the following main conclusions. First, they estimate that between 500 to 100,000 HSPCs engrafted in the recipients, with 10-fold fewer engrafted cells in older donors (more than 50 years old) than with younger donors. Second, they found that there was lower clonal diversity among recipients, corresponding to a 10-15 year greater estimated phylogenetic age. Third, they report that HSPC clones in the recipients frequently demonstrate lineage-specific biases. Fourth, in what may be considered the topline conclusion of the manuscript, the authors describe patterns of pre-transplant and peri-transplant clonal expansion, which they term “pruning” and “growth” selection, which are apparent consequences of a clone’s selective advantage in the setting of donation itself (mobilization/collection/etc.) or once placed within the body of the recipient, respectively. Fifth, the authors show that clones with the same driver mutations differ in their representation when comparing the donor and recipient. And, while somewhat tangential to the main story of clonal dynamics, the results also demonstrate APOBEC mutational signatures in recipients but not donors, suggesting an activation of a mutational process not usually found in HSPCs.

These experiments require complex analysis but the authors convey their message with thoughtful precision, as a result, the significant implications of this work are readily appreciated. Foremost, it is convincingly demonstrated that certain pre-existing clones within the donor may have a selective advantage during the donation process, and that this pruning selection may occur with similar

frequency to growth selection, which will undoubtedly spur research into how to identify these clones a priori and into the mechanisms of why they have an advantage during donation. Though this has been speculated for decades, this is the first experimental rigor illustrating the phenomenon. The research also raises the provocative question of whether there are meaningful correlations to clinical outcomes from pruning vs growth selection. Furthermore, while previous studies have also sought to quantify the number of engrafted HSPCs and loss of clonality, this manuscript provides the most precise and robust figures to date, we imagine that it will likely be well-received by researchers in the fields of HSC biology, hematologic malignancy, and clonal dynamics, among others.

Major Points:

1. Understandably from a logistical perspective, the studied participants were chosen for likelihood of available sample from long intervals (~10y). However, the natural bias then is against unsuccessful transplants: pairs in which engraftment failed in the recipient, transplants in which the recipient developed subsequent hematological malignancy, and/or died (whether or not the graft contributed) were excluded. This unfortunately limits the generalizability of the conclusions to only successful allogeneic transplants and dampens the translational potential. Some of the major clinical objectives in HSCT care have been to prevent graft failure, donor-derived malignancy, and reduce mortality, yet this study cannot speak to whether/how the biological phenomena that are very elegantly described (such as “pruning” vs “growth” selection) integrate into the aforementioned areas of unmet clinical need. The authors should tackle this in the discussion more completely.
2. Further, presumably the clones in the matched post-transplant sample are all donor derived (see below minor point #2)? Can the authors clearly articulate this case?
3. Several points of the manuscript rely on the assumption that the monocyte, B cell, and T cell populations were accurately ascertained and each lineage isolated into a pure fraction. However, no antibody information, gating strategies, nor FACS data are presented to allow for the verification of these claims. These data should be provided and the reporting summary updated.
4. Peripheral-blood-derived HSPCs may present a biased view of the overall stem and progenitor compartment. This might be particularly true if there is clonal hematopoiesis in epigenetic regulation genes, as expression of CXCR4, which plays a pivotal role in HSPC homing to/retention in the bone marrow niche, is highly sensitive to methylation (e.g., PMIDs 27899645 and 31018749). Indeed, there are changes in CXCR4 expression across the hematopoietic lineages in the context of mutations in DNMT3A and TET2 (e.g., PMIDs 30890702 and 33155517). If certain clones have more (or less) CXCR4 expressed by their HSPCs, then they might be under- (over-) represented in the peripheral blood HSPCs as compared to the marrow HSPCs. This does not contradict the findings of the authors, but it does mean that their findings should not necessarily be interpreted as representing the clonal dynamics of the entire HSPC pool without proper examination of marrow-derived samples.

Minor Points:

1. Consider putting age brackets for younger and older donors in the abstract, as this information is highly pertinent to the result finding 10-fold fewer engrafted HSCs with older donors.
2. What is the percent engraftment for each recipient? Shall it be assumed that the cells taken from a recipient 10y after a successful transplant are 100% donor-derived? What do the authors think about low fraction persistent recipient chimeras and how do they account for this in their analysis?
3. Were CD34+ cells quantified prior to plating in methylcellulose? Description of the colony-forming units per input for each donor-recipient pair would speak to the similarities or differences of the current stem-like capacity within each individual's hematopoietic compartment. This would be of great interest, especially given how the authors demonstrate that HSCT advances the apparent mutational age of recipient HSPCs by more than a decade, on average.
4. Does the APOBEC mutational signature occur stochastically or in a biased manner? If biased, is it preferentially found in (or exclusive of) the dominant clonal populations? If in larger clones, is it in "pruning" or "growth" selection clones?
5. Page 22, Line 510: should be J. Clin. Oncol.
6. Page 30, Line 794: Were these 10 genes the only ones considered to have "driver mutations" as the term is used in the body of the manuscript? This does not appear to be the case, given several of the figures list driver mutations not listed here (e.g., TP53 mutations). If not, please provide a list of which genes would be considered driver genes (including any that may not have been observed in the study, e.g., JAK2).
7. Reporting summary, page 2: mentions optimizing sample size to describe, among other things, telomere lengths. No data on telomere lengths is presented in the manuscript.

Author Rebuttals to Initial Comments:

Below, the reviewer comments are in blue, with our response in black and actions that we have undertaken for the revision in red and bold.

Referee #1 (Remarks to the Author):

This is a paper bringing tools for analysis of clonal phylogenies from somatic mutations to the problem of understanding dynamics of allogeneic haematopoietic cell transplantation (HCT) and explaining some findings of clinical importance, such as loss of diversity in recipients relative to donors. It is a great question and the paper makes a valuable contribution in bringing these kinds of analysis methods to HCT. The use of sequencing here appears nicely done and there is reasonable handling of anticipated technical artifacts. The sample sizes in terms of numbers of cells are very good. While the number of research subjects may be too low to be powered for some purposes, it is understandable given the technical challenges. The work makes effective use of sophisticated computational tools from phylogenetic inference and related ideas such as mutation signature analysis. There are a number of intriguing conclusions, most notably in teasing apart the interplay of “pruning selection” relating to clonal expansions at the time of transplant versus “growth selection” occurring post-transplant and quantifying a concept of “phylogenetic” age with clinical implications. While the paper focuses on the basic research findings, there would seem to be at least translational potential for using these findings to better serve HCT patients. Overall, I think it is a sound and scientifically innovative study that is likely to be of broad interest.

We thank the reviewer for these generous comments.

There are, however, a few points where I felt I could use more clarity and perhaps statistical rigor, mostly on establishing that the interpretation of the data in terms of clonal selection is compelling and that one can rule out alternative hypotheses:

1. I have a few questions on the analysis of selection for expanding clones. The figures tell an interesting story but I am still unsure what can be asserted with statistical confidence. It is clear that some lineages expand and others presumably are lost, but I was not fully convinced on the assertion

that this is explained by selection and cannot be explained by drift. One would expect to see lineages expand and contract even from a uniform branching process and one might expect to see this drift more dramatically post-transplant just because of the relative population bottlenecks in recipient versus donor. The authors commendably consider these two alternative hypotheses and assert that one needs selection to account for the data, and the figures do seem to point that way (e.g., Ext. Fig. 12), but I was unclear what the actual statistical evidence for this was to assert the conclusion.

We agree with the reviewer that our arguments about selection causing the observed patterns of clonal expansion in the donor and recipient would benefit from more statistical rigour. To this end, we have now performed posterior predictive checks on three alternative models of selection by extending the Approximate Bayesian Computation framework we developed for the first submission. This approach essentially estimates the posterior p value that the observed summary statistics (or more extreme) could arise from the given model under its optimal parameter estimates. In this approach we use the output from the ABC method to estimate a posterior predictive p-value – this is based on a generalized version of the chi-squared statistic (the chi-squared discrepancy variable) which combines information from all the summary statistics¹. The posterior predictive p-value can be interpreted as a Bayesian average of classical p-values, which takes account of the uncertainty about the values of the model parameters, as represented by the posterior distribution. Like a classical frequentist p-value, the posterior predictive p-value informs us about the degree to which the observed data (as represented by our summary statistics) is unsurprising, or surprising (in some way extreme, or an outlier) under the specified model.

The three models we compared were –

1. *Null model of age-related selection with bottleneck* – based on the parameters for the extent of positive selection in HSPCs with normal ageing estimated in our earlier work², we added the bottleneck associated with stem cell transplantation in the recipient into the simulation framework.
2. *Model of age-related selection and bottleneck plus pruning selection* – This used the same core framework as the null model, but included an additional opportunity for pruning selection. Essentially, the transplanted stem cells had an additional parameter (drawn from a gamma distribution) representing their relative increase in probability of surviving the transplant procedure to engraftment.
3. *Model of age-related selection and bottleneck plus growth selection* – This used the same core framework as the null model, but included an additional opportunity for growth selection. Essentially, transplanted stem cells had an additional parameter for their fitness to expand after engraftment.

QQ plots of the Bayesian p values arising from the posterior predictive checks for each model are shown in **Reviewer Figure 1** below. The null model generates generally poor fits to the observed summary statistics with three donor-recipient pairs having p values below 0.05 and three further between 0.05 and 0.06. Notably, the only two pairs where the null model provided good fit were the two youngest pairs (Pairs 1 and 2), where no substantial clonal expansions were observed. In the case of the two other pairs (Pairs 7 and 8) where the null model provided an adequate fit (as indicated by the moderately high p-values), the donors were also both relatively young at the time of the transplant.

The models including either pruning selection or growth selection moved the QQ plots closer to the expected line, suggesting they generally improved the fit to the observed data. However, the QQ plots still suggested that one or other remained insufficient to fully explain the data for some cases, especially pairs 4, 6 and 10. It is possible that both types of selection would be required to explain the observed data for these pairs (we would have liked to develop a model combining both forms of selection, but the computational burden would be prohibitive – the reason this resubmission has taken us 11 months to complete is because of the computational complexity in generating the three models reported here).

Thus, in summary, the null model of typical age-related selection in HSPCs combined with a transplant-related bottleneck in the recipient is insufficient to explain the observed phylogenetic trees in the majority of donor-recipient pairs, especially with donors who were older at the time of harvesting. Models incorporating either pruning or growth selection do improve the fit for some donor-recipient pairs but do not fully explain all pairs. **We have included the information on posterior predictive checks in the manuscript ('Pruning' selection versus 'growth' selection'; pp. 13-14), incorporated Reviewer Figure 1 as Extended Figure 14d and detailed the specifics of the models in the Methods ('Simulation frameworks'; pp. 34-36).**

Reviewer Fig. 1. QQ plots of posterior p values for the 10 donor-recipient pairs across different models of transplant-related selection. In each panel, the posterior p values are ranked (x axis; quantile) and the posterior p value is shown (y axis), coloured by donor-recipient pair. The blue lines represent $x=y$ and the grey lines represent $y=0.05$. (i) Model of age-related selection combined with a bottleneck for transplant into recipient; (ii) model of age-related selection, bottleneck plus pruning selection; (iii) model of age-related selection, bottleneck plus growth selection.

2. As a more minor technical follow-up to the previous question, what was the motivation in Simulation model 3 in having time-limited selection following engraftment? Is there a specific proposed mechanism behind this that the authors had in mind here to explore?

The motivation here is that the immediate post-transplant environment is unusual for several reasons: there is profound pancytopenia and the recipient bone marrow is hypoplastic following conditioning chemotherapy; the marrow microenvironment has recently been affected by leukaemia and intensive chemotherapy that may alter the selective landscape; there are frequently multiple infective or inflammatory episodes during the first few years after transplant as the innate and adaptive immune

systems reconstitute; there is often residual host immunity that wanes over time. All of these factors are most pronounced in the early post-transplant period and are likely to resolve, at least partially, with time. The choice of 5 years is empirical, but the motivation of this analysis is to demonstrate that the concept of post-HCT growth selection produces phylogenies that are more consistent with the data.

We agree that this was not adequately explained in the original submission – **we have therefore explained the motivation for the time-limited selection in the manuscript (Methods; ‘Simulation frameworks’; pp. 34-36).**

3. Assuming we accept that this the effects here show selection, I also was not convinced by the argument that it is positive versus negative selection. The paper here seems to go on the assumption that the system works similarly to what we expect in clonal lineages cancers: that the story is ultimately explained by the interplay of neutral evolution with some clones acquiring driver mutations that lead to positive selection. This is not a cancer system, though, and the “normal” evolutionary rule --- that most mutations are deleterious --- would seem to apply. One might then hypothesize that selection would mostly be driven by deleterious mutations one normally would not see in cancers, rather than the driver genes on which the paper focuses. If the results cannot be explained by neutral drift, is it clear that they are explained by acquisition of positive selection in some clones versus acquisition of negative selection by some clones? Is the analysis here ruling out one or the other hypothesis? Or does it not matter to the story what the answer is?

This is an interesting question. We can use the ratio of non-synonymous to synonymous mutations to distinguish whether positive or negative selection is the dominant mode shaping a population’s clonal dynamics³⁻⁵. In this approach, synonymous mutations are assumed to be selectively neutral because they do not change the amino acid sequence of the protein. Notwithstanding the occasional instances where synonymous mutations alter splicing or mRNA stability⁶, the vast majority do appear to evolve neutrally and thus provide an accurate estimate of the background mutation rate in that sample. From the number of synonymous mutations, the expected number of non-synonymous mutations in the absence of selection can be inferred. Then, if the observed number of non-synonymous mutations is lower than this expectation, there has been a net loss of non-synonymous variants, implying an excess of negative over positive selection. In contrast, observing a greater-than-expected number of non-synonymous mutations implies net positive selection. Previously, we developed an algorithm based on these principles, fine-tuned for somatic mutations^{3,7}, which corrects for the influences of mutational signatures, replication timing and epigenetic activity on mutation rates across the genome. The algorithm estimates the dN/dS ratio, with a value of 1 implying a balance of positive and negative selection (or neutrality), a value >1 implying net excess of non-synonymous mutations (and therefore predominance of positive selection) and a value <1 implying net depletion of non-synonymous mutations (more negative than positive selection).

We applied this algorithm to calculate a global dN/dS ratio across all coding regions in the genome for the allogeneic stem cell transplant data reported here. The estimated global dN/dS ratio was 1.09 (CI_{95%} = 1.06-1.13), suggesting net positive selection (**Reviewer Fig. 2**). This estimate is very similar to what we observed in the haematopoietic compartment of healthy individuals from across the lifespan². If transplant caused an increase in negative selection among engrafting clones, we would

expect dN/dS values to be lower in recipients compared to donors. However, we see similar values, with a trend towards higher values in recipients (**Reviewer Fig. 3**).

These dN/dS values suggest that positive selection is numerically more important in shaping clonal dynamics than negative selection. However, it does not preclude that negative selection is at play for some clones. The phylogeny also does not help with this question. Negative selection at the time of transplant is almost impossible to distinguish from a tighter transplant bottleneck. Clones harbouring negatively selected mutations would fail to engraft or expand, leaving the clones without such variants to repopulate the marrow – in this situation, the recipient phylogeny would show features of a more significant bottleneck, but the clones would be balanced, showing features similar to a neutral bottleneck (**Extended Fig. 7**). The phylogeny is therefore not a sensitive way to detect negative selection during HCT.

In conclusion, our data reveals evidence of a predominance of positive selection over negative selection. However, we cannot state that negative selection is entirely absent. **We have discussed these points in the manuscript ('Dynamics of driver mutations through HCT'; p. 16) and included Reviewer Figure 3 in the Extended Data (Extended Figure 15d).**

Reviewer Fig. 2. dN/dS ratio for the full combined mutation set.

Reviewer Fig. 3. dN/dS ratio combined across pairs, but split by mutations found in donors or recipients.

4. Related to the above points, I would like to see a more explicit statement of exactly how the authors are defining drivers or likely drivers. Are the genes identified as likely drivers those known to be active or to drive positive selection in cancers? In leukemias specifically? I ask because the system again is not quite the same as clonal expansion in cancers and I would think that depending on the driver mechanism, we might expect some cancer drivers to be drivers here and others not to be. I think the paper could benefit from a bit more discussion and analysis to explain why we would expect the specific genes flagged as drivers here to be drivers in HCT and if they are behaving as we would expect.

When considering drivers, we have included only genes that have been recognised as drivers of haematological malignancies (primarily those associated with myeloid neoplasms) or ageing (namely those associated with clonal haematopoiesis). Most of the lifetime of the HSC clones has been under the conditions of 'normal ageing' within the donor and will have been subject to the usual selective pressures of the general population.

We agree that we did not provide sufficient information on how we classified mutations as drivers. **We have therefore added a section to the Materials and Methods detailing the approach for annotating driver mutations ('Driver mutation annotation', p. 31), and added two new Supplementary Tables 1-2) that include the full gene list and the raw list of variants in those genes that were subsequently manually curated. We have also briefly explained in the main text what we are defining as drivers in the analysis ('Dynamics of driver mutations through HCT'; p. 16).**

We very much agree with the reviewer that genes conferring a selective advantage in HCT are likely to have only partial overlap with those seen in normal ageing and blood cancers. There are a number of transplant-specific selective pressures involved in mobilising, transfusing and engrafting stem cells from donor to recipient, and there would be little *a priori* reason to believe that the set of genes enabling cells to survive these pressures would be identical to those seen in unperturbed haematopoiesis. In fact, the distinct patterns of selection we observe in recipients compared to donors (now supported by the more formal Bayesian analysis discussed in our response to point 1) supports the reviewer's point.

Furthermore, the global dN/dS values discussed in our response to point 3 suggest that there are many driver mutations among recipient clones that we have not been able to formally identify. To quantify this, a global dN/dS ratio of 1.09 implies about 1 in 11 non-synonymous mutations are drivers³ – thus, across a total of 7809 such variants in our recipient colonies, we can infer that at least 775 (CI_{95%} 454 – 1081) are under positive selection. Using the known blood cancer and clonal haematopoiesis genes described in the previous paragraph, we only identify 70 as drivers with high confidence. The implication is that there are many driver genes remaining to be discovered that could confer transplant-specific selective advantage. Our only means to identify these will be to sequence sufficiently large sample sizes of clones from recipients, but such genes may indeed provide interesting insights into the nature of the selective pressures experienced by a transplanted haematopoietic stem cell. **We have discussed these inferences and their implications in the manuscript ('Dynamics of driver mutations through HCT'; p. 16).**

5. I also wanted to request a bit more clarity on the timing of coalescence with respect to HCT. The paper emphasizes the distinction between pruning selection (preference at the time of transplant) and growth selection (selection following transplant). I am not fully convinced that the approximate Bayesian computation (ABC) procedure could not be inadvertently introducing any biases in timings of coalescences that could skew the answer to this question. The major point of ABC is to get an efficiency boost in sampling by collapsing states that are distinct but have the same summary statistics. This is fine if those summary statistics more or less distinguish all of the trajectories that are meaningfully different by the properties we care about. If those collapsed subsets of states are skewed with respect to features missing from the summary statistics, though, that could introduce a skew in the statistics of the final sample. I am not asserting there is definitely a problem, but I would appreciate a bit more convincing that the summary statistics are sufficient to separate trajectories with different coalescence timings in ways that might affect the final statistical analysis of those timings in the paper. For example, might the use of summary statistics grouping trajectories into epochs around HCT still hide a skew within these epochs that could bias the final statistics?

We thank the reviewer for this thoughtful point. It is true that the timing of coalescences relative to the defined 'transplant epoch' in the phylogeny has a key role in informing the ABC. It is also the case that the timing of the coalescences is subject to some random variation in that mutations are acquired at a fairly constant rate, but the absolute numbers acquired in a given time period are subject to at least Poisson variation.

First, we assessed whether this variation leads to significant uncertainty in the numbers of coalescences in each epoch. We used a bootstrapping approach where all branch lengths were redrawn from a negative binomial distribution with μ equal to the original number of mutations, and the Θ overdispersion parameter estimated from distribution of HSPC mutation burdens in that pair (100 bootstraps performed for each pair). We then repeated the steps of making the tree ultrametric and scaling to time, and calculated the number of coalescences falling in each epoch used in the ABC. This demonstrated that the numbers are robust, with only subtle variation in some values where coalescences fall close to the borders between epochs (**Reviewer Fig. 4**).

Reviewer Fig. 4. Robustness of the coalescence-timing summary statistics to bootstrapping. For each pair, the four coalescence-timing summary statistics for the data are shown as red crosses. Those for the 100 bootstraps are shown as “box-and-whisker” plots with the raw data shown as black circles with some jittering.

Second, we assessed whether varying the specific definitions of the epochs used for summary statistics meaningfully altered the posterior distributions of the ABC. Specifically, we assessed four alternative sets of epochs – (1) dividing the pre-transplant interval into more epochs; (2) dividing the peri-transplant interval into more epochs; (3) using a narrower range of molecular time for the peri-transplant interval; and (4) using a wider range of molecular time for the peri-transplant interval. Reassuringly, across the different ABC models and parameters, the different donor-recipient pairs and the different methods for estimating the posterior, we see that the four alternative definitions of HCT epochs have minimal effect on the inferred posterior distributions. This is illustrated in **Reviewer Fig. 5** for one model parameter in one of the donor-recipient pairs; others show the same pattern.

We have added these supportive analyses to the manuscript (Materials and Methods, ‘Approximate Bayesian Computation of engrafting cell number’; pp. 36-39). Reviewer Fig. 4 has been included as Supplementary Fig. 2 and Reviewer Fig. 5 as Supplementary Fig. 3.

Reviewer Fig. 5. Robustness of the posterior distributions to different definitions of epochs used in the summary statistics. The data shown are for model 3 and the parameter estimated is the size of the HSCT bottleneck; other parameters in other models showed similar patterns. Each facet plot shows the prior (blue) and estimated posterior (red) distributions for size of the HSCT bottleneck for one of the donor-recipient pairs (Pair 7). From left to right, the facets are organised by different regression methods for estimating the posterior parameter values within the ABC. From top to bottom, the facets are organised by the different definitions of epochs (original method on top followed by the four different definitions described above).

6. The use of a maximum parsimony algorithm is unusual for this domain, although the authors do give citations to justify it. Maximum likelihood is generally viewed as more robust to different assumptions about the tree. And there are now many specialized phylogeny methods specifically for cell lineage analysis and similar sorts of sequence data that the authors might have used and that at first glance would appear more appropriate than a generic phylogeny model like maximum parsimony. Was the choice just driven by a need for efficiency?

This point does indeed merit further justification, which we have discussed below. Before getting to this point, for further reassurance we have performed phylogeny inference for all trees with the maximum-likelihood algorithm 'IQtree' (<http://www.iqtree.org/>) and compared the resulting

phylogenies to those from MPBoot. These showed extremely similar structures in all cases as shown by high Robinson-Foulds and Quartet similarity scores (**Reviewer Table 1**). In almost all cases the differences were in the orientation of early developmental splits that would have no bearing on the downstream analysis (**Reviewer Fig. 6**).

Reviewer Fig. 6. Comparison of five phylogenies generated by MPBoot with those generated by IQtree. These are the raw phylogenies without branch length correction or removal of duplicates. Short branches with <10 mutations assigned have been extended to a length of 10 to facilitate visualisation of differences. Branches present in one tree but absent in the other are highlighted in red.

	Robinson-Foulds similarity	Quartet similarity
Pair_1	0.984	1.000
Pair_2	0.978	1.000
Pair_3	0.959	0.994
Pair_4	0.993	1.000
Pair_5	0.985	0.903
Pair_6	0.963	0.980
Pair_7	0.955	0.998
Pair_8	0.963	0.992
Pair_9	0.974	0.993
Pair_10	0.993	1.000

Reviewer Table 1. Similarity of the phylogenies generated by MPBoot and IQtree using the Robinson-Foulds similarity and Quartet similarity metrics.

Justification for use of maximum parsimony phylogeny inference algorithm

Many different algorithms have been developed to reconstruct phylogenetic trees based on DNA sequences. These character-based algorithms rely on different approaches: maximum parsimony, maximum likelihood, or Bayesian inference⁸. Maximum parsimony-based algorithms seek to produce a phylogeny that requires the fewest discrete changes on the tree. Because the number of nucleotide changes is minimised, this approach implicitly assumes that mutations are likely to occur only once. Hence, maximum parsimony may produce erroneous phylogenies when there is a high likelihood of recurrent or reversal mutations, such as with long divergence times or high mutation rates, neither of which generally apply to mutations in normal somatic cells. Phylogenetic tree algorithms relying on maximum likelihood or Bayesian inference are model-based, in that they require a specific estimation of the parameters governing genetic sequence evolution to calculate either distances or likelihoods. Often, this involves a general time-reversible model of sequence evolution⁹. The maximum likelihood approach will attempt to identify the phylogeny with the highest log-likelihood of the data given the underlying model of sequence evolution, while Bayesian inference methods will seek the posterior distribution given the priors and observed data. All these approaches have been widely applied to the reconstruction of phylogenetic trees between species or individuals⁸. However, the task of constructing a phylogeny of somatic cells derived from a single individual is fundamentally different from reconstructing species trees in three ways: (1) precise knowledge of the ancestral state; (2) the inequality of forward and reversion rates of somatic mutations; and (3) the low number of mutations compared to the size of the genome. We can expand on these three points:

- (1) *Precise knowledge of the ancestral state* – In contrast to the unknown ancestral genetic state in alignments of sequences from multiple species, the ancestral DNA sequence at the root of the tree (namely the zygote) can readily be inferred from the data. Since all cells in the body are derived from the fertilised egg, any post-zygotic mutation will be present in only a subset

of the leaves of the tree. Hence, the genetic sequence at the root of the tree is defined by the absence of all of these mutations. This simple observation effectively roots the phylogeny.

- (2) *Unequal rates of somatic mutation versus reversion* – In order to accommodate the uncertainty in the ancestral state and the direction of nucleotide substitutions, model-based phylogeny reconstruction has relied on a time-reversible model of nucleotide changes⁹. In principle, this states that the probability of a certain substitution (e.g. C>T) is equal to its inverse (T>C). In somatic mutagenesis, since the direction of change is known, assuming general reversibility of mutational probabilities fails to acknowledge the genuine discrepancies in the likelihood of certain (trinucleotide) substitutions. For example, a C>T mutation in a CpG context is much more probable than a T>C at TpG due to the specific mutational processes acting on the genome, in this case, spontaneous deamination of methylated cytosine (commonly referred to as SBS1).
- (3) *Low somatic mutation rates in a human lifespan* – When accounting for the size of the human genome, the number of mutations that are informative for purposes of phylogeny reconstruction, namely SNVs shared between two or more samples, is generally low compared to the settings of phylogenies of species or organisms. This means that the probabilities of independent, recurrent mutations at the same site or reversals of those nucleotide changes ("back mutations") are small and have negligible effects on the accuracy of phylogenetic reconstruction. Therefore, a mutation shared between multiple samples can generally be assumed to represent a single event in an ancestral cell that has been retained in all its progeny – the underlying principle of maximum parsimony.

We have added these theoretical thoughts to the manuscript (Materials and Methods, ‘Phylogeny inference’; pp. 24-26). Furthermore, we have included the comparison of tree reconstructions for maximum likelihood versus maximum parsimony methods in this section of the Methods and have included Reviewer Fig. 6 as Supplementary Fig. 1.

7. This is a small and optional thing, but I thought the APOBEC observation was very interesting and was curious if anything more could be said about it. Is there any insight to offer into evolutionary dynamics here that might at least give suggestions about mechanism? For example, is there a loss of selection against clones with mutant APOBEC signatures post-transplant, that might suggest that there is some mechanism of clearing these clones that is active in donor but missing from recipients? Alternatively, is there a rise in appearance of APOBEC mutants post-transplant that might suggest a distinct source of mutability in recipients not active in donors? I realize, though, this is a tangent and perhaps a story to explore more fully in another paper.

We believe that the unusual APOBEC mutations are acquired at some point between harvesting from the donor and post-HCT life in the recipient, rather than being related to a loss of negative selection. There are several reasons for this. Firstly, the APOBEC signature has not been seen in any other studies of mutational signatures in HSCs, including those looking at foetal HSPCs¹⁰, cord blood², healthy ageing blood², and blood cancers¹¹. Secondly, in our study, cells harbouring APOBEC mutations have persisted decades after HCT. Therefore, if there was loss of selection against APOBEC in recipients, this ‘loss’ would have to persist long-term after HCT despite immune reconstitution, which seems unlikely. Finally, in cases where the APOBEC signature may be timed it is always consistent with post-HCT acquisition (**Reviewer Fig. 7a**).

We agree that we did not adequately explore potential factors relating to the risk of APOBEC mutation acquisition beyond recipient/donor status. Therefore, we have performed a series of generalised linear mixed effects regression analyses to explore clone- or recipient-specific factors that may affect the likelihood of APOBEC activation (**Reviewer Fig. 7b**). This showed that while APOBEC activation was indeed much more likely in recipient than donor colonies ($p = 1.7 \times 10^{-8}$), there was no difference between (1) colonies within clonal expansions or ‘singletons’ ($p = 0.39$), (2) colonies with or without drivers ($p = 0.58$), (3) recipients receiving different types of conditioning ($p = 0.38$) or stem cell source ($p = 0.71$). Overall, this suggests that the factors affecting the likelihood of APOBEC activation, beyond being in some way linked to HCT recipient status, are not captured in our data.

We also analysed the genomic distribution of the observed APOBEC mutations (**Reviewer Fig. 8**; 3 pages on). We found that they were more likely to occur near cruciform inverted repeats, as described previously¹², and in regions with higher Alu density (typically active regions of the genome, also known to associate with APOBEC activity¹³). Interestingly, we found a strong association of increased APOBEC mutation density in regions with low GC content – we are not sure why this might occur, but one hypothesis is that very AT-rich regions show lower stability of the DNA duplex and might therefore more often be in a single-stranded state, the usual template for APOBEC activity.

We have now included these data in the manuscript (‘Mutation signatures and burdens following HCT’; p. 6). Reviewer Figures 7 and 8 have been included as Extended Fig. 5.

Reviewer Figure 7. Exploration of factors predicting APOBEC mutations in recipient colonies. (a) Estimated timings of branches containing >10 APOBEC mutations. Using the estimated clock-like mutation rate for HSPCs, branch start- and end-points in molecular time were converted to estimated chronological age and plotted as a vertical bar. Estimated timing of transplant for each recipient is plotted as a horizontal dashed line. All bars end above this age-of-transplant line, and many begin above it, suggesting that APOBEC mutations occur at the time of or subsequent to the transplant. (b) P values of a generalised linear mixed effects model to identify factors predicting presence of APOBEC mutations in a given colony, showing that the only significant variable was the enrichment of the signature in recipient versus donor colonies.

8. One minor error is that for Fig 4a, the figure identifies it as about Pair 3 but the caption says Pair 4.

Thank you, **we have now corrected this error.**

Referee #2 (Remarks to the Author):

The manuscript by Chapman et al. describes a long-term follow-up cohort of 10 donor and recipient pairs after allogeneic hematopoietic stem cell transplantation in which the authors have used whole genome sequencing of individually grown CD34+ clones to assess interindividual and interclonal differences in mutational load in the hematopoietic system of donor and recipient. The authors use complex statistical modeling to infer longitudinal dynamics of hematopoietic clones from this data, including selection processes preceding and following the transplantation procedure.

The methodology and research questions of this study are similar to other recently published works by the group and others (Mitchell et al., Nature 2022; Fabre et al., Nature 2022 and Williams et al., Nature 2022) which have all demonstrated the life-long selection of hematopoietic clones that can lead to premalignant and malignant transformation.

The strength of this work is that by comparing the same hematopoietic system in its native (donor) and its transplanted (recipient) form, the authors are able to demonstrate additional selective pressures induced by the transplantation process, which would have been predicted in this form by most researchers in the field.

On the other hand, in its present form, the manuscript represents an extension in the transplant context of their previous studies. I think that at this point they need to complement their well-oiled computational pipelines to infer clonal relationships with some functional data/mechanistic insides. For example, they observed an APOBEC signature only in recipients, which would be interesting to dive in, or, alternatively, another interesting avenue could be to understand why CHEK2 mutations seem to favor engraftment and/or long term survival. As it is now, this paper is a well executed exercise in style which does not provide many potentially actionable new insights.

We agree that the APOBEC signature merits further exploration. We have undertaken further analyses to address, first, the patient- and colony-specific factors that correlate with activity of the APOBEC mutational process; second, the timing of APOBEC mutations relative to transplant; and, third, the factors that predict its genome-wide distribution in those colonies:

1. *Patient and colony-specific factors* – We assessed what factors might predict APOBEC activity, and found no effects of whether the cell was in a clonal expansion or not; whether it had a driver mutation; whether the conditioning was myeloablative or reduced intensity; whether the donor was male or female; or whether the source of stem cells was bone marrow or peripheral blood (**Reviewer Figure 7b**; two pages above). In fact, the only statistically significant association was whether the colonies derived from the recipient rather than the donor (OR = 33; $p = 1.7 \times 10^{-8}$).
2. *Timing of APOBEC mutations* – Branches on the phylogenetic tree that had >10 mutations attributed to APOBEC in the recipient had an interesting distribution relative to the timing of the transplant (**Reviewer Figure 7a**; two pages above). Most of these branches spanned the time of transplant, but some were clearly restricted to a period of time after the transplant. This suggests that APOBEC mutagenesis in recipients can occur throughout the post-transplant period.
3. *Genome-wide distribution of APOBEC mutations* – We analysed which features of the genome correlated with the density of APOBEC mutations (**Reviewer Figure 8**; below). APOBEC mutations were more likely to occur near cruciform inverted repeats, as described previously¹², and in regions with higher Alu density. Interestingly, we found a strong association of increased APOBEC mutation density in regions with low GC content. The reasons for this association are unclear – one possible explanation is that the DNA duplex in high-AT areas is more likely to denature in and out of the single-stranded configuration that APOBEC acts upon.

APOBEC mutagenesis in general remains a mysterious process in human somatic cells and cancers. It appears to act in episodic bursts¹⁴, generating hundreds of mutations in a single event. APOBEC signatures are only rarely seen in normal somatic cells^{15,16}, with small intestine, bladder and bronchial epithelium being the only organs that shows appreciable levels of APOBEC signatures¹⁷⁻²⁰ – despite this, APOBEC signatures are widespread in human cancers, representing one of the most prevalent of all mutational processes¹¹.

There has been considerable speculation on what triggers these bursts of APOBEC activity in somatic cells²¹. APOBECs are an important component of the host defence systems against viruses, and there is evidence that viral infection can induce APOBEC expression and activity, potentially mediated by interferon signalling²². In the stem cell transplant setting, recipients are routinely immunosuppressed and therefore prone to a wide variety of infections that are not evident in the healthy population, so such a mechanism could explain the recipient bias and post-transplant timing we observe here.

We have now included these data in the manuscript ('Mutation signatures and burdens following HCT'; p. 6). Reviewer Figures 7 and 8 have been included as Extended Fig. 5.

Reviewer Figure 8. Genomic features significantly associated with distribution of APOBEC mutations in recipient colonies. Associations between different genomic properties (rows) and all mutations (left column), APOBEC mutations (middle column) and APOBEC mutations normalised by the density of non-APOBEC mutations (right column). Each density curve represents the quantile distribution of the genomic property values at observed positions of mutations compared to random genome positions. Shown are the genomic properties that are statistically significant after multiple hypothesis test correction ($q < 0.1$).

As a side criticism, they mention in the introduction genotoxicity from antiviral drugs as a factor potentially influencing HSC dynamics, but do not test it in their cohort, which would be highly relevant for transplant recipients

The genotoxicity of antiviral drugs in the transplant setting has been previously investigated – ganciclovir results in a specific pattern of C>A mutations at CpA dinucleotides, while most other

antiviral drugs (including foscarnet) do not²³. Given the distinctive spectrum of C>A mutations associated with ganciclovir, we would expect this to be readily extracted using our algorithms were it to be present. However, this signature was not extracted using formal statistical analysis, nor did we see any hint of increased mutations of C>A at CpA contexts in the raw mutational profiles. In light of this absence of the ganciclovir signature, we reviewed the clinical notes of our patients. None experienced CMV reactivation, and none were treated with ganciclovir. Therefore it is not surprising that we did not see any signatures associated with the drug.

We have added a sentence to the manuscript to clarify this point ('Mutation signatures and burdens following HCT'; p. 6).

The study is otherwise well-executed and the manuscript is written in a concise and well-structured manner. Other minor comments / queries:

- The recurrent loss of chromosome Y in pair 9 is suggestive of a specific selective pressure. Can the authors comment on what they believe the mechanisms behind this observation may be?

Mosaic loss of chromosome Y leading to a mix of XY/XO cells commonly occurs in whole blood during ageing, particularly in smokers. Various studies have looked at germline predisposition to mosaic loss-of-Y^{24,25}, with the 'G' allele at snp rs2887399 (within the *TCL1A* gene) consistently being identified as a major risk allele. All three males in our study were homozygous for this risk allele.

It is notable therefore that Pair 9 was the oldest male donor in our study. Interestingly, in this pair (and indeed one of the other male donors), the loss-of-Y occurred through parallel evolution in independent clones – several of these were among the largest clones in that donor-recipient pair. As the reviewer suggests, one plausible explanation for this is that the loss-of-Y confers a selective advantage on HSPCs. This selective advantage is probably independent of the transplant procedure – some of the clones with loss-of-Y expanded more in the recipient and others expanded more in the donor. Indeed, we have observed this parallel evolution of loss-of-Y clones with preferential expansion in our study of normal ageing haematopoiesis². We are unsure of the mechanism by which loss-of-Y would confer a selective advantage, but one possibility is that loss of the Y-linked tumour suppressor gene *KDM6C* could play a role based on studies in mice and humans²⁶.

We have included these points in the manuscript ('Mutation signatures and burdens following HCT'; p. 7).

- As B cells and T cells have additional clonal markers in the form of BCR/TCR, did the authors attempt to correlate clonal diversity measured by somatic mutations and BCR/TCR in B/T cells?

Our sequencing of the B and T cells was deep targeted sequencing using a custom bait set looking for the somatic mutations identified in the WGS of colonies. Unfortunately, in order to get the depth of sequencing required for informative conclusions, we used all the DNA from the samples to make the library. However, we do have the ability to assess the impact that antigen-driven expansion of T-cell or B-cell clones has on the clonal composition by interrogating the clonal fraction of early embryonic mutations, which may be significantly altered by such expansions. The idea is that if there are

oligoclonal expansions in, say, T cells then the embryonic mutations in those clones would be over-represented in the bulk sample of T cells from peripheral blood – since we have reasonably complete characterisation of the early embryonic cell divisions, we can infer the presence of such skewing even if the adult clone was not discovered in our colony sequencing of HSPCs.

To do this, we compared the clonal composition of different mature cell types within and between donors and recipients using the ‘soft cosine similarity’ measure, something we have used previously for comparing clone sizes across cell types¹⁰. The metric essentially compares the similarity of the clone sizes between two samples in such a way that two mutations that are near to each other on the phylogeny are weighted more heavily than distantly separated mutations.

The results show that age and cell type were the major determinants of how similar the clonal compositions are between donor and recipient (**Reviewer Figure 9**). The numbering of the donor-recipient pairs is from youngest to oldest donor, and we can see that for the younger pairs (Pairs 1-5), the soft cosine similarities were in general very high (often >0.95) when comparing among cell types and between donor and recipient. For the older pairs (Pairs 6-10), the soft cosine similarities were often lower due to the effects of the emerging clonal expansions. Assessing the reviewer’s question about B and T cells with these data, we can see that the similarities between donor and recipient for each cell type (black boxes in **Reviewer Figure 9**) were generally more closely aligned with one another than with the other cell types. This suggests that the clonal composition of lymphocytes in the recipient closely resembled that seen in the donor without detectable transplant-related skewing towards a few dominant clones. The one exception to this pattern was the eldest pair (Pair 10), in whom the recipient T cells were more similar to recipient myeloid and B cells than the donor T cells – this is due to a massive clonal expansion that barely contributed to donor T cells but comprised ~20% of T cells in the recipient (**Figure 3d**, main paper). Thus, this approach can detect oligoclonal skewing in the mature lymphocyte populations, but this appears to be relatively infrequent.

We have included these analyses in the manuscript (‘Clonal contributions to mature blood cell compartments’; p. 13) and Reviewer Figure 8 as Extended Figure 13d.

Reviewer Figure 9. Soft cosine similarities of early embryonic mutations across mature cell types in the 10 donor-recipient pairs.

- Were there any cases with evidence of residual recipient-derived hematopoiesis?

This is an interesting point. We found no convincing evidence of residual, recipient-derived haematopoiesis in any of the recipients. The evidence comes from (1) the colony whole genome sequencing (WGS) data and (2) the deep targeted sequencing data.

1. *Colony WGS data* – Colonies from a different germline background, as would be the case for residual recipient-derived colonies, are clearly distinguishable in the WGS data. In three recipients there was at least one colony from a different germline background to the donor. However, this was clear contamination from another pair in one case, and likely contamination from an unrelated individual in the other two cases. **We have included this information in the Methods section ('Recognition of different germline background for samples', p. 27).**
2. *Targeted sequencing data* – If recipient haematopoiesis is fully donor-derived, all cells should derive from one or other branch of the donor's early phylogeny, such that the sum of the variant allele fractions (VAFs) of these early embryonic mutations would be 0.5. This is indeed the case for all individuals (shown with VAFs converted to clonal fractions, **Reviewer Figure 10**). For pairs 9 and 10, the B- and T- clonal fractions sum to less than 1, but the recipient fraction is in fact higher than the donor fraction, suggesting that this is due to missing lineages from the donor phylogeny rather than residual recipient chimerism. **We have included this information in the Methods section ('Defining posterior distribution of post-developmental clone fractions', p. 33).**

In addition to the detailed information above included in the Methods, **we have mentioned the absence of detectable recipient-derived haematopoiesis in the main text ('Whole genome sequencing of HSPC colonies'; p. 4). Reviewer Fig. 10 has been included as Extended Fig. 1c.**

Reviewer Figure 10. Line plot showing the sum of clonal fractions across the branches of the phylogenetic tree at different points in molecular time. This is divided by pair and by cell type. The earliest time point shows the sum of clonal contributions of the first two blastomeres of the embryo. Solid line shows the median posterior values, shaded areas show the 95% posterior intervals.

Referee #3 (Remarks to the Author):

The authors characterize the historical dynamics of the hematopoietic system within ten allogeneic HSCT recipient-donor sibling pairs. They employ two central laboratory-based methods: whole-genome sequencing of ~100-200 distinct peripheral-blood-derived colony-forming units per individual, and targeted bulk sequencing of lineage-sorted cells for mutations identified in the WGS assay. Using somatic variant calls (SNVs and SVs) from standard workflows on the WGS data, the authors create HSPC phylogenies based on adjusted mutational burden that can be aligned to the chronological age of the individuals, with care taken to remove non-clonal samples, duplicated colonies, and contaminants prior to analysis. The WGS data was further used to estimate clonal diversity and contribution of common trinucleotide mutational patterns to the observed mutation landscape. The data from the targeted sequencing was used to directly estimate the clonal contribution to each of the studied lineages, and was also overlaid back onto the WGS-derived phylogenies.

Based on these analyses, the authors make the following main conclusions. First, they estimate that between 500 to 100,000 HSPCs engrafted in the recipients, with 10-fold fewer engrafted cells in older donors (more than 50 years old) than with younger donors. Second, they found that there was lower clonal diversity among recipients, corresponding to a 10-15 year greater estimated phylogenetic age. Third, they report that HSPC clones in the recipients frequently demonstrate lineage-specific biases. Fourth, in what may be considered the topline conclusion of the manuscript, the authors describe patterns of pre-transplant and peri-transplant clonal expansion, which they term “pruning” and “growth” selection, which are apparent consequences of a clone’s selective advantage in the setting of donation itself (mobilization/collection/etc.) or once placed within the body of the recipient, respectively. Fifth, the authors show that clones with the same driver mutations differ in their representation when comparing the donor and recipient. And, while somewhat tangential to the main story of clonal dynamics, the results also demonstrate APOBEC mutational signatures in recipients but not donors, suggesting an activation of a mutational process not usually found in HSPCs.

These experiments require complex analysis but the authors convey their message with thoughtful precision, as a result, the significant implications of this work are readily appreciated. Foremost, it is convincingly demonstrated that certain pre-existing clones within the donor may have a selective advantage during the donation process, and that this pruning selection may occur with similar frequency to growth selection, which will undoubtedly spur research into how to identify these clones a priori and into the mechanisms of why they have an advantage during donation. Though this has been speculated for decades, this is the first experimental rigor illustrating the phenomenon. The research also raises the provocative question of whether there are meaningful correlations to clinical outcomes from pruning vs growth selection. Furthermore, while previous studies have also sought to quantify the number of engrafted HSPCs and loss of clonality, this manuscript provides the most precise and robust figures to date, we imagine that it will likely be well-received by researchers in the fields of HSC biology, hematologic malignancy, and clonal dynamics, among others.

We thank the reviewer for these generous comments.

Major Points:

1. Understandably from a logistical perspective, the studied participants were chosen for likelihood of available sample from long intervals (~10y). However, the natural bias then is against unsuccessful transplants: pairs in which engraftment failed in the recipient, transplants in which the recipient developed subsequent hematological malignancy, and/or died (whether or not the graft contributed) were excluded. This unfortunately limits the generalizability of the conclusions to only successful allogeneic transplants and dampens the translational potential. Some of the major clinical objectives in HSCT care have been to prevent graft failure, donor-derived malignancy, and reduce mortality, yet this study cannot speak to whether/how the biological phenomena that are very elegantly described (such as “pruning” vs “growth” selection) integrate into the aforementioned areas of unmet clinical need. The authors should tackle this in the discussion more completely.

This is an excellent point, and we are rather embarrassed that we did not note this in the original submission! Outright graft failure is thankfully rare these days, with a more common outcome being poor graft function, in which counts never quite recover to normal levels leading to susceptibility to recurrent infections. As the reviewer indicates, patients with poor graft function or post-transplant relapse would be very interesting to study with these approaches – we might hypothesise that the establishment of a clonally diverse, multipotent stem cell pool would be impaired in these patients. Also interesting to study would be patients with graft-versus-host disease, in whom we might hypothesise that there would be oligoclonal expansions of the alloreactive lymphocyte lineages. **We have included these points in the Discussion (p. 18).**

2. Further, presumably the clones in the matched post-transplant sample are all donor derived (see below minor point #2)? Can the authors clearly articulate this case?

We do not believe any of the post-transplant sample colonies are recipient-derived. The evidence for this comes from both the whole genome sequencing (WGS) of the colonies and the deep-targeted sequencing data of the bulk populations of mature cells –

1. *Whole genome sequencing of colonies* – Samples with a different germline are readily detected in the WGS data by the thousands of inherited polymorphisms that differ even between siblings. In three recipients there was at least one colony from a different germline background to the donor. However, this was due to contamination from one of the other donor-recipient pairs in one instance, and contamination from an unrelated individual in the other two cases. **We have included this information in the Methods section (‘Recognition of different germline background for samples’, p. 27).**
2. *Targeted sequencing of bulk mature cells* – We can also assess how complete the donor chimerism is in the recipient by using the early embryonic mutations on the phylogenetic tree. The idea is that the relatively high number of colonies we have sequenced means that we have relatively complete identification of the first 2-4 cell divisions of the embryo. All cells in the donor derive from one or other of these embryonic cells and will therefore inherit the mutations acquired during these first few cell divisions, whereas none of the recipient cells will carry them. With complete donor

chimerism in a given mature cell population, the total contribution of donor-derived embryonic mutations should be 100%. This is indeed what we see for mutations assigned to the earliest molecular time on the phylogeny (**Reviewer Figure 10**, 3 pages above) in 8 of the donor-recipient pairs. For the other two pairs (Pairs 9 and 10), the sum is less than 1, but the recipient sums are always higher than the donor sums, suggesting that there are some early embryonic lineages that are missing from the phylogeny rather than any residual recipient haematopoiesis. **We have included this information in the Methods section ('Defining posterior distribution of post-developmental clone fractions', p. 33).**

In addition to the detailed information above included in the Methods, **we have mentioned the absence of detectable recipient-derived haematopoiesis in the main text ('Whole genome sequencing of HSPC colonies'; p. 4).** **Reviewer Fig. 10 has been included as Extended Fig. 1c.**

3. Several points of the manuscript rely on the assumption that the monocyte, B cell, and T cell populations were accurately ascertained and each lineage isolated into a pure fraction. However, no antibody information, gating strategies, nor FACS data are presented to allow for the verification of these claims. These data should be provided and the reporting summary updated.

We apologise for not including this in the original submission. **The reporting summary has now been updated and the FACS data has been added as a new supplementary figure (Extended Figure 10a, included as Reviewer Figure 11 below).**

Reviewer Figure 11. Gating strategy for single cell sorting of B-cells, T-cells and monocytes, as illustrated by the gating used for Donor and Recipient 41 (Pair 2).

4. Peripheral-blood-derived HSPCs may present a biased view of the overall stem and progenitor compartment. This might be particularly true if there is clonal hematopoiesis in epigenetic regulation genes, as expression of CXCR4, which plays a pivotal role in HSPC homing to/retention in the bone marrow niche, is highly sensitive to methylation (e.g., PMIDs 27899645 and 31018749).

Indeed, there are changes in CXCR4 expression across the hematopoietic lineages in the context of mutations in DNMT3A and TET2 (e.g., PMIDs 30890702 and 33155517). If certain clones have more (or less) CXCR4 expressed by their HSPCs, then they might be under- (over-) represented in the peripheral blood HSPCs as compared to the marrow HSPCs. This does not contradict the findings of the authors, but it does mean that their findings should not necessarily be interpreted as representing the clonal dynamics of the entire HSPC pool without proper examination of marrow-derived samples.

This is an interesting point. For the reasons mentioned, the peripherally circulating HSPC pool may not faithfully represent the ‘whole body’ HSPC pool, including those resident in the bone marrow or spleen. To address this question, we have analysed our deep targeted sequencing data to assess for biases between the clonal composition of HSPC colonies we obtained and the bulk populations of mature cells from the same patients. The clonal composition of a bulk sample will reflect the whole-body stem cell pool that is currently contributing to that mature cell type – if there is a large contribution from a set of stem cells that we have not accessed with our colony approach, then the overall allele fractions of mutations we discover will be systematically lower in the bulk population than in the combined set of colonies.

Reassuringly, we find that in most cases (~86%), the clonal fraction within mature myeloid cell lineages lies within the 95% confidence interval of the clonal fraction inferred from the phylogeny (**Reviewer Figure 12**). This implies that in most cases there is no major skewing of the peripheral HSPC pool compared to the ‘whole body’ active myeloid HSPC pool. This is perhaps not as surprising as it may initially seem. In many ways, we are using the peripheral blood HSPC-derived colonies as a vehicle for identifying historic stem cell dynamics – most of the quantitative information in the donor-recipient pairs comes from mutations and cell divisions (manifesting as branch-points in the phylogeny) that occurred years before the time of sampling.

We have included this information in the manuscript (‘Clonal contributions to mature blood cell compartments’; p. 7) and Reviewer Figure 12 as Extended Figure 12a.

Reviewer Figure 12. Clonal fractions inferred from the phylogeny compared to targeted sequencing of monocytes. Plot shows only clones that are at least 5% clonal fraction in either donor or recipient. The x-axis shows clonal fractions inferred from the proportion of colonies from that individual coming from that clone, with

error bars giving the 95% confidence interval (exact binomial test). The y-axis shows clonal fractions inferred from the deep targeted sequencing of monocyte fractions. Confidence intervals for the targeted sequencing data are generally narrow and therefore not shown.

Minor Points:

1. Consider putting age brackets for younger and older donors in the abstract, as this information is highly pertinent to the result finding 10-fold fewer engrafted HSCs with older donors.

Thank you for this suggestion which we have incorporated into the updated abstract.

2. What is the percent engraftment for each recipient? Shall it be assumed that the cells taken from a recipient 10y after a successful transplant are 100% donor-derived? What do the authors think about low fraction persistent recipient chimeras and how do they account for this in their analysis?

We do not find any detectable, low-level residual recipient chimerism. Given the number of recipient colonies we sequenced and the depth of the targeted gene sequencing, our analyses would be able to detect down to 1% residual recipient haematopoiesis with confidence, and we did not identify any such contribution. **We have included this point in the manuscript ('Whole genome sequencing of HSPC colonies'; p. 4).**

3. Were CD34+ cells quantified prior to plating in methylcellulose? Description of the colony-forming units per input for each donor-recipient pair would speak to the similarities or differences of the current stem-like capacity within each individual's hematopoietic compartment. This would be of great interest, especially given how the authors demonstrate that HSCT advances the apparent mutational age of recipient HSPCs by more than a decade, on average.

We agree that this information would have been of interest, but unfortunately these data were not collected.

4. Does the APOBEC mutational signature occur stochastically or in a biased manner? If biased, is it preferentially found in (or exclusive of) the dominant clonal populations? If in larger clones, is it in "pruning" or "growth" selection clones?

As raised by the other two reviewers, this is an interesting question. We have analysed what properties of the colonies might predict the occurrence of APOBEC activity. Interestingly, we found no enrichment of APOBEC activity whether the cell was in a clonal expansion or not; whether it had a driver mutation; whether the conditioning was myeloablative or reduced intensity; whether the donor was male or female; or whether the source of stem cells was bone marrow or peripheral blood (**Reviewer Figure 7b**; on page 13). The only statistically significant association was for colonies grown from the recipient versus the donor (OR = 33; $p = 1.7 \times 10^{-8}$).

We also studied the timing of APOBEC mutations from the branches on the phylogenetic tree that carried >10 mutations attributed to SBS2 or SBS13 (**Reviewer Figure 7a**; on page 13). Interestingly, all such branches occurred relatively late in the phylogenetic tree, consistent with the mutational process

only being active in the recipient. Most of these branches spanned the time of transplant, but some were clearly timed as post-dating the transplant by some months to years of molecular time. This suggests that APOBEC mutagenesis in recipients can occur throughout the post-transplant period.

We have now included these data in the manuscript ('Mutation signatures and burdens following HCT'; p. 6). Reviewer Fig. 7 has been included as Extended Fig. 5.

5. Page 22, Line 510: should be J. Clin. Oncol.

Thank you for spotting this error. **We have corrected it.**

6. Page 30, Line 794: Were these 10 genes the only ones considered to have "driver mutations" as the term is used in the body of the manuscript? This does not appear to be the case, given several of the figures list driver mutations not listed here (e.g., TP53 mutations). If not, please provide a list of which genes would be considered driver genes (including any that may not have been observed in the study, e.g., JAK2).

We apologise for this omission. **We have also added in a section to the methods () outlining the approach. We have therefore added a section to the Materials and Methods detailing the approach for annotating driver mutations ('Driver mutation annotation', p. 31), and added two new Supplementary Tables 1-2) that include the full gene list and the raw list of variants in those genes that were subsequently manually curated. We have also briefly explained in the main text what we are defining as drivers in the analysis ('Dynamics of driver mutations through HCT'; p. 16).**

7. Reporting summary, page 2: mentions optimizing sample size to describe, among other things, telomere lengths. No data on telomere lengths is presented in the manuscript.

We have updated the reporting summary to remove reference to telomere lengths, which, as correctly noted, is not in the manuscript. While there is data on telomere lengths from these same individuals as previously reported²⁷ we were not able to successfully infer colony-specific telomere lengths due to issues with compatibility of telomere inference algorithms and the NovaSeq sequencing platform used for generating the whole genome sequences.

References

1. Gelman, A. *et al.* *Bayesian Data Analysis*. (CRC Press, Boca Raton, FL, 2013).
2. Mitchell, E. *et al.* Clonal dynamics of haematopoiesis across the human lifespan. *Nature* **606**, 343–350 (2022).
3. Martincorena, I. *et al.* Universal Patterns of Selection in Cancer and Somatic Tissues. *Cell* **171**, 1029–1041 (2017).
4. Goldman, N. & Yang, Z. A Codon-based Model of Nucleotide Substitution for Protein-coding DNA Sequences. *Mol Biol Evol* **11**, 725–736 (1994).
5. Nei, M. & Gojobori, T. Simple methods for estimating the numbers of synonymous and nonsynonymous nucleotide substitutions. *Mol Biol Evol* **3**, 418–426 (1986).

6. Chamary, J. V. & Hurst, L. D. Evidence for selection on synonymous mutations affecting stability of mRNA secondary structure in mammals. *Genome Biol* **6**, (2005).
7. Greenman, C., Wooster, R., Futreal, P. A., Stratton, M. R. & Easton, D. F. Statistical analysis of pathogenicity of somatic mutations in cancer. *Genetics* **173**, 2187–2198 (2006).
8. Yang, Z. & Rannala, B. Molecular phylogenetics: Principles and practice. *Nature Reviews Genetics* vol. 13 303–314 Preprint at <https://doi.org/10.1038/nrg3186> (2012).
9. Tavaré, S. Some Probabilistic and Statistical Problems in the Analysis of DNA Sequences. *Lectures on Mathematics in the Life Sciences* **17**, 57–86 (1986).
10. Spencer Chapman, M. *et al.* Lineage tracing of human development through somatic mutations. *Nature* **595**, 85–90 (2021).
11. Alexandrov, L. B. *et al.* The repertoire of mutational signatures in human cancer. *Nature* **578**, 94–101 (2020).
12. Zou, X. *et al.* Short inverted repeats contribute to localized mutability in human somatic cells. *Nucleic Acids Res* **45**, 11213–11221 (2017).
13. Morganello, S. *et al.* The topography of mutational processes in breast cancer genomes. *Nat Commun* **7**, (2016).
14. Petljak, M. *et al.* Characterizing Mutational Signatures in Human Cancer Cell Lines Reveals Episodic APOBEC Mutagenesis. *Cell* **176**, 1282-1294.e20 (2019).
15. Moore, L. *et al.* The mutational landscape of human somatic and germline cells. *Nature* **597**, 381–386 (2021).
16. Li, R. *et al.* A body map of somatic mutagenesis in morphologically normal human tissues. *Nature* **597**, 398–403 (2021).
17. Wang, Y. *et al.* APOBEC mutagenesis is a common process in normal human small intestine. *Nat Genet* **55**, 246–254 (2023).
18. Yoshida, K. *et al.* Tobacco smoking and somatic mutations in human bronchial epithelium. *Nature* **578**, 266–272 (2020).
19. Li, R. *et al.* Macroscopic somatic clonal expansion in morphologically normal human urothelium. *Science (1979)* **370**, 82–89 (2020).
20. Lawson, A. R. J. *et al.* Extensive heterogeneity in somatic mutation and selection in the human bladder. *Science (1979)* **370**, 75–82 (2020).
21. Butler, K. & Banday, A. R. APOBEC3-mediated mutagenesis in cancer: causes, clinical significance and therapeutic potential. *Journal of Hematology and Oncology* vol. 16 Preprint at <https://doi.org/10.1186/s13045-023-01425-5> (2023).
22. Baker, S. C. *et al.* Induction of APOBEC3-mediated genomic damage in urothelium implicates BK polyomavirus (BKPyV) as a hit-and-run driver for bladder cancer. *Oncogene* **41**, 2139–2151 (2022).
23. de Kanter, J. K. *et al.* Antiviral treatment causes a unique mutational signature in cancers of transplantation recipients. *Cell Stem Cell* **28**, 1726-1739.e6 (2021).
24. Zhou, W. *et al.* Mosaic loss of chromosome Y is associated with common variation near TCL1A. *Nat Genet* **48**, 563–568 (2016).
25. Thompson, D. J. *et al.* Genetic predisposition to mosaic Y chromosome loss in blood. *Nature* **575**, 652–657 (2019).
26. Gozdecka, M. *et al.* UTX-mediated enhancer and chromatin remodeling suppresses myeloid leukemogenesis through noncatalytic inverse regulation of ETS and GATA programs. *Nat Genet* **50**, 883–894 (2018).
27. Boettcher, S. *et al.* Clonal hematopoiesis in donors and long-term survivors of related allogeneic hematopoietic stem cell transplantation. *Blood* **135**, 1548–1559 (2020).

Reviewer Reports on the First Revision:

Referees' comments:

Referee #1 (Remarks to the Author):

I appreciate the thoughtful responses and substantial additional work done in response to the prior reviews. I found the responses convincing on each of the points I previously raised. While I am still curious to learn more about what is going on in the APOBEC story, I am satisfied there that this has been explored in appropriate depth for the present work and that deeper insight may have to wait for future studies. As before, I am persuaded that this is an important study likely to be of wide interest. With my concerns about rigor of a few technical points now addressed, I have no further criticisms to offer.

Referee #2 (Remarks to the Author):

1) In its current form, it is extremely difficult to outsiders to understand the hierarchy of scripts and the structure of the intermediate files in the data/ subdirectory of the indicated github repository.

It would greatly improve reproducibility, if the scripts and data files were at least ordered by numbers or structured more clearly, and accompanied by some basic documentation in the github repository itself. For example, by providing a table with figures from the paper and their associated code. Right now, there is no documentation.

2) A lot of data and additional scripts are apparently sourced from directories not provided in the github repository.

The scripts repeatedly refer to data stored under `/lustre/scratch119/casm/team154pc/ms56` (likely an absolute path on the cluster used for the computational work of this manuscript).

For example, in `simulation_scripts/H SCT_age_simulation_rsimpop_farm.R` on line 155 and 156, files are systematically processed that I could not find in the github repository or in the mendeley repository.

For reproducibility purposes, it would be necessary to provide the files under `/lustre/scratch119/casm/team154pc/ms56` in the github repository as well or elsewhere.

3) The scripts in the github repository utilize a lot of preprocessed data for which it is not clear how they were generated.

For example, multiple scripts utilize the `HDP_multi_chain.Rdata` file (`Plotting_trees.R` line 77, `Mutational_signature_analysis.R` line 101 and 110, `Compile_data_objects.R` line 86). However, it is difficult to understand where this file comes from. From the methods section, I could imagine that this is output from the HDP package, but then it is

not clear how this was done.

For reproducibility, it would be necessary to also provide the scripts used for preprocessing steps, or to at least document how the preprocessed data were generated.

Referee #2 (Remarks on code availability):

The authors provide R scripts in the indicated github repository (https://github.com/m SpencerChapman/Clonal_dynamics_of_HSCT/) that were used to generate the figures of the paper. This code looks reasonably complete (it is not realistically possible for a reviewer to go through all of the code in detail), but cannot be independently reproduced since not all required scripts and files are available.

Referee #4 (Remarks to the Author):

This is an interesting and well-articulated manuscript that dissect the dynamics of long-term stem cell engraftment within ten allogeneic HSCT recipient-donor pairs. They employ whole genome sequencing of ~100-200 distinct peripheral-blood-derived colony-forming units per individual, and targeted bulk sequencing of lineage-sorted cells for mutations identified in the WGS assay. The authors create HSPC phylogenies based on adjusted mutational burden that can be aligned to the chronological age of the individuals.

Several main findings were reported: ~10,000–50,000 stem cells had engrafted and contributed to haematopoiesis in younger donors (36-58 years), at time of sampling, while 10-fold lower in older donors (64-79 years). Engrafted stem cells made multilineage contributions to myeloid, B-lymphoid and T lymphoid populations, although individual clones often showed biases towards one or other mature cell type. Recipients had lower clonal diversity than matched donors, equivalent to ~10-15 years of additional ageing. Phylogenetic trees showed two distinct modes of HCT-specific selection 'pruning selection', and 'growth selection'. They also show that clones with the same driver mutations differ in their representation when comparing the donor and recipient.

The studies were well designed and nicely done, the approach is innovative and the analyses are well thought through to lead to proper interpretation. Prior reviewer's comments were all very well addressed, and the manuscript is significantly enriched. I don't have any additional comment.

Referee #4 (Remarks on code availability):

The codes are available, however I did not install them or try to run it.

Author Rebuttals to First Revision:

Below, the reviewer comments are in blue, with our response and actions in black.

Referees' comments:

Referee #1 (Remarks to the Author):

I appreciate the thoughtful responses and substantial additional work done in response to the prior reviews. I found the responses convincing on each of the points I previously raised. While I am still curious to learn more about what is going on in the APOBEC story, I am satisfied there that this has been explored in appropriate depth for the present work and that deeper insight may have to wait for future studies. As before, I am persuaded that this is an important study likely to be of wide interest. With my concerns about rigor of a few technical points now addressed, I have no further criticisms to offer.

We thank the reviewer for these supportive comments.

Referee #2 (Remarks to the Author):

1) In its current form, it is extremely difficult to outsiders to understand the hierarchy of scripts and the structure of the intermediate files in the data/ subdirectory of the indicated github repository. It would greatly improve reproducibility, if the scripts and data files were at least ordered by numbers or structured more clearly, and accompanied by some basic documentation in the github repository itself. For example, by providing a table with figures from the paper and their associated code. Right now, there is no documentation.

Many thanks for these comments on the github repository. We agree that the code was not in a fit state for easy understanding and full reproducibility. We have been through everything and have made comprehensive improvements.

- There is a much more comprehensive README file that gives an overall guide to the use and structure of the repo, including exactly which data objects need to be downloaded separately from Mendeley Data and where they need to be placed.
- The key analyses are placed in order within numbered directories. However, each stage can be run independently as intermediate data objects are provided.

2) A lot of data and additional scripts are apparently sourced from directories not provided in the github repository. The scripts repeatedly refer to data stored under /lustre/scratch119/casm/team154pc/ms56 (likely an absolute path on the cluster used for the computational work of this manuscript). For example, in simulation_scripts/HSCT_age_simulation_rsimpop_farm.R on line 155 and 156, files are systematically processed that I could not find in the github repository or in the mendeley repository.

For reproducibility purposes, it would be necessary to provide the files under `/lustre/scratch119/casm/team154pc/ms56` in the github repository as well or elsewhere.

We accept these criticisms and have improved the data directory and removed hard-coded links in the scripts.

- We have restructured the `data/` directory in order to make it easier to navigate. There is also a guide to the structure within the README file.
- All figures can now be generated from the data provided using code that is structured into separate scripts for each figure in a clear file structure. These scripts have been consistently annotated such that they can be navigated using the RStudio 'outline' functionality.
- References to local file paths have been removed, or highlighted where they need to be changed to the relevant local paths.

3) The scripts in the github repository utilize a lot of preprocessed data for which it is not clear how they were generated.

For example, multiple scripts utilize the `HDP_multi_chain.Rdata` file (`Plotting_trees.R` line 77, `Mutational_signature_analysis.R` line 101 and 110, `Compile_data_objects.R` line 86). However, it is difficult to understand where this file comes from. From the methods section, I could imagine that this is output from the HDP package, but then it is not clear how this was done.

For reproducibility, it would be necessary to also provide the scripts used for preprocessing steps, or to at least document how the preprocessed data were generated.

The reviewer is correct that the `HDP_multi_chain.Rdata` file was generated using the HDP package.

- The analysis for running HDP is now included in a dedicated directory.
- We have included the scripts for generating all the figures and extended figures and the necessary intermediate data files (or have indicated how to download them from the Mendeley Data repository when they are too large for github, such as the `HDP_multi_chain.Rdata` file).
- We have added a section on the initial mutation-calling, tree-building code, and discussed in the README how this might be adapted to a local set up.

Referee #2 (Remarks on code availability):

The authors provide R scripts in the indicated github repository (https://github.com/mspencerchapman/Clonal_dynamics_of_HSCT/ [github.com]) that were used to generate the figures of the paper. This code looks reasonably complete (it is not realistically possible for a reviewer to go through all of the code in detail), but cannot be independently reproduced since not all required scripts and files are available.

We are confident that the extensive revisions to the github repository as outlined above, together with the considerably more detailed README file, should make the code much more reproducible and straightforward to run (and adapt as necessary).

Referee #4 (Remarks to the Author):

This is an interesting and well-articulated manuscript that dissect the dynamics of long-term stem cell engraftment within ten allogeneic HSCT recipient-donor pairs. They employ whole genome sequencing of ~100-200 distinct peripheral-blood-derived colony-forming units per individual, and targeted bulk sequencing of lineage-sorted cells for mutations identified in the WGS assay. The authors create HSPC phylogenies based on adjusted mutational burden that can be aligned to the chronological age of the individuals.

Several main findings were reported: ~10,000–50,000 stem cells had engrafted and contributed to haematopoiesis in younger donors (36-58 years), at time of sampling, while 10-fold lower in older donors (64-79 years). Engrafted stem cells made multilineage contributions to myeloid, B-lymphoid and T lymphoid populations, although individual clones often showed biases towards one or other mature cell type. Recipients had lower clonal diversity than matched donors, equivalent to ~10-15 years of additional ageing. Phylogenetic trees showed two distinct modes of HCT-specific selection ‘pruning selection’, and ‘growth selection’. They also show that clones with the same driver mutations differ in their representation when comparing the donor and recipient.

The studies were well designed and nicely done, the approach is innovative and the analyses are well thought through to lead to proper interpretation. Prior reviewer’s comments were all very well addressed, and the manuscript is significantly enriched. I don’t have any additional comment.

Referee #4 (Remarks on code availability):

The codes are available, however I did not install them or try to run it.

We thank the reviewer for these supportive comments.

Reviewer Reports on the Second Revision:

Referees' comments:

Referee #2 (Remarks to the Author):

The authors have made a very commendable effort to improve the documentation and reproducibility of their published code.

The github repository https://github.com/mspencerchapman/Clonal_dynamics_of_HSCT looks great in its current form and I recommend publication.

Referee #2 (Remarks on code availability):

The github repository https://github.com/mspencerchapman/Clonal_dynamics_of_HSCT looks great in its current form and I recommend publication.